# Lithium-ion battery components are at the nexus of sustainable energy and environmental release of per- and polyfluoroalkyl substances

Jennifer L. Guelfo [1,7] ✉, P. Lee Ferguson [2,3,7] ✉, Jonathan Beck[4], Melissa Chernick [3], Alonso Doria-Manzur [1], Patrick W. Faught[2], Thomas Flug[4], Evan P. Gray [1], Nishad Jayasundara[3], Detlef R. U. Knappe [5], Abigail S. Joyce [2], Pingping Meng[5,6] & Marzieh Shojaei[2]

Lithium-ion batteries (LiBs) are used globally as a key component of clean and sustainable energy infrastructure, and emerging LiB technologies have incorporated a class of per- and polyfluoroalkyl substances (PFAS) known as bis-perfluoroalkyl sulfonimides (bis-FASIs). PFAS are recognized internationally as recalcitrant contaminants, a subset of which are known to be mobile and toxic, but little is known about environmental impacts of bis-FASIs released during LiB manufacture, use, and disposal. Here we demonstrate that environmental concentrations proximal to manufacturers, ecotoxicity, and treatability of bis-FASIs are comparable to PFAS such as perfluorooctanoic acid that are now prohibited and highly regulated worldwide, and we confirm the clean energy sector as an unrecognized and potentially growing source of international PFAS release. Results underscore that environmental impacts of clean energy infrastructure merit scrutiny to ensure that reduced $CO_2$ emissions are not achieved at the expense of increasing global releases of persistent organic pollutants.

Per and polyfluoroalkyl substances (PFAS) are anthropogenic compounds that have been used in numerous consumer and industrial products and processes, including non-stick coatings, industrial surfactants, and aqueous film-forming foams (AFFF)[1]. Studies have noted uses of PFAS in the energy sector including windmill coatings, semiconductors, solar collectors, and photovoltaic cells[1]. Literature[2–4] and patents[5–7] also document use of PFAS as electrolytes in rechargeable, lithium (Li)-ion batteries (LiBs). LiB electrolytes must be conductive and electrochemically stable, with low volatility and flammability[8].

Ionic liquids, including the Li⁺ salt of bis(trifluoromethylsulfonyl)imide (bis-FMeSI, CAS 90076-65-6), are used as a primary or secondary LiB electrolyte[5–7]. The Li salt of bis-FMeSI is also incorporated as an anti-static agent in polyvinylidene fluoride (PVDF) composites which are used in LiBs as electrode binders and as part of the separator between the cathode and anode[9–11]. Companies that hold patents for and/or advertise production or use of bis-perfluoroalkyl sulfonimide (bis-FASI) salts (Supplementary Fig. 1) including bis-FMeSI and its longer-chain homologs (e.g., bis(pentafluoroethylsulfonyl)imide; bis-FEtSI)

¹Department of Civil, Environmental, and Construction Engineering, Texas Tech University, Lubbock, TX, USA. ²Department of Civil and Environmental Engineering, Duke University, Durham, NC, USA. ³Nicholas School of the Environment, Duke University, Durham, NC, USA. ⁴Archer Science, Lake Elmo, MN, USA. ⁵Department of Civil, Construction, and Environmental Engineering, North Carolina State University, Raleigh, NC, USA. ⁶Department of Chemistry, Eastern Carolina University, Greenville, NC, USA. ⁷These authors contributed equally: Jennifer L. Guelfo, P. Lee Ferguson. ✉e-mail: jennifer.guelfo@ttu.edu; lee.ferguson@duke.edu

for use as an electrolyte or polymer additive include 3M, Solvay, and Arkema[6,7,12]. More information is available in Supplementary Note 1.

Rechargeable LiBs are a critical component of sustainable energy infrastructure[13,14], and demand for use in electric cars and electronics (e.g., cell phones, medical devices, smart watches, laptops) is anticipated to grow exponentially over the next decade[15]. Up to 96% of bis-FMeSI is recoverable[16], but studies estimate that as little as 5% of LIBs are recycled, which could yield a projected 8 million tons of LiB waste by 2040[17]. Additionally, LiB recycling could result in bis-FASI release to the environment[4]. In sum, there is potential for widespread environmental releases of PFAS such as bis-FMeSI during the manufacture of electrolytes, fluoropolymers, and LiBs and also during product use, recycling, and disposal. LiBs are used worldwide, so this is an issue of global concern. The occurrence of bis-FMeSI at low ng L$^{-1}$ levels in European and Chinese environmental water, wastewater, and drinking water was recently confirmed[18–22], but sources of release remain unclear. A limited number of studies indicate that bis-FMeSI may not be removed during conventional treatment[22], and only recently has regulatory scrutiny of this compound emerged[23]. When coupled with past and current challenges associated with PFAS such as perfluorooctanoic acid (PFOA)[24], this illustrates the need for studies of bis-FASI occurrence, toxicity, and treatability. More information is available in Supplementary Note 1.

This study is a cradle-to-grave evaluation of the environmental impacts of bis-FASI use in LiBs. First, bis-FASI occurrence in the environment was evaluated in the United States (USA) and internationally near sites of PFAS manufacturing. Next, the aquatic toxicity of bis-FMeSI was assessed using *Daphnia magna* (*D. Magna*) and *Danio rerio* (zebrafish) models. Additionally, the occurrence of bis-FASI in use and disposal scenarios was established through screening of LiBs and landfill leachate samples. Lastly, the treatability of bis-FASIs was evaluated with sorbents commonly used in water treatment (granular activated carbon [GAC] and ion exchange [IX] resin) and under conditions relevant to advanced oxidation. Although this study focused on bis-FASIs, occurrence and treatability assessments also included other PFASs including perfluoroalkyl carboxylates (PFCAs) and sulfonates (PFSAs), perfluoroalkyl sulfonamides (FASAs), per- and polyfluoroalkyl ether carboxylates and sulfonates (PFEAs), and n:2 fluorotelomer sulfonates (n:2 FTSAs) to frame bis-FASI results within the context of more well-studied PFAS. Acronyms for all PFAS discussed herein are included in Supplementary Table 1.

## Results

A total of 75 surface water, 5 tap water, 2 groundwater, 1 snow, 15 sediment, and 21 soil samples were collected from 87 sampling locations near Cottage Grove, Minnesota (MN), USA, Paducah and Louisville, Kentucky (KY), USA, Antwerp, Belgium, and Salindres, France between January and October of 2022 (Supplementary Tables 2–4, Supplementary Figs. 2–6). Results of PFAS quantitation showed near 1 to 1 agreement between two independent laboratories (Supplementary Fig. 7). Results of field blank and duplicate analysis, as well as PFAS results other than bis-FASIs are in Supplementary Note 2.

### Minnesota field sampling

In January 2022, 13 surface water samples and 1 snow sample were collected in the Cottage Grove, MN region, near Minneapolis−St. Paul and proximal to 3M's manufacturing facility. 3M Cottage Grove has been producing PFAS since 1947 and has historically released parts per million (ppm) concentrations to the creek[25]. Bis-FMeSI was detected in all but 3 samples at concentrations of 2.04–440 ng L$^{-1}$ (Supplementary Data 1), including 6.88 ng L$^{-1}$ in snow (Fig. 1), suggesting impacts of both outfall discharge and atmospheric deposition. A total of 15 additional PFAS were detected at concentrations of 1.10 (FBSA)–3279 ng L$^{-1}$ (PFBA), including PFBA in every sample (Supplementary Data 1). The highest bis-FMeSI concentration was detected at

MN 4, which is a creek receiving outfall discharge from 3M Cottage Grove Center prior to discharging into the Mississippi River (MN4; Supplementary Fig. 8)[25].

A second sampling event was conducted in the Minneapolis−St. Paul region in June 2022 to collect 24 water samples, 4 soil samples, and 4 sediment samples (Supplementary Data 2 and 3). In surface water samples, bis-FMeSI was detected at 1.12–2437 ng L$^{-1}$ (Fig. 1). A total of 19 additional PFAS were detected at concentrations of 1.04 (perfluoro-2-ethoxypropanoic acid [PFMBA])–5501 (PFBA) ng L$^{-1}$. Detection of comparatively low levels of PFMBA and other PFEAs was unexpected since 3M is not a known producer of PFEAs. This is further discussed in Supplementary Note 3. As of January, the maximum bis-FMeSI concentration (2437 ng L$^{-1}$) was detected near the 3M outfall (MN 4). Concentrations of bis-FMeSI decreased to 4.4 ng L$^{-1}$ next to Lock and Dam No. 2 of the Mississippi River (MN 10). Aqueous concentrations of bis-FMeSI at all sites sampled in both January and June were lower in June except MN 4 where the maximum bis-FMeSI concentration was an order of magnitude greater than in January.

The range of bis-FMeSI concentrations at MN 4 is consistent with aqueous concentrations of legacy PFAS, such as PFOA and perfluorooctane sulfonate (PFOS), observed as a result of historical manufacturing activities and AFFF use, which have led to some of the most significant PFAS impacts to date[24]. June sampling also included locations -13 mi north of Cottage Grove in Lake Elmo, MN, where 4.54 ng L$^{-1}$ bis-FMeSI was detected at MN 27. Results demonstrate a wide distribution of bis-FMeSI aqueous impacts in the Minneapolis−St. Paul region. Further discussion of detections in the Lake Elmo Region and Avian exposure to PFAS in the Cottage Grove region is presented in Supplementary Notes 4 and 5 and Supplementary Fig. 9.

The June 2022 sampling event also included 4 sediment and 4 soil samples. Bis-FMeSI was detected in sediments from MN 27 (Lake Elmo) and MN 4 (near 3M outfall) at concentrations of 22.9 and 1626 ng kg$^{-1}$, respectively, and in all soil samples (MN 28–31) at concentrations of 226–2300 ng kg$^{-1}$ (Supplementary Data 3). The maximum soil concentration occurred at MN 29, north of 3M at the fence line. Similar to surface water, 23 additional PFAS were detected in sediment at concentrations of 1.17 (perfluoropentanoic acid [PFPeA]) to 22,647 (N-ethylperfluorooctanesulfonamido acetic acid) ng kg$^{-1}$, and 19 additional PFAS were detected in soil at concentrations of 3.8 (perfluoroheptanoic acid [PFHpA]) to 44,259 (perfluorodecane sulfonate) ng kg$^{-1}$. Transport potential of bis-FMeSI was evaluated by comparing its field sorption coefficient ($K_d$) to those of other PFAS. Specifically, concentrations of PFAS in sediments ($C_s$; ng kg$^{-1}$) were divided by the co-occurring surface water concentrations ($C_w$, ng L$^{-1}$) to obtain $K_d$ (L kg$^{-1}$; Supplementary Fig. 10). Log $K_d$ values generally increased with increasing chain length for PFSAs and PFCAs, as observed in previous studies[26]. The log $K_d$ value of bis-FMeSI (−0.18 L kg$^{-1}$) was similar to those of PFPeA (−0.37 L kg$^{-1}$) and perfluorobutane sulfonate (PFBS; −0.46 L kg$^{-1}$), which are highly mobile in aqueous systems.

Similar to January 2022 snow results, June 2022 soil bis-FMeSI results are suggestive of atmospheric bis-FMeSI deposition. The highest bis-FMeSI concentration in soil was observed at the 3M fence line (MN 29), and concentrations decreased in the dominant wind direction (Fig. 1). Even at distances of >6 km downwind of 3M (MN 31) and despite low sorption potential described above, 227 ng kg$^{-1}$ of bis-FMeSI was present in surface soil. This suggests a widespread distribution of bis-FMeSI in surface soils of the Minneapolis−St. Paul region as a result of stack emissions, and further investigation of bis-FMeSI in air emissions from 3M, soil, and underlying groundwater in the region is warranted.

### Kentucky field sampling

In September 2022, 24 surface water, 4 sediment, and 8 soil samples were collected near Paducah and Louisville, KY proximal to Arkema

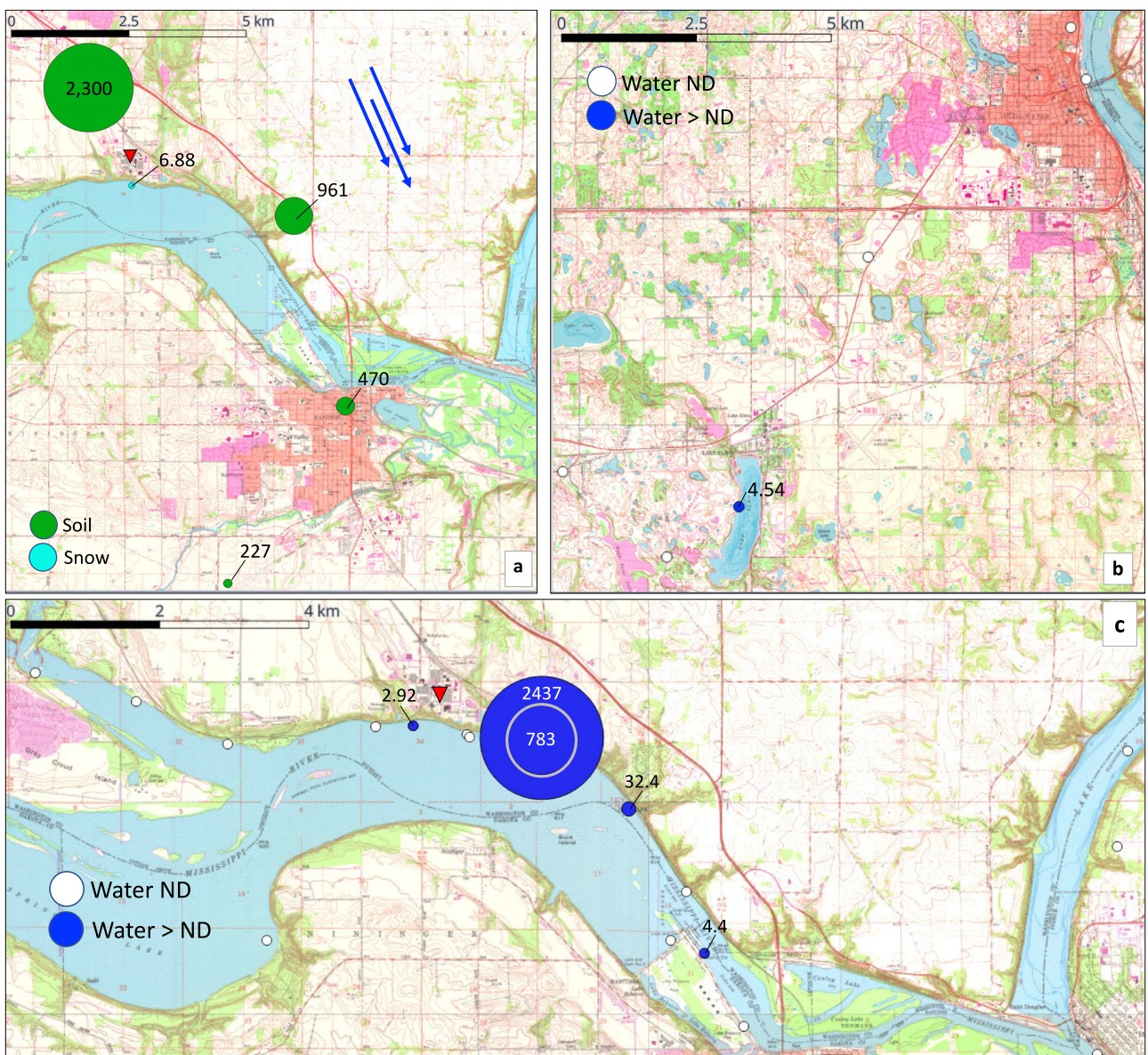

**Fig. 1 | Minnesota field data.** Concentrations of bis(trifluoromethylsulfonyl)imide (bisFMeSI) linked to atmospheric deposition in snow (January; ng L⁻¹) and soil (June; ng kg⁻¹) 2022 (**a**); concentrations of bisFMeSI in surface water (June; ng L⁻¹) and sediment (ng kg⁻¹) in the northern (**b**) and southern (**c**) sampling regions in June 2022. The concentric circle reflects concentration at MN 22, which is immediately downstream of MN 4. Blue arrows are the primary wind direction based on local windrose data[68], and red markers denote the location of 3M Cottage Grove. Base maps are from the United States Geological Survey. ND non-detect. Source data can be found in Supplementary Data 1–3.

production facilities. The maximum bis-FMeSI detection in water was 2.69 ng L⁻¹ (KY 21, Supplementary Data 4). Surface water samples contained 16 additional PFAS at concentrations of 1.25 (per-fluorodecanoic acid [PFDA]) to 19.7 (PFOS) ng L⁻¹. A single sediment sample collected in the Louisville region near the location of an Arkema outfall on the Ohio River (KY 20) contained 271 ng kg⁻¹ bis-FMeSI. However, soil and sediment samples collected in and around the Tennessee River near Paducah, KY contained a clearer archive of PFAS release (Supplementary Fig. 11, Supplementary Data 5). In contrast to water samples, PFAS concentrations in soil and sediment reached parts per million levels. This is likely attributable to transport properties (e.g., sorption), changes in PFAS use over time, dilution of manufacturing discharges in the river, and intermittent discharge release. Sediment samples collected at two observed outfalls adjacent to the Arkema facility (KY 12, 13) had a distinct long-chain PFCA signature including perfluorononanoic acid (PFNA; 1920–35,000 ng kg⁻¹),

perfluoroundecanoic acid (PFUnA; 1250–12,200 ng kg⁻¹), and per-fluorotridecanoic acid (PFTrDA; 1310–11,700 ng kg⁻¹). Concentrations of PFAS in soils surrounding the Arkema facility were 1–2 orders of magnitude higher than sediment with maximum detections of 308,000 ng kg⁻¹ PFNA, 1,380,000 ng kg⁻¹ PFUnA, and 1,840,000 ng kg⁻¹ PFTrDA in soil near the Arkema property line (KY 23). To our knowledge, these concentrations of C9–C13 PFCAs are the highest soil and sediment concentrations documented in the peer-reviewed literature[27]. Co-occurrence of these compounds with bis-FMeSI suggests a common source, as further discussed below.

The primary use of PFNA, which has been phased out in the USA, was as a PVDF manufacturing aid[28]. For example, the PVDF manufacturing aid Surflon S-111 contained high concentrations of PFNA, PFUnA, and PFTrDA[28]. Notably, the Arkema facility adjacent to the sampling locations produces Kynar PVDF, which is used in LiBs[10,29]. Bis-FMeSI and its homolog bis-perfluorobutanesulfonimide (bis-FBSI) co-

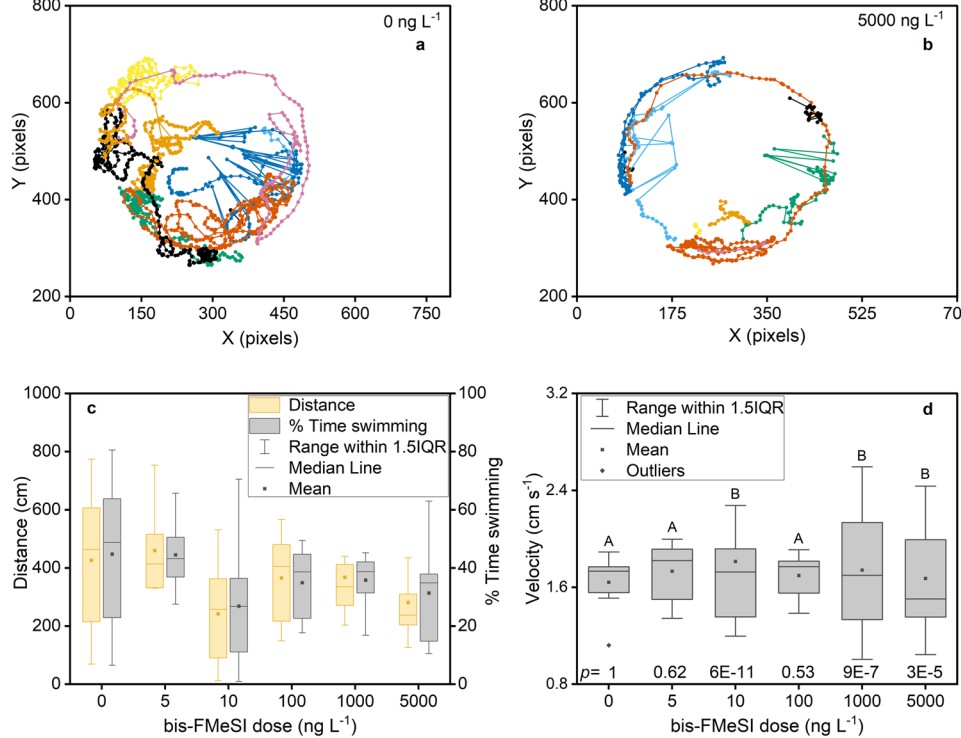

**Fig. 2 | *D. Magna* swimming track density and metrics.** One minute of representative swimming track density for each of 10 *D. magna* replicates exposed to 0 (**a**) and 5000 (**b**) ng L$^{-1}$ bis(trifluoromethylsulfonyl)imide (bis-FMeSI). Swimming parameters of distance and percent time swimming (**c**), and swimming velocity (**d**). In box and whisker plots, the lower and upper extent of each box represent the 25th and 75th percentiles, and the whisker represents the 1.5 interquartile range (1.5 IQR). Data beyond the 1.5 IQR are considered outliers. Letters reflect statistical differences in the variance of treatments vs. the control based on $X^2$ tests ($p \leq 0.002$). Note that $n = 10$ *D. magna* per dose except 5 ng L$^{-1}$ ($n = 7$), 1000 and 5000 ng L$^{-1}$ ($n = 9$) where organisms were immobilized or died prior to data collection. Source data can be found in Supplementary Data 8 and in the zip file Track Density Raw Data Full.zip.

occurred in sediment at concentrations up to 317 ng kg$^{-1}$ (KY 13) and 234 ng kg$^{-1}$ (KY 12), respectively, and in soils at concentrations up to 846 ng kg$^{-1}$ (KY 26) and 1130 ng kg$^{-1}$ (KY 30), respectively. As previously noted, bis-FMeSI is used in PVDF composites, and we have confirmed the occurrence of bis-FMeSI in PVDF battery binder (see the section "Disposal evaluation"). These data also demonstrate that the release of bis-FMeSI and homologs into the environment is linked to multiple companies and products (e.g., battery electrolyte, fluoropolymer additive).

## Europe field sampling

In October 2022, 21 surface water, 11 sediment, and 9 soil samples were collected near bis-FMeSI production facilities in Antwerp Belgium (3M), and Salindres, France (Rhodia-Solvay) (Supplementary Figs. 5, 6). Most notable in this sampling region was the detection of bis-FMeSI along with longer-chain homologs in surface water, sediment, and soil. In Antwerp, bis-FMeSI was detected in 50% of surface water samples at concentrations of 1.0–81.9 ng L$^{-1}$ (Supplementary Fig. 12) and bis-FEtSI in two samples at concentrations of 6.44 and 65.7 ng L$^{-1}$ (Supplementary Data 6). Bis-FMeSI and/or its homologs were detected at a $\Sigma_3$bis-FASI concentration of 101–10,746 ng kg$^{-1}$ in all sediments from the Antwerp region with the exception of EU 1, and in all soil samples at concentrations of 43–650 ng kg$^{-1}$ (Supplementary Fig. 13, Supplementary Data 6). Of the three homologs evaluated, bis-FBSI was the dominant homolog in soils and sediments having bis-FASI detections. A total of 30 additional PFAS were detected, and PFOS exhibited the maximum concentration in all media at 145,000 ng L$^{-1}$ (EU 17), 23,000,000 ng kg$^{-1}$ (EU 17), and 164,000 ng kg$^{-1}$ (EU 15) in surface water, sediment, and soil, respectively (Supplementary Data 6 and 7). The highest PFAS concentrations were generally observed in water collected from a ditch southwest of the 3M facility in Antwerp (EU 17),

located within an area designated by the Flemish government as a "red zone" (i.e., highest) for PFAS impacts from 3M[30]. These data confirm that bis-FMeSI releases via aqueous and atmospheric discharge are a worldwide concern, and as with legacy PFAS (e.g., PFCAs, PFSAs), data show that multiple bis-FASI homologs are environmentally relevant with potential for broad geographic distribution. These results are supported by the outcomes of sampling in the Salindres region (Supplementary Note 6 and Supplementary Fig. 14).

### *Daphnia magna* toxicity

The toxicity of bis-FMeSI to *D. magna* was assessed using acute and sub-lethal swimming endpoints based on environmentally relevant exposure concentrations. *D. magna* swimming performance was assessed by measuring swimming track density, distance, velocity, and percent time swimming. Swimming track densities show control *D. magna* utilizing greater test chamber area relative to organisms exposed to 5000 ng L$^{-1}$ bis-FMeSI which stayed near chamber walls (Fig. 2). Individual swimming metrics (Supplementary Data 8) were inversely correlated with increasing bis-FMeSI dose (Supplementary Fig. 15). Velocity had no relationship with dose (Supplementary Fig. 15), consistent with inverse correlations with distance and percent time swimming (i.e., at a constant velocity, a decrease in swimming time would lead to decreases in the distance). As a result of high variance, no bis-FMeSI dose led to significant differences in distance, velocity, or percent time swimming relative to controls (ANOVA, $n = 10$, $p > 0.05$; Fig. 2, Supplementary Data 9). Swimming metric data were normally distributed with equal variance except velocity which had unequal variance that increased with dose. Results of $X^2$ tests demonstrated that velocity variances at doses of 10, 1000, and 5000 ng L$^{-1}$bis-FMeSI were different from control variance. Prior studies have found heterogenous variance in the absence of significant changes to the mean

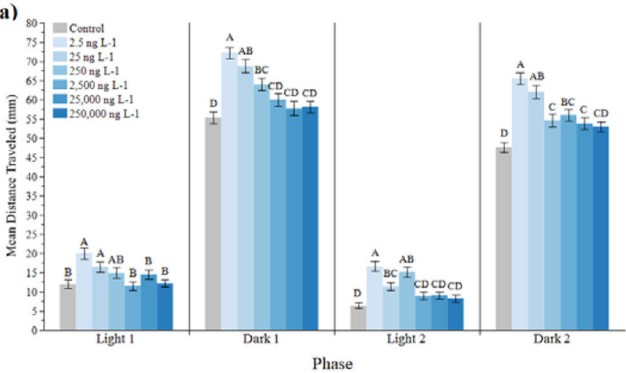
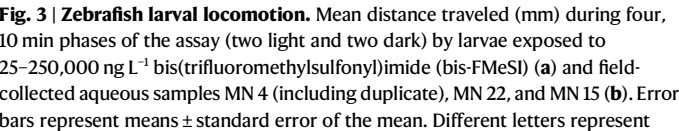
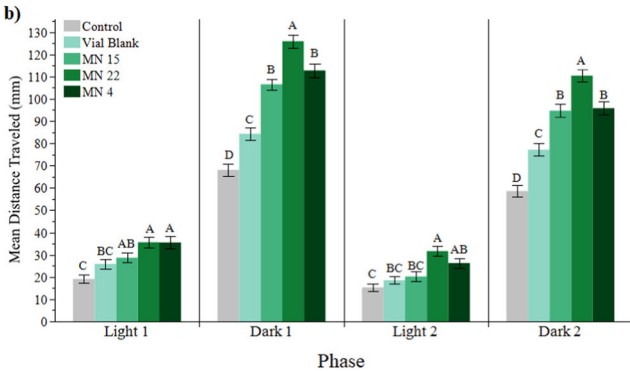

**Fig. 3 | Zebrafish larval locomotion.** Mean distance traveled (mm) during four, 10 min phases of the assay (two light and two dark) by larvae exposed to 25–250,000 ng L$^{-1}$ bis(trifluoromethylsulfonyl)imide (bis-FMeSI) (**a**) and field-collected aqueous samples MN 4 (including duplicate), MN 22, and MN 15 (**b**). Error bars represent means ± standard error of the mean. Different letters represent statistical differences within each parameter (one-way ANOVA or Kruskal–Wallis $p < 0.05$; $n = 30$ zebrafish per dose). Individual $p$ values are provided in Supplementary Data 25 (**a**) and Data 26 (**b**). A version of these plots overlaid with individual data points is in Supplementary Fig. 20. Source data can be found in Supplementary Data 21–29.

can be an earlier and more sensitive indicator of toxicological effects in 48-h *D. magna* exposures[31]. Prior studies have noted that impacts to *D. magna* swimming behavior often indicate a neuroactive effect because swimming in *D. magna* is controlled by the nervous system[32]. Although the mechanism of action would require further confirmation, results here indicate that bis-FMeSI has an effect on swimming velocity at concentrations as low as 10 ng L$^{-1}$, consistent with levels broadly detected in monitoring regions included in this study. To our knowledge, no prior studies have investigated the sublethal impacts of PFAS on *D. magna* swimming behavior, and results demonstrate the value of sublethal *D. magna* testing as a method of screening toxicity at the ng L$^{-1}$ level. Additional discussion of velocity variance, *D. magna* water quality parameters, measured exposure concentrations of bis-FMeSI, and lethality endpoints are in Supplementary Note 7, Supplementary Fig. 16 and Supplementary Table 6.

### *Danio rerio* toxicity
Survival, growth, developmental teratogenicity, mitochondrial function in embryos at 30 h post-fertilization (hpf) and larval locomotion at 6 days post-fertilization (dpf) were used to evaluate the impacts of bis-FMeSI exposure on zebrafish at concentrations of 2.5–250,000 ng L$^{-1}$ (Fig. 3; Supplementary Figs. 17–20; Supplementary Data 10–26). Larval locomotion at 6 dpf was also assessed in zebrafish exposed to field-collected samples from MN 4 and MN 22. These contained 2437 and 783 ng L$^{-1}$ of bis-FMeSI, respectively, but the exposure was conducted at 50% dilution. MN 15 did not have detectible levels of bis-FMeSI and was used as a field control representing other (e.g., co-occurring contaminant) exposures. Bis-FMeSI exposure induced significant changes in embryonic mitochondrial function. Specifically, bis-FMeSI concentration-dependent decreases in basal mitochondrial respiration, proton leak, spare capacity, and maximal mitochondrial respiration were detected in 30 hpf embryos. Similar effects were observed in previous studies for PFAS and PFOS but at much higher concentrations[33]. Additionally, bis-FMeSI exposures resulted in a non-monotonic larval (6 dpf) locomotion response, with hyperactivity in all phases at almost all concentrations (Fig. 3). Larva exposed to the lowest concentrations (i.e., 2.5 and 25 ng L$^{-1}$) was the most hyperactive. Field water also caused larval hyperactivity (Fig. 3). Although MN 15 did not have measurable bis-FMeSI, other PFAS (PFBA, PFPeA, PFHpA, and PFBS) were detected at a summed concentration of 130 ng L$^{-1}$ (1.14 ng L$^{-1}$ PFPeA to 116 ng L$^{-1}$ PFBS; Supplementary Data 2), and all field samples may have contained other compounds besides the targeted PFAS, potentially contributing to the behavioral effects detected. Prior studies have found individual PFAS and their mixtures induce hyper- and hypoactivity in zebrafish[34]. Similar to *D. magna*,

zebrafish experience neurobehavioral impacts from bis-FMeSI exposure at 10 s of ng L$^{-1}$. Relationships between mitochondrial function and zebrafish behavior are further described in Supplementary Note 8 and Supplementary Fig. 19. No effects on survival, teratogenicity and growth were detected (Supplementary Note 8, Supplementary Figs. 17, 18).

### Consumer products evaluation
This study included analysis of a variety of modern, consumer-grade LiBs to determine the prevalence and mass loadings of bis-FMeSI, bis-FEtSI, and bis-FBSI. Eleven of 17 batteries contained bis-FASIs at mass loadings above the analytical detection limit (5 ng per battery). Mass loadings were variable ranging from 7.2 ng to 35.6 mg across a range of battery types and sizes (Supplementary Table 7). The presence of 21.8 µg bis-FMeSI in a 21 g Ultrafire 14,500-type battery and >35 mg in a 47 g Samsung 18650-type battery suggests use as a primary electrolyte. Lower quantities (e.g. <1 µg, Supplementary Table 7) of bis-FMeSI were present in several other batteries, which is not readily rationalized by electrolyte use, so occurrence due to use as an additive in a polymer (e.g. PVDF) electrode binder is more likely[9,35]. A commercial PVDF battery binder was also analyzed, and bis-FMeSI was present at 385 ng g$^{-1}$ (Supplementary Table 7), confirming the potential for bis-FMeSI to occur in LIBs as a result of multiple applications at concentrations that span orders of magnitude.

### Disposal evaluation
Only a small fraction of LiBs are recycled, so the fate of bis-FASIs during product disposal in landfills at end-of-life is of concern. This study included an analysis of bis-FASIs in two leachates collected in 2022 from different municipal landfills in central North Carolina. Bis-FMeSI was present at concentrations of 195–881 ng L$^{-1}$ in untreated landfill leachate (Supplementary Table 7). The only other bis-FASI homolog detected in leachate was a single detection of bis-FBSI (5.3 ng L$^{-1}$; Supplementary Table 7) Although the source of the bis-FMeSI in leachate cannot be confirmed, LiB disposal is a likely source. Further, increasing LiB use and disposal may lead to increases in leachate bis-FMeSI concentrations over time.

### Adsorptive treatment
Collectively, the occurrence, toxicology, and lifecycle evaluations in this study suggest that bis-FASI treatment approaches capable of achieving low ng L$^{-1}$ levels in drinking water are needed. In general, PFAS treatment approaches are recognized as a critical need, but most studies have evaluated the treatment of legacy PFAS (e.g., PFCAs and PFSAs)[36], and to our knowledge, none have included bis-FASIs. As a

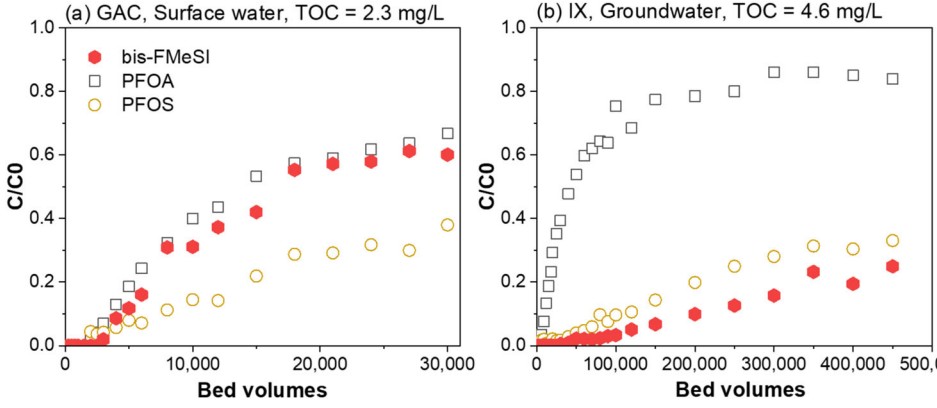

**Fig. 4 | Treatability of bis-FMeSI by GAC and IX.** Normalized breakthrough ($C/C_0$) curves of bis(trifluoromethylsulfonyl)imide (bis-FMeSI), perfluorooctanoic acid (PFOA) and perfluorooctane sulfonate (PFOS) obtained with granular activated carbon (GAC) in coagulated, settled surface water with a TOC concentration of 2.3 mg L$^{-1}$ (**a**) and ion exchange resin (IX) in groundwater with a TOC concentration of 4.6 mg L$^{-1}$ (**b**). The IX rapid small-scale column test (RSSCT) was also conducted with the coagulated, settled surface water, but no meaningful (<10%) bis-FMeSI breakthrough was observed. Breakthrough curves of all 21 PFAS are shown in Supplementary Figs. 21 and 22. Source data can be found in Supplementary Data 29. TOC total organic.

result, the fate of bis-FMeSI was screened using commonly applied treatment approaches to understand bis-FMeSI treatability relative to more well-studied PFAS.

Rapid small-scale column tests (RSSCTs) are commonly conducted to evaluate PFAS removal by GAC and IX in fixed-bed adsorbers[37]. To compare the treatability of bis-FMeSI to that of regulated PFAS (e.g., PFOA and PFOS) and emerging PFAS (e.g., PFEAs), the sorptive removal of 21 PFAS was evaluated using GAC and IX RSSCTs. PFAS were spiked at a nominal concentration of 80–100 ng L$^{-1}$ per PFAS into coagulated and settled surface water with a total organic carbon (TOC) concentration of 2.3 mg L$^{-1}$. In the GAC RSSCT, the breakthrough of bis-FMeSI closely matched that of PFOA (Fig. 4), PFHxS and PFO3OA (Supplementary Fig. 21, Supplementary Data 30 and 31), and occurred earlier than PFOS (Fig. 4). Specifically, bis-FMeSI, PFOA and PFOS reached 10% breakthrough after treating ~5000, 4000, and 8000-bed volumes of water, respectively. This indicates that GAC fixed-bed adsorbers designed for removing PFOA and PFOS can effectively remove coexisting bis-FMeSI, provided influent concentrations are comparable.

Compared to GAC, IX more effectively removed bis-FMeSI; no meaningful breakthrough (<10%) was observed after treating 200,000 BVs of the surface water. As a result, the IX RSSCT was repeated using groundwater with a higher TOC concentration (4.6 mg L$^{-1}$), which led to earlier PFAS breakthroughs as a result of IX resin fouling[38]. Bis-FMeSI breakthrough occurred later than PFOS in the high-TOC water (Fig. 4), indicating bis-FMeSI possessed the highest affinity for the evaluated IX resin among the 21 studied PFAS (Supplementary Fig. 22, Supplementary Data 30 and 32). Removal of long-chain PFAS such as PFOA and PFOS in IX resin likely results from a combination of hydrophobic and electrostatic interactions[39]. However, the log $K_d$ value of bis-FMeSI is similar to that of PFPeA (Supplementary Fig. 10) suggesting hydrophobic interactions are relatively weak for bis-FMeSI. Instead, the high affinity of bis-FMeSI for the IX resin is reflective of strong specific interactions characteristic of ionic liquids[40]. RSSCT results suggest that adsorption-based treatment systems can effectively remove bis-FMeSI and many coexisting PFAS.

**Oxidative treatment**

Three bis-FASI homologs (bis-FMeSI, bis-FEtSI, and bis-FBSI) were subjected to alkaline, heat-activated, persulfate oxidation to screen fate during advanced oxidation. All homologs were quantitatively recovered indicating that bis-FASIs are resistant to oxidative degradation (Supplementary Table 8, Supplementary Fig. 23). These results have implications for PFAS environmental fate, treatment,

and analysis. Although confirmation is needed, these results strongly suggest that bis-FASIs also will not undergo degradation in the environment under much weaker biotic or abiotic conditions. High persistence and mobility (i.e., weak sorption; Supplementary Fig. 10) of bis-FMeSI may meet the criteria for classification as a very persistent very mobile (vPvM) compound under recently proposed European Commission hazard classifications[41]. Bis-FMeSI has similar persistence and mobility as PFBS which is a vPvM PFAS recently designated under European guidelines as a substance with an equal level of concern as persistent, bioaccumulative, and toxic substances[41]. More details are in Supplementary Note 9. Additionally, results suggest that bis-FMeSI will not be effectively treated by routine oxidation approaches, similar to other PFAS. To achieve compound destruction, it will be necessary to evaluate high-energy treatment approaches such as supercritical water oxidation[42] or electrochemical techniques[43] for applicability to bis-FASIs. Lastly, as further described in Supplementary Note 9, these results are consistent with limited prior studies of bis-FASI transformation and have implications for routine PFAS analytical approaches that rely on oxidative conversion of PFAS.

**Discussion**

This study demonstrates an international release of LiB-associated PFAS (bis-FASIs, particularly bis-FMeSI) to soil, sediment, and surface water and that concentrations of these compounds in the parts per billion are common, near manufacturing areas. When coupled with low-level detections in three Chinese seawater samples[19] and characteristics consistent with vPvM classification, this suggests bis-FASI release is global. Furthermore, atmospheric emission of bis-FMeSI, as suggested by the MN data, may facilitate long-range transport of this subclass of PFAS. Toxicity data demonstrated that bis-FMeSI could change behavior and fundamental energy metabolic processes of aquatic organisms at low ng L$^{-1}$ levels, suggesting that even relatively low-level concentrations will be of concern for aquatic vertebrates and invertebrates. In addition, the use of bis-FMeSI and other PFAS in LiB-enabled consumer products will lead to environmental contamination at end-of-life disposal (i.e., municipal solid waste landfills). Lastly, due to recalcitrance and solubility, bis-FASIs pose similar treatment challenges as those associated with other PFAS although adsorptive treatment approaches designed to remove PFOA and PFOS are expected to effectively remove bis-FASIs. Where these treatment approaches are not in place or are not engineered for bis-FASI removal, the potential for exposure may be elevated. This underscores a need for additional studies of bioaccumulation and human health impacts.

In general, the challenges associated with bis-FASI occurrence, mobility, ecotoxicity, and recalcitrance are similar to those that have been realized for other PFAS; however, the potential for ongoing and increasing release of bis-FASIs resulting from exponentially growing demand for LiBs is distinct. Further, other uses of bis-FASIs (e.g., $CO_2$ capture, solar cells; see Supplementary Note 1) should be evaluated to determine their contribution to ongoing release. Without changes in manufacturing, use, disposal, and treatment practices, concentrations of bis-FASIs in soil, groundwater, surface water, wastewater residuals, and landfill leachate are likely to increase, along with associated human and environmental exposure. It is important to emphasize that bis-FASIs are not currently regulated anywhere in the world, so there is a lack of regulatory drivers to catalyze changes needed to mitigate these exposures.

There is no universal definition of PFAS, but bis-FMeSI is considered to be a PFAS according to two of three commonly referenced definitions (see Supplementary Note 1)[44,45]. The USEPA recently added PFAS as a class to a list of unregulated contaminants that will be monitored in drinking water across the United States[46] and provided a list of 10,239 PFAS[47] that meet the definition of PFAS used in the study. Bis-FMeSI does not appear on the list despite inclusion on another USEPA PFAS list[48]. In contrast, the bis-FEtSI and bis-FBSI homologs are included along with longer homologs not studied here and other PFAS structurally similar to bis-FMeSI (e.g., trifluoro(trifluoromethoxy)methane; CASRN 1479-49-8). Failure to categorize bis-FMeSI as PFAS, therefore, seems arbitrary and without scientific rationale based on considerations such as occurrence, fate, and toxicological effects. Further, the exclusion of bis-FMeSI paves the way for continued, unregulated emissions of a global contaminant into the environment.

Aqueous emissions of bis-FASIs, as observed in this study, are particularly concerning. Given that maximum contaminant levels (MCLs) were recently issued for six PFAS at low ng L$^{-1}$ levels in the US[49], and regulatory interest is increasing internationally[50], the use of GAC and IX is likely to increase rapidly. Immediate consideration of bis-FMeSI as a removal target would capitalize on the ongoing design and installation of infrastructure at water treatment facilities in response to increasing regulation and provide an opportunity to include proper engineering considerations during system design for regions where bis-FASIs are of concern. Bis-FASIs in concentrated wastes (e.g. landfill leachates and spent IX/GAC) will require destructive treatment for complete mineralization, and research is needed to evaluate the efficacy of such methods.

Bis-FASIs are associated with both LiB electrolytes and fluoropolymers (e.g., PVDF). There has been a recent debate within the practitioner and research communities regarding the designation of fluoropolymers as polymers of low concern (PLCs)[51]. Concerns with this designation include leaching of low molecular weight PFAS from fluoropolymers. The presence of bis-FMeSI in PVDF in this study supports those concerns. Results of this study provide a clear indication that the impacts of manufacturing, use, and end-of-life management associated with infrastructure components such as LiBs require additional consideration along with other issues such as resource recovery. This includes environmental impacts associated with other fluorinated, but non-PFAS LiB electrolytes $PF_6^-$ and $BF_4^-$ detected in drinking water[21] and with metals (e.g., Cu, Al) used in LiB electrodes, cables, and battery packs[52]. Researchers are in the early stages of studying fluorine-free electrolytes[53], and although promising, alternative materials also merit close evaluation of potential environmental and human health risks. Rigorous lifecycle assessments are needed to ensure that reduced $CO_2$ emissions are not outweighed by increasing global releases of persistent organic pollutants. Failure to do so may lead to a classic case of regrettable substitution and a missed opportunity to maximize sustainability and improve environmental health.

## Methods

This study complies with all relevant ethical regulations. The zebrafish studies conducted in this manuscript were approved by Duke University Institutional Animal Care and Use Committee, Protocol Registry Number A069-22-04

### Materials

Standards of all measured PFAS and their isotopically labeled standards for use as internal standards (IS, Supplementary Table 1) were purchased from Wellington laboratories, with the exception of standards for bis perfluoroalkyl sulfonimides (bisFASIs), which were purchased from Sigma Aldrich and branched perfluoroalkyl ether acid standards which were provided by Chemours. All standards were prepared in liquid chromatography–mass spectrometry (LC–MS) grade methanol (MeOH; Honeywell). Potassium persulfate (Sigma-Aldrich), sodium hydroxide (Fisher Scientific), ammonium acetate (BRAND), Envi-carb™ (envicarb; Supelco), ammonium hydroxide (BRAND), hydrochloric acid (HCl; Thermo Fisher Scientific), and acetic acid (Honeywell) were acquired from Fisher Scientific and VWR, USA.

### Sample collection and transport

In general, samples were collected in rivers (Mississippi, Ohio, Tennessee, and Scheldt Rivers) and streams (L'Arias and L'Avene) that are or are suspected to be receiving outfall discharges from major fluorochemical manufacturers (e.g. 3M, Arkema, and Solvay) and in the surrounding regions. In total, 31 surface water samples, 1 snow sample, 5 tap water samples, 2 groundwater samples, 4 sediment samples, and 4 soil samples were collected in January and June 2022 from 31 locations near Cottage Grove, MN (Supplementary Table 2). Additionally, a total of 24 surface water samples, 4 sediment samples, and 8 soil samples were collected from 32 locations in Paducah, KY and Louisville, KY (Supplementary Table 3). Finally, a total of 21 surface water samples, 11 sediment samples, 9 soil samples, and 1 foam sample were collected from 25 locations in Antwerp, Belgium and Salindres, France in October 2022 (Supplementary Table 4).

It was necessary to install ice cores at all MN locations in Pool 2 of the Mississippi River prior to the collection of surface water samples. An 8" diameter ice auger was used to core through ice at all locations. Because of subfreezing temperatures, the auger could not be cleaned between locations, but as described below, samples were collected well below the water surface. Additionally, the most contaminated sample locations were completed last. All project surface water samples were collected into new high-density polyethylene (HDPE) containers that were triple rinsed with LC–MS grade MeOH and air dried, and samplers were wearing clean, nitrile gloves that were changed between each sampling location. Surface water and deep water sediment samples (i.e., that could not be reached by trowel) were collected using an extendable dipper rod attached to an HDPE sample cup. All surface water samples collected in January 2022 were collected ~3 ft below the water surface to minimize the potential for cross-contamination due to the ice auger. In all sampling events, the sample cup was triple-rinsed with water from that sample location prior to sample collection. The snow sample was scooped directly into a clean HDPE container. Soil samples and shallow sediment samples were collected with a stainless steel trowel that was triple-rinsed with MeOH and DI water between sampling locations. Tap water and groundwater (i.e., a private well) samples were collected from the taps of the homes they served directly into the sampling container. Samples were stored on ice and shipped to laboratories at Texas Tech and Duke Universities. Sample amounts remaining after preparation for analysis were frozen as archives. Each sampling trip also included the collection of duplicates and field blanks, as discussed in the results.

All samples were collected from publicly accessible sampling locations. Soil and sediment samples collected in the US were prepared and extracted as described below in the US European soil and

sediment samples were extracted using an identical protocol in Europe to avoid concerns associated with the export of international soils. In accordance with the extraction protocol soil and sediment extracts were evaporated to dryness. When conducted in Europe, the extraction procedure was halted at this stage to avoid shipping hazardous substances (i.e., solvents). Reconstitution and cleanup of extracts were then completed after receipt of dried extracts in the US.

## Soil and sediment extraction

Soil and sediment extraction was completed in accordance with existing methods[54,55]. Specifically, 0.5–1.5 g (wet weight) of soil or sediment was weighed into 50 mL polypropylene (PP) centrifuge tubes and spiked with 2–9.1 ng of each IS (Supplementary Table 1). A 7 mL aliquot of basic MeOH (1% [V/V] ammonium hydroxide in MeOH) was added to the PP tube, vortexed (30 s), placed in a heated sonication bath (60 °C, 1 h, VWR, 97044-006), and placed on a horizontal shaker for two hours. Samples were then centrifuged (3180×$g$, 20 min), and the supernatant was transferred into a clean, 20 mL glass scintillation vial. The extraction steps were repeated twice for a total of three rounds, and the combined supernatant was evaporated under nitrogen (Organomation Associates Inc. N24EVAP, Berlin, MA), and reconstituted in 700–1400 μL acidic methanol (1% [V/V] acetic acid in methanol). The extract was transferred to a microcentrifuge tube containing 20–40 mg envicarb for clean-up. The microcentrifuge tube was vortexed (30 s) and centrifuged (19,800×$g$ for 30 min). An aliquot of 29.75–119 μL of the supernatant was transferred to an autosampler vial and amended with 731–820.20 μL of LC-MS grade methanol and 840 μL ultrapure water to achieve a final vial composition of 50% water and 50% methanol, containing 100–400 ng L$^{-1}$ of each IS.

## Aqueous sample preparation

Aqueous samples were prepared by serial dilution for direct injection. Samples were diluted 1–8× in autosampler vials with ultrapure water to which an equal volume of LC-MS grade MeOH containing the IS mixture was added for a final dilution of 2–16× containing 200 ng L$^{-1}$ of each IS.

## Battery extraction

A Soxhlet extraction technique was used for LiBs. Specifically, a 33 mm × 94 mm disposable thimble was positioned in the Soxhlet apparatus. The apparatus was then fitted with a condenser and placed on a 250 mL boiling flask containing -150 mL methanol and 3–4 clean boiling stones. The boiling flask was heated for -6 h using heating a mantle connected to a variable autotransformer. After 6 h, a membrane from a disassembled LiB was carefully placed in a thimble, and the solvent in the flask was replaced with -180 mL of clean methanol. The LiB case was rinsed 3× with methanol (-5 mL per rinse) which was poured into the thimble, and batteries were then extracted for 24 h. After Extraction, the solvent was evaporated to -30 mL using a vacuum rotary evaporator and transferred to a 50 mL polypropylene tube. The extracts then were concentrated to -1 mL under nitrogen and brought the sample volume to 10 mL by adding clean methanol. For analysis 10 μL of each extract was transferred to an autosampler vial containing 270 μL methanol and 700 μL deionized water. Each sample was spiked with 25 ng L$^{-1}$ (20 μL of 1.25 μg L$^{-1}$ in 1000 μL) of IS prior to analysis.

## HPLC–QTOF–MS analysis at Texas Tech University

Methods are the same as those used by the authors in previous work[54,55]. The chromatographic separation was performed on a C18 analytical column (Gemini®, 3 μM, 100 × 3 mm ID, Phenomenex, CA, USA) coupled with a guard column (Gemini®, C18 4 × 2.0 mm ID, Phenomenex, CA, USA) with a SCIEX Exion LC high-pressure liquid chromatography (HPLC) pump. A delay column (Luna®, 5 μm, C18, 30 × 3 mm, Phenomenex, CA, USA) was installed between the mobile phase mixer and sample injector to separate background contamination that may come from solvent reservoir tubing and nonreplaceable

PTFE pump parts. The C18 and guard columns were maintained at 40 °C throughout the run. The aqueous phase consisted of 20 mM ammonium acetate solution (A), and the organic phase was 100% methanol (B). 500 μL of sample was injected during the analysis. The mobile phase flow rate was maintained at 600 μL min$^{-1}$ throughout the run, and the composition was ramped from 95% A to 35% A over the first min, and further ramped to 5% A at 8 min, 1% to in the next 0.1 min, held constant until 12.5 min, and at the end ramped to 95% A at 13.0 min and equilibrated the column for 3.5 min.

Analyses were performed on a quadrupole time of flight mass spectrometry (QTOF-MS) system (X500R, SCIEX, Framingham, MA, USA). Turbo ion spray was used as the ion source and maintained at 500 °C during the sample acquisition with the following conditions: ion spray voltage −4500 (v); curtain gas 30 (PSI); ion source gas 1 40 (PSI), ion source gas 2 60 (PSI). Collision-activated dissociation (CAD) gas was maintained at 10 PSI. Ultra-pure nitrogen was used for source, exhaust, and CAD gases. PFAS in Supplementary Table 1 were monitored and analyzed using an MRMHR acquisition method.

## HPLC–MS/MS analysis at Duke

At Duke University, an analogous quantitative method was implemented on a Thermo TSQ Altis HPLC-triple quadrupole mass spectrometer (MS/MS) with minor modifications, Specifically, aqueous and organic phase contained 2 mM ammonium acetate and 0.1% (V/V) acetic acid, sample injection volume was 50 μL and flow rate was maintained at 500 μL min$^{-1}$. The sheath gas, aux gas, sweep gas, ion transfer tube temperature, and vaporizer temperature were set to 50, 10, 1 arb, 325, and 300 °C, respectively. For monitoring PFEAs, the sheath gas was 30 arb, and the ion transfer tube temperature was 150 °C. This method monitored all PFAS in Supplementary Table 1 except 8Cl-PFOS, FDSA and n:x FTCA.

## HPLC–MS/MS analysis at NCSU

At North Carolina State University (NCSU), an analogous quantitative method was implemented on an Agilent Ultivo Triple Quadrupole LC/MS equipped with a 4.6 × 50 mm HPLC column (ZORBAX Eclipse Plus C18, 3.5μ, Agilent) using large volume (200 μL) injection. The column temperature was kept at 50 °C. An additional Agilent ZORBAX Eclipse Plus C18 column was connected before the injector to separate any background PFAS contamination. Mobile phases were 5 mM ammonium acetate in deionized water (solvent A) and a mixture of water: methanol 5:95 by volume (solvent B) at a flow rate of 0.7 mL min$^{-1}$. The composition was ramped from 95% A to 0% A at 18 min, held constant until 22 min, further ramped to 95% A at 22.1 min and equilibrated for 6 min. Each sample was injected twice at high (400 °C) and low (250 °C) ion source temperature settings to maximize responses for the targeted PFAS based on a previous study[56]. This method was used to analyze all PFECAs and PFESAs (Supplementary Table 1).

## Quality control

IS recoveries were used to evaluate extraction efficiency and matrix effects during analysis. Internal standard recoveries in unknown samples were calculated relative to the average of IS peak areas in calibration standards. If IS recoveries were outside the acceptable range of (50–150%), target analyte concentrations were flagged. Peaks of IS and calibrants (target analytes) in unknown samples were only considered for further analysis if retention times were ±30 s of calibration standards, signal-to-noise ratios were >10 and also at least 3× higher than the response in instrument blanks. Each analytical run consisted of 14 calibration standards (0.5–5000 ng L$^{-1}$), method blanks, instrument blanks, instrument sensitivity checks (ISCs, 0.5–10 ng L$^{-1}$), low concentration continuing calibration verification (CCV, 10 ng L$^{-1}$), and midpoint CCV (200 ng L$^{-1}$). All quality control samples except instrument blanks contained 200 ng L$^{-1}$ of each IS. The vial composition of all quality control samples was the same as unknown samples.

ISCs were performed by running 0.5–10 ng L$^{-1}$ standards immediately prior to unknown samples. The LOQ of an analyte was the lowest ISC where the calculated concentration was ±30% of the true concentration or the concentration detected in the method blank, whichever was higher. CCV was performed by injecting a standard after every 10 unknown samples (alternating between 10 and 200 ng L$^{-1}$) and sample data were accepted only if CCVs were ±30% of true value. Calibration curves were fit with regression equations ($R^2 > 0.99$) and used to quantify analytes in unknown samples. Every sample was quantified using an isotope dilution method, and concentrations of samples are reported as average triplicates. Relative standard deviation (RSD) of replicates was calculated and presented as a measure of variability during the analysis.

### *Daphnia magna* exposures

*Daphnia magna* (*D. magna*) neonates (<24-h old) were obtained from Aquatic Biosystems (Ft Collins, CO), and to bis-FMeSI following a modified 48 h acute, static non-renewal design described in EPA-821-R-02-012[57,58]. Exposures were conducted on individual *D. magna* to allow accurate video recordings of organism swimming behavior. Exposure media was EPA moderately hard comprised of 18-MΩ water (ELGA PURELAB® flex 5, Woodridge, IL 60517, USA) containing 96 mg L$^{-1}$ NaHCO$_3$ (sodium bicarbonate HPLC grade, 99%, Acros Organics, 446230010), 60 mg L$^{-1}$ CaSO$_4$ (calcium sulfate dihydrate analytical grade, 98%, Acros Organics 225275000), 60 mg L$^{-1}$ MgSO$_4$ (magnesium sulfate anhydrous 97%, Acros Organics 413485000), and 4 mg L$^{-1}$ KCl$_2$ (potassium chloride analytical grade, 99%, Acros Organics, 196770010). Measured dilution water quality parameters were measured at exposure setup are listed below in Supplementary Table 6. Exposure temperature, pH, conductivity, and dissolved oxygen were using a handheld HQ40d portable multimeter (Hach, CO, USA), while hardness and alkalinity were measured by titration (CHEMetrics, K-4520 Titrimetric Hardness and K-9810 Titrimetric Total Alkalinity). All experimental chambers were 100 ml polypropylene beakers pre-rinsed in triplicate with LC-MS-grade methanol. Ten *D. magna* were exposed individually to 0, 5, 10, 100, 1000 and 5000 ng L$^{-1}$ bis-FMeSI. Individual exposures allowed measurement of swimming track density, swimming distance, time, and velocity. Test chambers were randomly assigned locations within an environmental test chamber maintained at 20 ± 1 °C with a 16-h light 8-h dark cycle (Fisherbrand™ Isotemp™ BOD, retrofitted with timer-controlled lighting).

Concentrations of bis-FMeSI were evaluated using LC–QTOF–MS methods described above. Both acute and sublethal swimming endpoints were evaluated using the same *D. magna* exposures. Acute endpoints included lethality and immobilization while swimming track density, swimming distance, time, and velocity were measured by recording each *D. magna* for 10 min ring stand-based apparatus to ensure consistency. A UV-light was used to stimulate *D. magna* swimming behavior. And swimming behavior was recorded at 30 frames per second (fps). Raw mp4 data was processed using a frame-by-frame method with a post-image processing script from the Open CV2 Library for Python M[58,59]. Videos were converted to sets of still images where *D. magna* are assigned X and Y reference coordinates that can be tracked relative to an initial image. Movement maps were built with the image-processing Python application for each of the organisms to determine movement behaviors in relation to their positioning. Changes in coordinates vs. the time associated with each frame were used to evaluate location (i.e., swimming track density), distance, and velocity. Frames with movement vs. those where coordinates were unchanged were used to evaluate the percent time swimming. Because it captures location, swimming track density is an assessment of swimming performance that captures behavioral changes not observed in individual swimming metrics (e.g. distance, velocity)[59,60].

GraphPad PRISM (Version 9.5.1 for Mac, San Diego, USA) was used to conduct statistical analyses on *D. magna* swimming behavior endpoints. First, data were tested for normality using the Shapiro–Wilk test, which has high statistical power and can be used for small sample sizes ($n < 50$). Outliers within datasets were removed using the ROUT method ($Q = 5\%$)[61] using swimming distance as the main parameter, removing six datapoints on top of eliminating swimming data for the six dead or immobilized *D. magna*. Bartlett's test was used to evaluate homogeneity of variances in all remaining data. Correlations between concentrations and sublethal endpoints were determined by using the Pearson correlation test. One-way analysis of variance (ANOVA) with Tukey's multi-comparison test ($p \leq 0.05$) was performed on distance and %time swimming data Swimming velocity did not have equality of variance and therefore was evaluated using a $X^2$ test between all treatments[62] using Bonferroni correction to adjust significance level for multiple $X^2$ tests (30 comparisons total) resulting in an overall significance level of 0.002.

### Zebrafish exposures

Adult Ekkwill zebrafish (Ekkwill Waterlife Resources, Ruskin, FL) were maintained in a recirculating AHAB system (Pentair Aquatic Ecosystems, Apopka, FL) at 28 °C with a 14:10 light: dark cycle. Fish were fed *Artemia* nauplii in the mornings and Zeigler's Adult Zebrafish Complete Diet (Pentair Aquatic Ecosystems) in the afternoons. Breeder tanks, each containing 3 females and 2 males, were set up at 4 p.m. on the day before embryo exposures. Adults were spawned naturally the following morning within 2 h of the beginning of the light cycle. Embryos were transferred to Petri dishes (VWR International, West Chester, PA, USA) containing 30% Danieau's medium (17.4 mM NaCl, 0.21 mM KCl, 0.12 mM MgSO$_4$, 0.18 mM Ca(NO$_3$)$_2$, and 1.5 mM HEPES, pH 7.2) and placed in an incubator at 28 °C until exposure.

At 6 h post-fertilization (hpf; shield stage), embryos were screened for viability and development[63,64] using a dissecting stereomicroscope (Nikon SMZ1500, Nikon Instruments, Inc., Melville, NY). Normal embryos were selected and transferred to small glass Petri dishes each containing 10 embryos in 10 mL of treatment solution (i.e., one technical replicate). To make treatment solutions, bis-FMeSI was first dissolved in nanopure water to make a stock concentration of 0.48 mg mL$^{-1}$. This stock was diluted into test concentrations with 30% Danieau's medium to make the following concentrations (ng L$^{-1}$): 0 (control) 2.5, 25, 250, 2500, 25,000, or 250,000. Three additional concentrations—2,500,000, 25,000,000, 250,000,000 ng L$^{-1}$—were tested for acute endpoints of survival, hatching rate, and development at 144 hpf. There were three replicate dishes per treatment group. Larval locomotion assays had 9 replicate dishes across three biological replicates. Mitochondrial assays had three replicate dishes per experimental group ($n = 30$). Exposures were also conducted with field-collected water samples from the MN region (MN4, MN15, MN22). Field water samples were transferred to 50 mL centrifuge tubes and blinded as to their identity for toxicity testing. A vial blank was also tested, consisting of 30% Danieau's medium placed in the same type of centrifuge tube as the water samples for at least 1 h prior to use. Water samples were brought to 28 °C just prior to use. These samples were diluted to 50% with 30% Danieau's medium for exposures[64]. There was a final density of 1 embryo/mL/dish, and three replicate dishes per treatment group ($n = 30$) for each assay. Dishes were placed in an incubator at 28 °C with a 14:10 h light:dark cycle. Embryos were observed daily for survival and hatching. Mortality was determined by lack of heartbeat, and those individuals were removed from the dish. Larvae were assessed for developmental deformities at 6 days post-fertilization (dpf), before larval locomotion assays (see description of assay below).

A subset of thirty larvae per bis-FMeSI experimental group were also imaged in lateral orientation with a Nikon SMZ1500 microscope with a Nikon DXM1200 digital camera and NIS-Elements 3.10 software. Standard lengths (mm) of these larvae were measured with ImageJ

1.52a software[65,66]. Thirty larvae per group were placed in 10 L tanks and placed on the AHAB system and fed Ziegler's dry larval diet under the colony regimen for grow-out under clean conditions. At 6 weeks post-fertilization (wpf), these larvae were anesthetized with cold water, measured for standard length (mm), and returned to their tanks.

Embryos ($n = 14$/group) were assayed in vivo at 30 hpf for mitochondrial function using the Agilent Seahorse XFe96 Extracellular Flux Analyzer (Agilent Instruments, Santa Clara, CA) using established methods (10). Oxygen consumption (OCR; pmol min$^{-1}$) was measured during a time course in which drugs were injected in order to quantify various mitochondrial processes by modifying functions of the electron transport chain (ETC). Oligomycin A (9.4 $\mu$M; Sigma-Aldrich, St. Louis, MO) was injected following basal measurements to inhibit ATP synthase (complex V) and measure ATP-linked respiration. Proton leak was calculated based on the difference between ATP-linked respiration and basal respiration. Carbonyl cyanide-$p$- (trifluoromethoxy)phenylhydrazone (FCCP, 2.5 $\mu$M; Sigma-Aldrich) is an uncoupler that collapses the proton gradient and disrupts mitochondrial membrane potential. The resultant uninhibited flow of electrons through the ETC and oxygen consumption by complex IV pushes mitochondria to their maximum respiration rate. From this, maximum respiration and spare capacity (the difference between maximal and basal respiration), the ability to respond to an increased energy demand or stress, was calculated. Sodium azide (6.25 mM; Sigma-Aldrich) inhibits oxidative phosphorylation via inhibition of cytochrome c oxidase, rapidly depleting intracellular ATP and effectively shutting down mitochondrial respiration. The remaining non-mitochondrial respiration is attributed to enzymes and other factors within the cells that continue to consume oxygen.

Larval locomotion was measured at 6 days post-fertilization (dpf) using the DanioVision video-tracking system with Ethovision XT 13 software (Noldus, Leesburg, VA, USA). Larvae were individually transferred to wells of a clear 96-well plate (Greiner Bio-One, Monroe, NC) containing clean 30% Danieau's medium ($n > 60$/group for bis FMeSI and 30/group for water samples). Then larvae were acclimated in the plate at 28 °C for 1 h. At the end of this acclimation period, the plate was placed in the DanioVision observation chamber at 28 °C. A 50-min-long assay was then begun. The first 10 min of the assay was a habituation period in the dark to these conditions followed by two, alternating 10 min long light ("Light 1" and "Light 2") and dark ("Dark 1" and "Dark 2") periods. Larvae were assessed for mean distance traveled (mm) within each of the light and dark periods during the assay, not including the habituation period.

A series of statistical analyses were conducted on zebrafish data. For mitochondrial bioenergetics data, the interquartile method was used to identify and remove outlier points by cycle within each treatment group within each treatment group. Then individual parameters were calculated for each embryo. Basal respiration was the average of the three lowest oxygen consumption rate (OCR; pmol/O$_2$/min/embryo) values before the introduction of any drugs. Non-mitochondrial respiration was the average of the three lowest OCR values following the addition of sodium azide. Basal mitochondrial respiration was defined as basal respiration minus non-mitochondrial respiration. ATP production was calculated by subtracting the three lowest OCR values following the addition of Oligomycin A from basal respiration. Proton leak was calculated by first averaging the three lowest OCR values following the addition of Oligomycin A and then subtracting non-mitochondrial respiration. To find maximal mitochondrial respiration, the three highest OCR values following the addition of FCCP were averaged, and then the non-mitochondrial respiration was subtracted from this average. This average was also used to calculate spare capacity by subtracting basal respiration from it. For larval locomotion, the total distance traveled (mm) for each light and dark period, not including habituation, was averaged.

Statistics were done in GraphPad Prism 9.5.1 (GraphPad Software, Inc., La Jolla, CA) and JMP Pro 17 (SAS Institute, Cary, NC). All data were tested for normality with a Shapiro–Wilk test and homogeneity of variance with Bartlett's test. One-way ANOVA or Kruskal–Wallis tests with a post-hoc Tukey test ($\alpha = 0.05$) were used to compare treatment groups within each parameter for mitochondrial function as well as locomotion. All data are presented as mean ± SEM.

## Rapid small-scale column tests

The parameters of granular activated carbon (GAC) and ion exchange (IX) columns used in this study are shown in Supplementary Table 5. The rapid small-scale column test (RSSCT) column was assembled using polypropylene (PP) tubing with a 0.318 cm inside diameter and stainless steel valves and fittings from Swagelok (Solon, OH). A ~3 cm glass wool base was packed at the bottom of the column to provide support and stabilize the GAC/IX bed position during the run. Crushed GAC (0.042 g) and IX (0.011 g) were weighed in a beaker and then soaked in deionized water, followed by degassing in a vacuum desiccator for 24 h prior to packing. Degassed GAC/IX were loaded into the column using a glass Pasteur pipette. The packed GAC/IX column was equilibrated for 24 h by pumping deionized water through the column at the design flow rate using a Shimadzu LC pump (Model LC-20AT, Shimadzu Scientific, Columbia, MD). The influent solution was then switched to surface water or groundwater, which was dosed with 21 PFAS (Supplementary Figs. 17 and 18) at concentrations ranging from 80 to 100 ng L$^{-1}$. Influent samples were collected directly from the container at the beginning and end of the run. During the RSSCT run, effluent samples were collected periodically for PFAS analysis together with the influent samples. To capture the breakthrough of PFAS breaking through both early (e.g., PFMOAA and PFBA) and late (e.g., PFDA and PFOS), around 4 mL of samples were taken approximately every 250-bed volumes (BVs) for the first 1000 BVs treated, then every 500 BVs until 2000 BVs treated, then every 1000 BVs until 15,000 BVs treated, then every 5000 BVs until 50,000 BVs (GAC column run was completed) treated, and then every 20,000 BVs (IX only) until the run was complete.

## Alkaline heat-activated oxidation

Three bis-FASI homologs, bis-FMeSI, bis-FEtSI, and bis-FBSI, were subjected to an alkaline, heat-activated persulfate oxidation to screen fate during advanced oxidative treatment and during application of the total oxidizable precursor (TOP) assay. The TOP assay is a commonly applied PFAS analytical tool that evaluates the total molar concentration of oxidizable PFAS by measuring PFCA concentrations pre- and post-oxidation. The oxidation step converts perfluoroalkyl acid (PFAA) precursors to terminal PFCAs such that the change in PFCA concentration can be used to infer a total molar, oxidizable precursor concentration. The TOP assay facilitates the evaluation of concentrations of "unknown" PFAS in samples since many PFAS do not have the analytical standards needed to analyze them directly using targeted analysis. In this study, deionized water was spiked with 1000 ng L$^{-1}$ of each of the 3 bis-FASI homologs and subjected to the TOP assay following published techniques[66,67]. Spiked samples were amended with 100 mM potassium persulfate and 220 mM sodium hydroxide and placed in a water bath (Fisher Scientific, FSSWB15) for 12 h at 85 °C. Following the oxidation, samples were cooled down to room temperature, and pH was adjusted to 5−9 by HCl. Aqueous samples were then prepared for targeted analysis as described above.

## Reporting summary

Further information on research design is available in the Nature Portfolio Reporting Summary linked to this article.

## Data availability

Source data are provided with this paper in the Excel file entitled Guelfo et al. Supplementary Data S1–S32_v2 and a zip file containing swimming track density data for *D. magna* organized by exposure dose.

## Code availability

Raw mp4 data for *Daphnia Magna* was processed using a frame-by-frame method with a post-image processing script from the Open CV2 Library for Python M. A detailed citation can be found in ref. 59.

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

## Acknowledgements

We thank Michael Halverson for providing background knowledge related to the Minnesota study area. We also thank Dr. Juliane Hollender and Heinz Singer from the Swiss Federal Institute of Aquatic Science and Technology (EAWAG) in Dübendorf, Switzerland for providing laboratory space and resources used in processing field samples collected in Europe. We acknowledge funding support from the Ed and Linda Whitacre Faculty Fellowship at Texas Tech University, the Duke University Superfund Research Center (National Institute of Environmental Health Sciences award number 5P42ES010356-21), and the North Carolina PFAS Testing Network, which provided funding for efforts at Duke University and North Carolina State University.

## Author contributions

Conceptualization: J.L.G., P.L.F., E.P.G., N.J., D.R.U.K., P.M.; Formal analysis (field data): J.L.G., P.L.F.; Formal analysis (*D. magna*): A.D.M., E.P.G.; Formal analysis (zebrafish): M.C., N.J.; Formal analysis (GAC/IX treatment): P.M., D.R.U.K.; Formal analysis (lifecycle and oxidation): P.L.F., M.S.; Investigation (field data): J.L.G., P.L.F., J.B., T.F. P.W.F., M.S., A.S.J.; Investigation (*D. magna*): A.D.M.; Investigation (zebrafish): M.C.; Investigation (GAC/IX treatment): P.M.; Investigation (lifecycle and oxidation): M.S.; Project administration: J.L.G.; Resources: J.B., T.F.; Supervision: J.L.G., P.L.F., E.P.G., N.J., D.R.U.K.; Writing—original draft: J.L.G.; Writing—original draft (*D. magna*): A.D.M., E.P.G.; Writing—original draft (zebrafish): M.C., N.J.; Writing—original draft (lifecycle): P.L.F., M.S.; Writing—original draft (treatment columns): P.M.; Writing—original draft (oxidation): M.S.; Writing—review & editing: J.L.G., P.L.F., E.P.G., N.J., D.R.U.K.

## Competing interests

The authors declare no competing interests.
