## [Peer Review File · Nature Communications]

REVIEWER COMMENTS

Reviewer #1 (Remarks to the Author):

Summary:

The manuscript by Guelfo et al. presents information on a class of per- and polyfluoroalkyl substances (PFASs) known as bis-perfluoroalkyl sulfonamides (bis-FASIs). They present monitoring data, conduct ecotoxicity testing and test their removal efficiency in water treatment processes. The authors conclude that bis-FASIs are “comparable to PFAS that are now prohibited and highly regulated worldwide” (lines 29-30). The analytical data are of good quality and the testing also appears to have been performed with rigor.

My main criticism of the manuscript is that the authors oversell the risks posed by bis-FASIs based on limited data. This overselling of the significance of the work is a shame because it undermines their otherwise good work. The PFAS which was prohibited and highly regulated worldwide are those PFAS listed on the Stockholm Convention and are: 1) perfluorooctane sulfonic acid (PFOS) its salts and perfluorooctane sulfonyl fluoride (PFOSF), 2) perfluorooctanoic acid (PFOA) its salts and PFOA-related compounds, and 3) perfluorohexane sulfonic acid (PFHxS) its salts and PFHxS-related compounds. PFOS, PFOA and PFHxS are known to be bioaccumulative in humans and also to cause a range of (eco)toxicological effects to humans and wildlife. Guelfo et al. have demonstrated the presence of bis-FASIs in Minnesota at parts per trillion to low parts per million levels (particularly near manufacturing point sources) and shown that they cause ecotoxicological effects in non-standardized toxicity tests. They have not demonstrated bioaccumulation potential and cannot claim that the level of concern is the same as for PFOS, PFOA and PFHxS. In Europe, regulators are using a new paradigm for regulation based on if a substance is persistent, mobile and toxic (PMT) or very persistent (vP) and very mobile (vM). Bis-FASIs seem to meet the PMT/vPvM criteria (M indicated by low log K_d value) which would make them candidates for listing as Substances of Very High Concern in the European Union. I would have preferred arguments along these lines.

A few other major concerns are listed below:

This was not the first report of bis-FASIs in the environment as the authors admit, and this further reduces the significance of the work. Although monitoring of bis-FASIs is needed, monitoring is not usually considered sufficient for publication in high-impact journals. Also, the sampling was biased toward areas close to 3M manufacturing sites so it's difficult to get a representative picture of the national/global environmental contamination with bis-FASIs. The addition of the ecotoxicity tests and

water treatment tests improves novelty. However, treatability is not a criterion used in the chemical assessment, and other more typical assessment criteria (e.g. bioaccumulation potential) are missing.

It is assumed that bis-FASIs are solely derived from lithium ion batteries (LIBs), but in the recent review by Rensmo et al. (ref. 21 in the manuscript) it is suggested that bis(trifluoromethylsulfonyl)imide and similar ionic liquids have mostly been investigated for their use as corrosion inhibitors. A more thorough review of the possible sources and uses of bis-FASIs is warranted.

Based on extensive discussions with electrolyte manufacturers, it is my understanding that most LIB electrolytes are based on LiPF₆, organic cyclic and linear carbonates, and various additives. The same is true for Na-ion batteries. I have been informed that it only makes sense, for very specific applications such as Li//S-cells, to use lithium bis(trifluoromethanesulfonyl)imide (LiTFSI) and PFAS-based ethers, such as 1,1,2,2-tetrafluoroethyl 2,2,3,3-tetrafluoropropyl ether. In Bolloré cells, LiTFSI/PEO was used in the past, whereas Li-Triflate (CAS 33454-82-9), LiBETI (CAS 132843-44-8), LiFAP (LiPF₃(CF₂CF₃)_{3n}), Tris(2,2,2-trifluoroethyl)borate (TFEB CAS 659-18-7) and trifluorotoluene (TFT CAS No. 98-08-8) have no current commercial relevance. In summary, according to electrolyte manufacturers fluorinated ionic liquids are only used in very limited applications in LIBs, e.g. as additives. The manuscript may exaggerate the current situation, based on the discussions that I have had. A more thorough review and some discussion with electrolyte manufacturers is needed to back up the strong claims made in the manuscript. LIB manufacturing is an important industry for the green energy transition. Scientific accuracy is therefore important so as to properly weigh the risks and benefits of LIBs.

I am not an ecotoxicologist but, as I understand, swimming behavior tests are not standardized tests, although they appear to be gaining acceptance in the scientific community. I'd strongly recommend that the editor reaches out to an expert ecotoxicologist to review this section of the manuscript. The authors are experts on PFAS treatment and analytical chemistry and I feel confident that this part of the manuscript is well done.

Some more minor points:

Line 31: No temporal trend data are included so how can the authors conclude that the clean energy sector is a growing source of PFAS release? It seems likely that this is true, but the study does not confirm an increasing trend.

Lines 278-279: It is not clear how representative these LIBs are of products on the market? The authors should have made efforts to discuss this with the LIB industry. As I mentioned above, my discussions have indicated that bis-FASIs are not widely used in LIBs. Is the LIB industry being deliberately misleading?

Lines 353: I don't see how global release is proven by this study? The monitoring was regional and biased to an area near to manufacturing. These statements are an overextrapolation.

Reviewer #2 (Remarks to the Author):

This paper by Guelfo et al. examines the effects of bis-perfluoroalkyl sulfonimides (bis-FASIs), a novel class of PFAS found in lithium-ion batteries, to *Daphnia magna* and zebrafish. This paper is a comprehensive evaluation of bis-FASIs and may be the first paper to examine the environmental impacts of this class of PFAS compounds from manufacturing until disposal. Their environmental sampling efforts are particularly notable and useful to risk assessors, since they include data from many countries (USA, Antwerp, France) and matrixes (surface water, soil, sediment etc.). The experimental design is also strong, since the paper includes a wide range of samples and concentrations (5 concentrations and a control) that span levels found in the environment. The behavioral and mitochondrial tests conducted in the animals are standard. However, this reviewer was unable to find any data provided on growth and survival, which are critical toxicological endpoints, and this information should be included within the publication. Furthermore, it is unclear to the reader why behavior and mitochondrial function were selected to evaluate toxicity of bis-FASIs and no other standard endpoints like growth and survival. There is no hypothesis presented in the introduction for why these endpoints were selected and there is no link to why behavior and/or mitochondrial respiration are expected to be impacted. It is also unclear how mitochondrial function at 48 hpf relates to behavior measured at 144 hpf. Overall, the field data and environmental sampling are the major strength of this paper, while the toxicological endpoints seem to offer very little evidence of dose response effect and need clarification.

Direct comments:

Line 29-30: list the PFAS that bis-FASIs are comparable to.

Lines 31-33: please include the median concentration. Giving just the max with no other context is not helpful to the reader.

Lines 47-84: This main section does a great job of explaining how bis-FASIs end up in the environment, as well as the rationale for the different areas sampled. What is missing from this section is why the only way aquatic toxicity was assessed was through behavioral assays in zebrafish and daphnia and a mitochondrial function assay in zebrafish. What was the hypothesis of the test? Why were normal parameters of toxicity, such as growth reproduction or survival, not considered or measured?

Line 207: Did exposure to bis-FASIs affect the growth or survival of daphnia?

Lines 222-225: There was no change in average velocity, just a change in the variance of velocity. Is this how one would define a neuroactive effect?

Lines 237: The distance/velocity figures here with just mean +/- SD are not very useful in their current format for publication. A box plot, a violin plot, and/or a plot with the individual replicates displayed, would do a better job visually displaying the change in variance the authors are attributing to bis-FASI exposure.

Lines 225-227: The lack of dose response, or effect to changes in the distance travelled, swimming duration or velocity, indicates to me a lack of toxicological effect.

Lines 253-255: What do decreases in these parameters mean for overall mitochondrial function? According to Chacko et al. 2014 Clinical Science (doi: 10.1042/CS20140101), damaged/stressed mitochondria tend to have increased basal respiration and proton leak, along with reduced reserve capacity and ATP linked respiration. In this paper, the results seem to be decreased across the board which makes it challenging to interpret what they mean for overall mitochondrial health.

256-259: Are these behavioral results toxicity or just random variance? Are non-monotonic responses in LPR typical for zebrafish exposed to PFAS? The effect at 25 ng/L looks within the realm of control fish since the controls on the field panel looks to be higher than the fish exposed to 25 ng/L.

Demonstrating that this effect is reproducible, along with testing a few concentrations below this "threshold" would convince readers that there is something going on here.

267-272: Why is the zebrafish behavior displayed like this? The materials and methods listed this assay as a 50-minute LPR. It is unclear what this data is showing. Is this light and dark cycle data averaged together? If so, why was this done? Did exposure to these compounds affect behavior in light or dark phases, did it affect how the fish transition between phases? Overall, this is an unusual way of evaluating the LPR assay and I would have to ask, why perform light/dark transitions if averaging it all together?

357-360: there is insufficient evidence in this study to make such strong conclusions. Based on the evidence in this study there seems to be no dose-dependent effects on behavior and potential mitochondrial dysfunction at 250,000 ng/L.

Methods/Results: a summary table of measured bis-FMeSI for the daphnia and zebrafish exposures should be presented somewhere to allow better interpretation of the nominal values presented in this study.

Reviewer #3 (Remarks to the Author):

The paper present unexpected and worrying data showing the release of PFAS into the environment near lithium-ion battery related production facilities.

The work is original and extremely important with respect to the "green credentials" of lithium-ion batteries and the race to nett zero.

The paper is entirely self-contained and the data presented does support the conclusions drawn.

I can see no flaws in the analyses-but I must confess to not being an expert in analysis

As far as I can see, the work is methodical and precise and good science.

There is very great experimental detail provided in the paper which would allow the tests to be reproduced.

I recommend publication.

RESPONSE TO COMMENTS

The dirty side of clean energy: Lithium ion batteries as a source of PFAS in the environment

We have provided responses to the reviewer feedback below in a point-by-point format. Text shown in **blue** highlight edits that were made to the manuscript. We have provided copies of all edits in our point-by-point responses, but references within these edits have been removed. References cited in this document are only those which are cited within responses and not the associated manuscript. Line numbers refer to the revised manuscript with changes tracked unless otherwise noted.

Reviewer #1 (Remarks to the Author):

1. **Summary: The manuscript by Guelfo et al. presents information on a class of per- and polyfluoroalkyl substances (PFASs) known as bis-perfluoroalkyl sulfonamides (bis-FASIs). They present monitoring data, conduct ecotoxicity testing and test their removal efficiency in water treatment processes. The authors conclude that bis-FASIs are “comparable to PFAS that are now prohibited and highly regulated worldwide” (lines 29-30). The analytical data are of good quality and the testing also appears to have been performed with rigor.**

Thank you for the summary of our manuscript and favorable view of our data quality. We would like to emphasize that our abstract does not, “conclude that bis-FASIs are comparable to PFAS that are now prohibited...” Rather we specifically state (**lines 28-30**) that the, “...**occurrence, ecotoxicity, and treatability** of this novel class of PFAS are comparable to PFAS that are now prohibited and highly regulated.” We view this as a fundamental distinction to point out since this manuscript contains original data on occurrence, ecotoxicity, and treatability of bis-perfluoroalkyl sulfonimides (bis-FASIs). Subsequent comments by Reviewer 1 (e.g., **Comments 2-4**), indicate that we have stated bis-FASIs pose the same “level of concern” as perfluorooctane sulfonic acid (PFOS), perfluorohexane sulfonic acid (PFHxS), and perfluorooctanoic acid (PFOA). In fact, we have only drawn comparisons between bis-FASIs and other per- and polyfluoroalkyl substances (PFAS) in areas where we have original data to compare directly to that of prior studies (i.e., occurrence, ecotoxicity, and treatability). To be clear, we have focused our comparisons on those supported directly by our own original data and results reported in the literature. We did not make generalized conclusions that bis-FASIs are equivalent to other PFAS (i.e., that they are the same “level of concern”), and feel it is important to clarify this for the reviewers and the editor. More details regarding our comparisons of occurrence, ecotoxicity, and treatability of bis-FASIs to other PFAS are provided in our response to **Comment 4**.

2. **My main criticism of the manuscript is that the authors oversell the risks posed by bis-FASIs based on limited data. This overselling of the significance of the work is a shame because it undermines their otherwise good work.**

We disagree with this assessment. It is crucial to note that **we do not use the word “risk”** within our manuscript. Risk assessment was not an objective of this study, so this feedback misrepresents our work. Notably, Reviewer 1’s view that we have overstated risks associated

with bis-FASIs is not shared by Reviewers 2 or 3, who are also experts in their fields. More details are provided in our responses to **Comments 4**, and **6-10**.

- 3. The PFAS which was prohibited and highly regulated worldwide are those PFAS listed on the Stockholm Convention and are: 1) perfluorooctane sulfonic acid (PFOS) its salts and perfluorooctane sulfonyl fluoride (PFOSF), 2) perfluorooctanoic acid (PFOA) its salts and PFOA-related compounds, and 3) perfluorohexane sulfonic acid (PFHxS) its salts and PFHxS-related compounds.**

Thank you for the feedback. We agree with the reviewer's assessment that PFOS, PFOA, PFHxS, and compounds related to each of these are listed in the Stockholm Convention. We also note that long-chain perfluoroalkyl carboxylic acid (PFCAs; e.g., those with 8-20 carbons) are candidate persistent organic pollutants (POPs) under the Stockholm Convention¹. However, we disagree that the Stockholm Convention should be the only metric of gauging worldwide regulation of PFAS. Prominent examples of regulating additional PFAS include regulations in the United States (US) and Europe.

The US has signed, but not ratified, the Stockholm Convention. However, the US Environmental Protection Agency (USEPA) has proposed federal drinking water maximum contaminant levels (MCLs) for PFAS on the Stockholm Convention list as well as 2 additional PFAS. Specifically, the proposed MCLs are 4 ng/L for PFOS and PFOA^{2,3}, and a hazard index (HI)-based MCL for perfluorononanoic acid (PFNA), perfluorobutane sulfonate (PFBS), PFHxS, and hexafluoropropylene oxide dimer acid (HFPO-DA, trade name GenX)^{4,5}. Additionally, the German, Swedish, Norwegian, and Danish governments have collectively proposed to regulate PFAS as a class⁶, where the Organization for Economic Cooperation and Development (OECD) definition of PFAS (which includes bis-FASIs) is used to bracket the scope of this regulation⁷. These proposed regulations demonstrate that the scope of international focus on PFAS extends far beyond the individual compounds listed in the Stockholm Convention.

- 4. PFOS, PFOA and PFHxS are known to be bioaccumulative in humans and also to cause a range of (eco)toxicological effects to humans and wildlife. Guelfo et al. have demonstrated the presence of bis-FASIs in Minnesota at parts per trillion to low parts per million levels (particularly near manufacturing point sources) and shown that they cause ecotoxicological effects in non-standardized toxicity tests. They have not demonstrated bioaccumulation potential and cannot claim that the level of concern is the same as for PFOS, PFOA and PFHxS.**

Thank you for the feedback. The reviewer here casts our toxicity tests as “non-standardized.” We address this directly in our response to **Comment 11**. Bioaccumulation is further addressed in **Comment 8**, but measurement of bis-FASI bioaccumulation was beyond the scope of the current study and will be addressed in future work

The reviewer's feedback states that we have “claimed that the level of concern...” for bis-FASIs, “is the same as for PFOS, PFOA, and PFHxS.” However, our manuscript **does not** contain statements that are this generalized. References to PFOS, PFOA, and/or PFHxS are points of comparison of specific results of our work to results of prior studies. These include:

- Occurrence: A comparison of the measured aquatic concentrations of bis-FASIs near manufacturing facilities to those of PFOS and PFOA (e.g., **lines 122-124**),
- Ecotoxicity: A comparison of zebrafish ecotoxicity of the bis-FASI homologue bis(trifluoromethylsulfonyl)imide (bis-FMeSI) evaluated in our study to the effects of PFOS in similar studies (e.g., **lines 277-278**); and
- Treatability: A comparison of bis-FMeSI treatability in granular activated carbon and ion exchange (both evaluated in our study) to removal of previously studied PFAS such as PFOS and PFOA (e.g., **Fig. 4** of the manuscript).

Since it is necessary to compare study results to those of prior work, we have not made any edits related to these statements. Additionally, in **Comment 5** the reviewer concurs with our assessment of mobility and persistence.

The following, general statements were also included in the manuscript:

- A general statement in the summary paragraph (**lines 28-30**) that mentions occurrence, ecotoxicity, and treatability (see response to **Comment 1**).
- Our conclusions state that, “In general, the challenges associated with bis-FASI occurrence, mobility, toxicity, and recalcitrance are similar to those that have been realized for other PFAS.”

No edits were made to the summary paragraph statement since again, it is related to original data generated in our study. To ensure we have limited our comparisons of bis-FASIs vs. other PFAS to areas that are specific to our study, we have made a minor clarification in the conclusions. **Line 443** of the manuscript now states that:

“In general, the challenges associated with bis-FASI occurrence, mobility, ecotoxicity, and recalcitrance...”

- 5. In Europe, regulators are using a new paradigm for regulation based on if a substance is persistent, mobile and toxic (PMT) or very persistent (vP) and very mobile (vM). Bis-FAs seem to meet the PMT/vPvM criteria (M indicated by low log Kd value) which would them candidates for listing as Substances of Very High Concern in the European Union. I would have preferred arguments along these lines.**

Thank you for the suggestion. We assume the reviewer is referring to the European Commission’s introduction of a very persistent very mobile (vPvM) hazard class⁸. We agree that bis-FMeSI meets criteria for designation as vPvM, and our revisions are discussed below. We also note that we find this comment combined with feedback in **Comment 4** and elsewhere to be contradictory. Reviewer 1 disagrees with the concept that bis-FASIs pose the same “level of concern” as other PFAS (e.g., Comment 4). Here the reviewer seems to argue the opposite. European Commission reports and peer reviewed studies argue that persistent and mobile compounds represent an equivalent level of concern as persistent, bioaccumulative, and toxic (PBT) compounds^{9,10}, and this argument has been used to designate PFBS (a PFAS with persistence and mobility similar to bis-FMeSI) as a substance of very high concern. To be clear, this means that vPvM PFAS are considered to carry the same “level of concern” as PFAS, such as PFOA that are considered PBT. **So although we did not make generalized statements that bis-FASIs represent the same level of concern as other PFAS, this comment suggests that we would have been justified in doing so.**

Nevertheless, we agree that bis-FMeSI meets criteria for designation as vPvM, and have edited the manuscript as described below.

Lines 409-415 state that:

“High persistence and high mobility (i.e., weak sorption; **Fig. S9**) of bis-FMeSI may meet criteria for classification as a very persistent very mobile (vPvM) compound under recently proposed European Commission hazard classifications. Bis-FMeSI has similar persistence and mobility as PFBS, which is a vPvM PFAS recently designated under European guidelines as a substance with an equal level of concern as persistent, bioaccumulative, and toxic substances. More details are in the supplementary text.”

The following has been added to the supplementary text (see **Treatment: Oxidation**):

“The European Commission’s (EC’s) recently proposed addition of a very persistent very mobile (vPvM) hazard class to multiple regulations. Under the vPvM paradigm, biodegradation $\frac{1}{2}$ -life is used to evaluate persistence and mobility is evaluated using the organic carbon-water partitioning coefficient (K_{oc}). Specifically, compounds with $\log K_{oc} < 2.0$ and biodegradation $\frac{1}{2}$ -life > 180 days meet criteria for designation as vPvM. Field K_d values for bis-FMeSI were similar to PFPeA and PFBS (**Fig. S9**), both of which have $\log K_{oc}$ values < 2 , suggesting that bis-FMeSI is very mobile. As observed for PFOA and PFOS in prior studies, bis-FMeSI was recalcitrant under alkaline, oxidative conditions (**Fig. S19**). Although confirmation is needed, these results suggest that bis-FASIs also will not undergo degradation in the environment, which is also true of PFOA and PFOS. Persistence and mobility data presented here suggest that bis-FMeSI meets criteria for classification as vPvM under the new EC hazard classification.

The vPvM classification recognizes the potential for compounds to impact aqueous systems over large (e.g., global) regions for extended periods of time, even if production volumes are low relative to other contaminants. In fact, the European Commission has recognized that the concerns about contaminant mobility should be considered equivalent to concerns about bioaccumulation. PFBS is designated as a vPvM compound, and in 2019 it was identified as a substance of very high concern (SVHC) based on its equivalent level of concern (ELoC) to persistent, bioaccumulative, and toxic (PBT) substances. The decision to designate PFBS as a SVHC was based on characteristics including difficulty in removing PFBS during drinking water treatment, continuous exposure as a result of persistence and mobility, accumulation in organisms at an equilibrium level as a result of continuous exposure, high global transport potential based on low volatility, high solubility, and low sorption to soils and sediments. A recent study used systematic case studies to demonstrate that vPvM substances across contaminant classes pose ELoC to PBT substances under REACH. Based on their outcomes, it is reasonable to hypothesize that bis-FMeSI, which has persistence and mobility similar to PFBS, also poses ELoC as PBT substances. Importantly this does not supersede the value in additional assessments of bis-FASI bioaccumulation and toxicity.”

6. A few other major concerns are listed below:

This was not the first report of bis-FASIs in the environment as the authors admit, and this further reduces the significance of the work. Although monitoring of bis-FASIs is

needed monitoring is not usually considered sufficient for publication in high-impact journals.

Thank you for the feedback. The reviewer states that monitoring is not considered sufficient for publication in high impact journals, and we agree. However, many studies published in high impact journals included monitoring as part of a multi-faced study, as presented here. To be clear, we do not rely exclusively on our monitoring results to establish the novelty or importance of our work. This is the first study to consider bis-FASI occurrence in consumer products and disposal scenarios, ecotoxicity, or treatability. Additionally, we argue that the very limited prior environmental measurements of bis-FASIs in environmental samples (i.e., detections of ≤ 2 g/L bis-FMeSI in 12 aqueous samples across two studies) do not diminish the novelty of our monitoring data. Our monitoring data are international, and unlike the previous work, we include multiple bis-FASI homologues and data for soil and sediment. The novelty and need for our work is also supported by statements made in these prior studies about the urgent need for additional investigations of bis-FASIs. Lastly, the importance and quality of the monitoring work are supported by Reviewer 2 (see **Comments 15, 17**).

The reviewer's feedback suggests that further description of the prior study results vs. the contributions of the monitoring data in our study is warranted. **Lines 77-78** of the manuscript now state:

“More information is available in the supplementary text.”

The following was added to the supplementary text (see **Background**):

“There are limited prior reports of bis-FMeSI in surface water and riverbank filtration samples. Neuwald et al. (2022) detected bis-FMeSI in 9 samples at concentrations of \$\leq 2\$ ng/L (median 0.8 ng/L) in surface water and riverbank filtration samples, and Wang et al (2023) detected bis-FMeSI in 3 sea water samples at concentrations of 0.296-1.5 ng/L. There are no reports of bis-FMeSI in soil or sediment, and no reports of bis-FMeSI homologues such as bisperfluoroethanesulfonimide (bis-FEtSI) and bisperfluorobutanesulfonimide (bis-FBSI) in any environmental media. These prior studies, which referred to bis-FMeSI as NTF2, note the urgent need for additional bis-FASI investigation. Wang et al. (2023) noted that, “...further field data on the environmental behavior of NTF2 is urgently needed.” In reference to bis-FMeSI, Neuwald et al. (2022) stated, “Currently, the lack of occurrence data makes it impossible to evaluate if its use in energy storage leads to its environmental release.” Conclusions by these authors further highlight the novelty and immediacy of our study objectives and results. Importantly, we will demonstrate detections of bis-FMeSI three orders of magnitude higher than those of previous studies, the first bis-FMeSI detections in soil and sediment, and the first detections of bis-FBSI or bis-FEtSI in any environmental media.”

- 7. Also, the sampling was biased toward areas close to 3M manufacturing sites so it's difficult to get a representative picture of the national/global environmental contamination with bis-FASIs.**

Thank you for the feedback. We disagree with the reviewer's description of the sampling as “biased” since a primary objective of this study was to evaluate release near sites of PFAS manufacturing, as stated in **Lines 80-81**. Use of the word bias implies that our study design prevented our ability to adequately address one of our study objectives, but our objectives did

not include completion of a “national/global” inventory of bis-FASIs in the environment. In order to evaluate release near sites of PFAS manufacturing, it was necessary to sample near sites of PFAS manufacturing. We evaluated PFAS release from **multiple manufacturers** in order to meet this study objective. Specifically, we sampled near 3M (2 locations), Solvay (1 location), and Arkema (2 locations) facilities. Thus, the assessment that sampling was biased towards only 3M facilities, fails to recognize a significant portion of the study scope and led us to wonder whether the study design and results were carefully considered by the reviewer.

This feedback also conflicts with comments the reviewer provided in **Comment 5**, where they suggested that bis-FMeSI may meet criteria for classification as vPvM. According to published studies, the new vPvM classification was developed partly in recognition that relatively small releases of vPvM compounds may cause global impacts to water resources that persist long after release has stopped⁸. Reviewer 1 suggested we make a case that bis-FMeSI may meet criteria for designation as vPvM (i.e., high potential for global distribution), and we agreed. However, here the reviewer argues inconsistently that bis-FASIs may not be a global issue.

The reviewer’s feedback suggested it would be beneficial to clarify our objective of monitoring near manufacturing facilities. The supplementary text (see **Background**) now states that:

“PFAS releases have occurred from primary manufacturers of PFAS (e.g., Chemours, Parkersburg, WV) and from sites where PFAS are used during the manufacturing process (e.g., ChemFab performance plastics in Bennington, VT). For example, atmospheric emissions from secondary manufacturer ChemFab leached into groundwater after surface deposition and caused perfluorooctanoic acid (PFOA) concentrations up to 600 ng/L in public supply and residential wells over an area >200 km². By the time this release was discovered in 2015, use of PFOA was phased out, but it had been replaced with other PFAS including perfluoro-2-propoxypropanoic acid (HFPO-DA; trade name GenX). Information about replacement PFAS is considered confidential business information in the US, so releases of compounds such as HFPO-DA were initially discovered through monitoring efforts in the Cape Fear River near a Chemours primary manufacturing facility. This led to subsequent discovery of concentrations of HFPO-DA up to 631 ng/L in raw water at a drinking water treatment plant in the same watershed. Collectively, these studies demonstrate potential for widespread environmental impacts of PFAS as a result of both primary and secondary manufacturers, and the value of independent monitoring of manufacturing sites to elucidate undisclosed releases of PFAS. Primary manufacturers have a documented history of failing to disclose information on the human health impacts of PFAS, which adds a layer of importance to independent studies of PFAS occurrence as well as toxicity and treatability.”

To ensure that the implications of our work are clear, we have edited the manuscript conclusions (**Lines 423-427**) to state that:

“This study demonstrates for the first time the ~~global~~ **international** and heretofore unrecognized release of novel, LiB-associated PFAS (bis-FASIs, particularly bis-FMeSI) to soil, sediment, and surface water, and that concentrations of these compounds in the parts per billion are common, near manufacturing areas. **When coupled with low-level detections in three Chinese sea water samples and characteristics consistent with vPvM classification, this suggests bis-FASI release is global.**”

8. The addition of the ecotoxicity tests and water treatment tests improves novelty. However, treatability is not a criterion used in the chemical assessment, and other more typical assessment criteria (e.g. bioaccumulation potential) are missing.

Thank you for the feedback. Use of the term “missing” in this feedback implies that data needed to support our objectives were not included in the study. However, our study objectives did not include an assessment of bioaccumulation. Additionally, this feedback is contradictory to **Comment 5** provided by the reviewer, which suggests that assessments can be made based solely on persistence and mobility, factors that were evaluated in our study. As noted in our earlier response, vPvM compounds have been found to have the same level of concern as PBT compounds, and have been designated as substances of very high concern, equivalent to PBT compounds. This suggests that persistence and mobility can be used in the absence of bioaccumulation as “typical assessment criteria” for completing “chemical assessment.”

We would also like to address the reviewer’s feedback that “treatability is not a criterion used in the chemical assessment.” As noted in Comment 5, PFBS has been designated as a substance of very high concern based on its vPvM classification. Specific characteristics that were considered were a high potential for exposure because PFBS can pass through drinking water treatment systems as a result of its persistence and mobility⁹. So the vPvM classification suggested by the reviewer, appears to be an approach to chemical assessment that considers treatability.

However, most critical to this response, is that we are unaware of a requirement to always use a uniform approach for assessing chemicals and would argue that constant use of such approaches without consideration of the data that would be most informative, runs the risk of hindering scientific creativity and novelty. In this study, an objective was to evaluate treatability, so we generated the data needed to address this objective. To provide improved context regarding the inclusion of treatability studies, the following was added to the supplementary text (see **Background**):

“Collectively, studies of PFAS occurrence and toxicity indicate a need for effective PFAS treatment approaches, but most conventional treatment approaches for media such as drinking water are ineffective for complete removal of PFAS. Highly recalcitrant PFAS such as PFOA, perfluorooctanesulfonic acid (PFOS), and their homologues are not readily mineralized during oxidation. Under some conditions, concentrations of perfluoroalkyl carboxylates (PFCAs; i.e., PFOA and homologues) may increase during oxidation because they are terminal daughter products of oxidizable PFAS. Even PFAS that can be degraded during oxidation are recalcitrant because their terminal daughter products are still PFAS. Since oxidative approaches are routinely employed for disinfection during treatment, it is important to understand how PFAS may behave in these systems. Although removal is unlikely, parent PFAS may have different properties than their terminal oxidative transformation products, which may impact the approach used to remove those products during subsequent treatment steps (e.g., adsorption). Researchers are investigating destructive techniques for PFAS, but adsorption-based removal using with granular activated carbon (GAC) and/or ion exchange (IX) resin is more readily implemented in full-scale systems. Collectively, screening novel PFAS for fate during oxidation and adsorption-based treatment will inform their recalcitrance and treatability relative to PFAS for which there are already regulatory drivers for removal (e.g., PFOA, PFOS).”

As noted in previous work¹¹, screening treatability of PFAS also informs potential for exposure. If exposure potential is low, then additional studies of bioaccumulation and toxicity may not need immediate prioritization in future studies. If treatability studies highlight treatment challenges, exposure potential will be higher (as noted for vPvM compounds), and this highlights a more pressing need for studies of bioaccumulation. To better emphasize this point, we have added the following statement to the conclusions (**Lines 436-443**):

“Where these treatment approaches are not in place or are not engineered for bis-FASI removal, potential for exposure may be elevated. This underscores a need for additional studies of bioaccumulation and human health impacts.”

- 9. It is assumed that bis-FASIs are solely derived from lithium ion batteries (LIBs), but in the recent review by Rensmo et al. (ref. 21 in the manuscript) it is suggested that bis(trifluoromethylsulfonyl)imide and similar ionic liquids have mostly been investigated for their use as corrosion inhibitors. A more thorough review of the possible sources and uses of bis-FASIs is warranted.**

Thank you for the feedback. We agree that Rensmo et al. (2021) state that, “Bis(trifluoromethylsulfonyl)imide and similar ILs have mostly been investigated for their use as corrosion inhibitors¹²,” but we disagree with their conclusion. They provide a single citation to support their statement, which we suggest that the reviewer consider more carefully. The referenced study by Assenine et al. (2021) investigates use of bis-FMeSI as a corrosion inhibitor; however, the study does not provide any evidence that bis-FMeSI is used primarily as a corrosion inhibitor vs. other applications (e.g., electrolytes). We conducted a search on Web of Science for references containing the terms “LiTFSI” (a trade name for bis-FMeSI) and “corrosion inhibitor.” This search resulted in a total of 5 publications between 2014 (none earlier) and 2023 that were cited 116 times. In contrast, the search for “LiTFSI” and “electrolyte” identified 3190 publications between 1992 and 2023 that were cited a total of 153,693 times. Thus it appears that bis-FMeSI has been far more extensively examined as an electrolyte than as a corrosion inhibitor. Because of the scarcity of information related to use of bis-FASIs as corrosion inhibitors, we have declined to make edits in response to this feedback. Additional information on the use of bis-FASIs in LiBs is provided in our response to **Comment 10**.

- 10. Based on extensive discussions with electrolyte manufacturers, it is my understanding that most LIB electrolytes are based on LiPF₆, organic cyclic and linear carbonates, and various additives. The same is true for Na-ion batteries. I have been informed that it only makes sense, for very specific applications such as Li//S-cells, to use lithium bis(trifluoromethanesulfonyl)imide (LiTFSI) and PFAS-based ethers, such as 1,1,2,2-tetrafluoroethyl 2,2,3,3-tetrafluoropropyl ether. In Bolloré cells, LiTFSI/PEO was used in the past, whereas Li-Triflate (CAS 33454-82-9), LiBETI (CAS 132843-44-8), LiFAP (LiPF₃(CF₂CF₃)₃n, Tris(2,2,2-trifluoroethyl)borate (TFEB CAS 659-18-7) and trifluorotoluene (TFT CAS No. 98-08-8) have no current commercial relevance. In summary, according to electrolyte manufactures fluorinated ionic liquids are only used in very limited applications in LIBs, e.g. as additives. The manuscript may exaggerate the current situation, based on the discussions that I have had. A more thorough review and some discussion with electrolyte manufacturers is needed to back up the strong**

claims made in the manuscript. LIB manufacturing is an important industry for the green energy transition. Scientific accuracy is therefore important so as to properly weigh the risks and benefits of LIBs.

Thank you for the feedback. Although these insights are interesting, the reviewer has requested we implement modifications to this manuscript based on *unpublished and personal conversations* with industry representatives, which we cannot verify. The primary question of relevance in this feedback appears to be the use of bis-FASIs as electrolytes relative to other LiB electrolytes. First, we note that **our manuscript describes multiple uses** of bis-FASIs including uses as primary or secondary electrolytes and as a component of polyvinylidene fluoride (PVDF) electrode binders and separators used in LiBs (**Lines 59-61**, and in the supplementary text, **Background**). These uses are supported by information obtained from electrolyte manufacturers and our own data as summarized below.

The reviewer has suggested that we need to speak with electrolyte manufacturers; however, **information in our manuscript was obtained directly from literature generated by electrolyte manufacturers** and manufacturers of battery binders (e.g., 3M, Solvay, and Arkema). As noted in the supplementary text, 3M and Solvay collectively account for 70% of the global production of bis-FMeSI¹³⁻¹⁵, and both companies advertise bis-FMeSI for use as an electrolyte and polymer additive on their web pages^{16,17}. For the reviewer's benefit we note the following excerpts from the manufacturers web pages that unambiguously indicate marketing of bis-FASIs as electrolytes and polymer additives:

- Solvay describes bis-FMeSI under the trade name LiTFSI and states that, “LiTFSI (or TFSILi) is a key component for Li-ion batteries in liquid electrolyte as well as solid electrolyte (LMP batteries)...”
- “Solvay lithium bis-trifluoromethanesulfonimide can be used as electrolyte additive or main salt.”
- Lastly, Solvay states that “LiTFSI and its derivatives, such as ionic liquid, acts as an antistatic agent in polymer matrix...”
- Similarly, 3M markets their product as 3M™ Battery Electrolyte HQ-115, and states that, “3M™ Battery Electrolyte HQ-115 (lithium bis-trifluoromethanesulfonimide) is a high purity electrolyte salt for use in lithium batteries. HQ-115 is widely used as an additive and as the primary salt to enhance battery performance and life.”

Peer-reviewed studies demonstrate that information sourced directly from PFAS manufacturers may not always be trustworthy¹⁸. So although we considered available information from the primary manufacturers of bis-FASIs, we disagree with the reviewer's assessment that information from industry should be considered the only and/or most reliable source of information on bis-FASI use. To verify the information from manufacturers and the literature, we included product analysis in the study. We identified bis-FMeSI and/or homologues in 11 of 17 batteries tested (**Table S7**), and hypothesize that varied levels of detection are reflective of use as a primary electrolyte vs. secondary electrolyte or additive (**Lines 316-320**). These data confirm publicly-available, published information from electrolyte manufacturers, and we view the combination of manufacturer information and original data as more valid than personal conversations relayed by the reviewer since, as mentioned, this is information we cannot verify.

Information in this response was included in the original submission, but we have edited the introduction to ensure we are absolutely clear that the companies we are describing market bis-FASIs for use in LiBs. As in the original submission, more details on the patents and advertised information are provided in the supporting text (see **Background**). **Lines 61-64** now state:

“Companies that hold patents for and/or advertise production or use of bis-perfluoroalkyl sulfonimide (bis-FASI) salts including bis-FMeSI and its longer-chain homologues (e.g., bis(pentafluoroethylsulfonyl)imide; bis-FEtSI) **for use as an electrolyte or polymer additive** include 3M, Solvay, and Arkema.”

To ensure, the full extent of bis-FASI detection in consumer products is clear, we also made edits to the “Consumer products” section. **Lines 313-317** now state:

“**Eleven** of 17 batteries contained **bis-FASIs** at mass loadings above the analytical detection limit (5 ng per battery). Mass loadings were variable, ranging from 7.2 ng to 35.6 mg, across a range of battery types and sizes (**Table S7**). The presence of **21.8 µg bis-FMeSI in a 21 g Ultrafire 14,500-type battery** and **>35 mg bis-FMeSI in a 47 g Samsung 18650-type battery** suggests its use as a primary electrolyte.”

Lastly, we note that this comment is another instance where the reviewer’s feedback is inconsistent with their suggestion to describe bis-FMeSI as a vPvM compound. As noted, this classification was created in recognition of the potential environmental impacts that could be caused by releases of relatively small amounts of persistent and mobile compounds along with the challenges in treating such constituents (see responses to **Comments 5, 8**). So even if LiPF₆ is a more common electrolyte relative to bis-FMeSI, our data demonstrate clearly that bisFASIs occur in LiBs. As a vPvM compound, even small release volumes associated with use in LiBs are of concern. **The frequency or amount of use relative to other electrolytes such as LiPF₆ does not change the conclusions of this work or the concerns associated with bis-FASI use in LiBs.**

11. I am not an ecotoxicologist but, as I understand, swimming behavior tests are not standardized tests, although they appear to be gaining acceptance in the scientific community. I’d strongly recommend that the editor reaches out to an expert ecotoxicologist to review this section of the manuscript. The authors are experts on PFAS treatment and analytical chemistry and I feel confident that this part of the manuscript is well done.

Thank you for the feedback and for the positive assessment of our expertise in PFAS treatment and analytical chemistry. We note that work by the lead authors of the *Daphnia magna* and *Danio rerio* testing (Gray, Jayasundara) has been cited over 2500 times, which supports their expertise in their respective fields. We also note that the reviewer does not claim expertise in this field but describes the experimental approach used as “gaining acceptance.” This is in direct contrast to Reviewer 2, who is an expert in the field, and who states that, “...behavioral and mitochondrial tests conducted in the animals are standard (see **Comment 16**).” Because this feedback otherwise appears to be primarily for the editor, we have not provided additional response. Additional responses focused on results of *Daphnia magna* and *Danio rerio* testing can be found in responses to comments provided by **Reviewer 2**.

12. Some more minor points:

Line 31: No temporal trend data are included so how can the authors conclude that the clean energy sector is a growing source of PFAS release? It seems likely that this is true, but the study does not confirm an increasing trend.

Thank you for the feedback. The trends with time are supported by studies that document increasing use of LiBs and our own data showing bis-FASI occurrence in LiBs. The line the reviewer references is in the Summary Paragraph, and the support for this statement becomes more evident in the main manuscript. Nevertheless, we agree that the language should be softened for the Summary Paragraph. **Line 31** now states:

“...and confirm the clean energy sector as an unrecognized and **potentially** growing source of global PFAS release.”

13. Lines 278-279: It is not clear how representative these LIBs are of products on the market? The authors should have made efforts to discuss this with the LIB industry. As I mentioned above, my discussions have indicated that bis-FASIs are not widely used in LIBs. Is the LIB industry being deliberately misleading?

Thank you for the feedback. We have addressed this in our response to **Comment 10**. Here we will additionally note that **Table S7**, which is unchanged from the original submission, documents the source of batteries including common consumer products such as iPad, iPhone, Tesla, and others). We agree that more additional assessment of bis-FASI occurrence in commercial LiBs would be valuable, but such a wide survey of occurrence beyond the 17 LiBs that we considered was outside the scope of this study.

14. Lines 353: I don't see how global release is proven by this study? The monitoring was regional and biased to an area near to manufacturing. These statements are an overextrapolation.

Thank you for the feedback. We have addressed this in our response to **Comment 7**.

Reviewer #2 (Remarks to the Author):

15. This paper by Guelfo et al. examines the effects of bis-perfluoroalkyl sulfonimides (bis-FASIs), a novel class of PFAS found in lithium-ion batteries, to *Daphnia magna* and zebrafish. This paper is a comprehensive evaluation of bis-FASIs and may be the first paper to examine the environmental impacts of this class of PFAS compounds from manufacturing until disposal. Their environmental sampling efforts are particularly notable and useful to risk assessors, since they include data from many countries (USA, Antwerp, France) and matrixes (surface water, soil, sediment etc.). The experimental design is also strong, since the paper includes a wide range of samples and concentrations (5 concentrations and a control) that span levels found in the environment.

We would like to thank Reviewer 2 for their time and effort in reviewing our manuscript, and for the favorable view of our work. We really appreciate the value that this feedback brought to the work and have addressed each comment on a point by point basis below. We hope that we have alleviated remaining concerns regarding the manuscript.

16. The behavioral and mitochondrial tests conducted in the animals are standard. However, this reviewer was unable to find any data provided on growth and survival, which are critical toxicological endpoints, and this information should be included within the publication.

Thank you for the feedback. Information on lethality and immobilization in *Daphnia magna* (*D. Magna*) were provided in the original submission in **Figure S15**. No significant lethality was found through the exposure period. The experimental design for *D. Magna* exposures was based on United States Environmental Protection Agency Method EPA-821-R-02-012, which uses neonates and evaluates impacts of a 48-hour exposure. Neonate exposure over a 48-hour duration precludes evaluation of growth and reproduction.

We also measured mortality, hatching rate, and development in *Danio rerio* (zebrafish) during exposure to bis-FMeSI and field-collected water samples, but a description of these data was unintentionally omitted from the supplementary text. There were no significant differences from controls for these endpoints in any of the concentrations of bis-FMeSI or water samples tested. As noted in the edits summarized below, we have also tested three additional, higher concentrations (2,500,000, 25,000,000, and 250,000,000 ng/L) for acute endpoints (i.e., survival hatching rate, and development) in response to this reviewer feedback. These additional experiments confirm and extend findings reported in the previous version of the manuscript. Specifically, they verify lack of acute mortality even at very high, environmentally-unrealistic concentrations.

The scope of this study was limited to very early life stages. Therefore, reproduction was not measured. However, we measured the standard length of larvae after hatching when we repeated the behavior assay experiments (see **Comment 26**). There were no differences between groups at 6 days post fertilization (dpf). To examine potential effects of developmental exposure on later life growth, we measured standard length again at 6 weeks post-fertilization, in clean (no chemical exposure) conditions. While this does not address long term changes in growth that may be attributable to early life exposure to bis-FMeSI, it does provide additional information to address the reviewer's concern regarding growth.

We have added this information as text and new figures to the supplementary text (see Toxicity: zebrafish) as well as the main manuscript. Specific edits are described below.

The following was added to the main manuscript (**lines 261-262**):

“Survival, growth, developmental teratogenicity, mitochondrial function in embryos at 30 hours post fertilization (hpf) and larval locomotion at 6 days post fertilization (dpf) were used to evaluate impacts of bis-FMeSI exposure...”

The following were added to the Materials and Methods (see **Zebrafish Exposures**):

“Three additional concentrations – 2500000, 25000000, 250000000 ng/L – were tested for acute endpoints of survival, hatching rate, and development at 144 hpf. There were 3 replicate dishes per treatment group. Larval locomotion assays had 9 replicate dishes across three biological replicates. Mitochondrial assays had 3 replicate dishes per experimental group (n=30).”

“Mortality was determined by lack of heartbeat, and those individuals were removed from the dish. Larvae were assessed for developmental deformities at 6 days post-fertilization (dpf), before larval locomotion assays (see description of assay below).”

“A subset of thirty larvae per bis-FMeSI experimental group were also imaged in lateral orientation with a Nikon SMZ1500 microscope with a Nikon DXM1200 digital camera and NIS-Elements 3.10 software. Standard lengths (mm) of these larvae were measured with ImageJ 1.52a software. Thirty larvae per group were placed in 10L tanks and placed on the AHAB system and fed Ziegler’s dry larval diet under the colony regimen for grow-out under clean conditions. At 6 weeks post-fertilization (wpf), these larvae were anesthetized with cold water, measured for standard length (mm), and returned to their tanks.”

The following text and figures were added to the supplementary text (see **Toxicity: zebrafish**):

“There were no differences from controls in any of the concentrations of bis-FMeSI or water samples tested for survival, hatching, or developmental deformities in zebrafish used for mitochondrial and locomotion assays (**Fig. S16**). In the additional three highest concentrations tested for acute endpoints only, the highest concentration (250,000,000 ng/L) had >80% of larvae with uninflated swim bladders and ≈20% of larvae with pericardial and yolk sac edema. Body lengths were the same between larvae in all groups used for mitochondrial and locomotion assays (**Fig. S17A**). At 6 wpf, body lengths were mostly similar between experimental groups. There were statistically significant differences between fish that had been exposed to 25 ng/L bis-FMeSI compared to control and 25,000 ng/L (**Fig. S17B**). Likewise, fish in the 250 ng/L were different than those in control. This is very likely due to the lower tank density in 25 ng/L (n=8) and 250 ng/L (n=9) compared to other tanks such as control (n=17). It is well established in zebrafish that fish naturally grow larger when tank densities are lower. Therefore, this is very likely due to this rather than a treatment effect.”

Fig. S16. Acute toxicity endpoints in zebrafish larvae exposed to bis-FMeSI (A-C) or field collected water samples (D-F) including survival (A,D), hatching rate (B,E), developmental deformities (C,F), and standard lengths (D,H). Points and bars represent means ± SEM. Abbreviations for types of developmental deformities: Spine – curvature of the spine; Tail – bending shortening, or alteration of the caudal tail; PE – pericardial edema; YSE – yolk sac edema; CF – craniofacial deformity; Fin – alteration of the pectoral fin or fin fold; SB –

uninflated or less inflated swim bladder. No statistical differences between any groups ($P>0.05$).

Fig. S17. Growth of zebrafish exposed to bis-FMeSI. Standard lengths (mm) at 6 days post-fertilization (A) and 6 weeks post-fertilization (B). Bars represent means \pm SEM. Statistical differences between groups ($P<0.05$).

17. Furthermore, it is unclear to the reader why behavior and mitochondrial function were selected to evaluate toxicity of bis-FASIs and no other standard endpoints like growth and survival. There is no hypothesis presented in the introduction for why these endpoints were selected and there is no link to why behavior and/or mitochondrial respiration are expected to be impacted. It is also unclear how mitochondrial function at 48 hpf relates to behavior measured at 144 hpf. Overall, the field data and environmental sampling are the major strength of this paper, while the toxicological endpoints seem to offer very little evidence of dose response effect and need clarification.

Thank you for the feedback and the opportunity to clarify the rationale for the zebrafish experimental approach. Mitochondrial function was measured between 30-34 hours post fertilization (hpf) to examine developmental toxicity of bis-FMeSI to fundamental energy metabolic processes. Changes in mitochondrial dynamics can affect neurodevelopment and fish behavior via multiple mechanisms. For example, mitochondrial integrity is critical for neuronal function and formation and sustaining energetic needs for fish movement. However, without further studies, it is difficult to discern the precise link between mitochondrial function and behavior and our data indicate the need for future analyses. The link between behavior and energy production are also addressed in more detail in our response to **Comment 25**.

We have addressed questions about growth, survival, and reproduction in our response to **Comment 16**. Additionally, we have added rationale for the selected assays to the main manuscript, and in more detail in the supplementary text:

Lines 75-78 of the manuscript now state:

“When coupled with past and current challenges associated with PFAS such as perfluorooctanoic acid (PFOA) this illustrates the need for studies of bis-FASI occurrence, toxicity, and treatability. More information is available in the supplementary text.”

The supplementary text (see **Background**) states:

“Zebrafish larval locomotion is a widely used method for identification of neurobehavioral effects and an indicator of sub-lethal and sub-teratogenic toxicity resulting from chemical exposure. Behavioral alterations have been reported in zebrafish larvae at non-teratogenic concentrations of several types of PFAS. Chemical exposure has been shown to alter cellular energy metabolism directly by causing mitochondrial dysfunction. Mitochondrial function is also considered to be a biomarker for energy metabolism, and it is one that can be studied in a vertebrate, whole organism by use of embryonic zebrafish. While behavioral alterations have been shown following exposure to PFAS of a variety of structural subclasses, mitochondrial effects are considerably less well studied for these types of compounds particularly in zebrafish. Hagenaaers et al. (2013) reported PFOA caused an increase in mitochondrial permeability as well as a decrease in electron transport chain activity likely resulting from a decrease in ATP production. Similar evaluations of bis-FASIs are not available.

18. Direct comments:

Line 29-30: list the PFAS that bis-FASIs are comparable to.

Thank you for the feedback. We have edited **Line 30** of the manuscript. It now states that:

“Here we demonstrate that occurrence, ecotoxicity, and treatability of this novel class of PFAS are comparable to PFAS such as perfluorooctanoic acid...”

19. Lines 31-33: please include the median concentration. Giving just the max with no other context is not helpful to the reader.

Thank you for the feedback. We have edited **Lines 32-34** to state that:

“U.S. and European surface water, soil, and sediment measurements confirmed bis-FASI release internationally at a median concentration of 53 parts per trillion (ppt) and concentrations as high as 2,437 ppt.”

20. Lines 47-84: This main section does a great job of explaining how bis-FASIs end up in the environment, as well as the rationale for the different areas sampled. What is missing from this section is why the only way aquatic toxicity was assessed was through behavioral assays in zebrafish and daphnia and a mitochondrial function assay in zebrafish. What was the hypothesis of the test? Why were normal parameters of toxicity, such as growth reproduction or survival, not considered or measured?

Thank you for the feedback. Based on this **Comment 17** by Reviewer 2 and feedback from Reviewer 1 (e.g., **Comments 7 and 8**), Edits to the main manuscript and the rationale for the zebrafish experimental approach were described in our response to **Comment 17**. We also added additional rationale for *D. magna* testing to the supplementary text.

The supplementary text (see **Background**) now states that:

“*Daphnia magna* (*D. Magna*) have been used to evaluate the toxicity of individual toxicants and effluent wastes for more than 90 years. These organisms are excellent indicators of toxicity because they are sensitive to low concentrations of different toxicants (1mg/L – 100 µg/L). Further, they are easily maintained in lab cultures, reproduce asexually, eliminating genetic variability in the test population, and are a representative species at the bottom freshwater food

chains. Most published *D. magna* toxicity testing data uses lethality as the main test endpoint. However, there has been a paradigm shift from lethality based toxicity testing to developing sublethal methods capable of identifying effects to environmentally relevant exposure. *D. magna* are an excellent candidate for sublethal toxicity testing as there are currently 48 sublethal effects identified across four classes (reproduction, swimming behavior, biochemical, and physiological changes). Perturbations to *D. magna* swimming behavior as a result of toxicant exposure is a consistently sensitive endpoint class with respect to dose.

Much of the sublethal endpoints identified in *D. magna* were identified using exposures to a variety of pharmaceuticals (e.g. antibiotics, beta blockers). Effects to swimming parameters have been observed as low as 500 ng/L. This effect level underscores the utility of *D. magna* as a sensitive indicator of toxicity. Further, this growing body of pharmaceutical effects data allow conclusions to be drawn between known human mechanism of action and the effects observed in *D. magna*. Compared to pharmaceuticals, there is a limited knowledge of the toxicity of PFAS to *D. magna*, none of which evaluate sublethal effects on swimming. Therefore, the aim of this work is to evaluate the effects of bis-FMeSI on *D. magna* at environmentally relevant concentrations reported in this study.”

Lastly, additional information on text that was added related to PFAS occurrence and treatability can be found in our responses to **Comments 7 and 8**.

21. Line 207: Did exposure to bis-FASIs affect the growth or survival of daphnia?

Thank you for the question. We have addressed this in our response to **Comment 16**.

22. Lines 222-225: There was no change in average velocity, just a change in the variance of velocity. Is this how one would define a neuroactive effect?

Thank you for the question. Based on this feedback, we recognize that the manuscript would benefit from additional clarification regarding use of variance as an indicator of toxicological effects and the implications of the results of bis-FMeSI exposure. Because the manuscript length needs to be consistent with the journal format, we have edited the main manuscript for clarity and provided additional discussion regarding use of variance testing in toxicological assessments in the supplementary text.

Lines 228-234 now state that:

“Prior studies have found heterogenous variance in the absence of significant changes to the mean can be an earlier and more sensitive indicator of toxicological effects in 48-hr *D. magna* exposures. Prior studies have noted that impacts to *D. magna* swimming behavior often indicate a neuroactive effect because swimming in *D. magna* is controlled by the nervous system. Although mechanism of action would require further confirmation, results here indicate that bis-FMeSI has an effect on swimming velocity at concentrations as low as 10 ng/L, consistent with levels broadly detected in monitoring regions included in this study.”

The following was added to the supplementary text (see **Toxicity: *D. magna***):

“As noted in the main manuscript, there is recognition in the literature that heterogeneity of variance can be an earlier and more sensitive indicator of toxicological effects relative to changes in the mean. Studies attribute changes in variance to experimental factors (e.g.,

analytical variability), genetic variability, and non-genetic phenotypic response. In this study, bis-FMeSI exposure concentration was the only change in experimental condition, and *D. magna* are genetically identical to each other. Thus, changes in variance of a monitored endpoint relative to the control are attributable to changes in exposure concentration. As noted above, changes in variance are evaluated by comparing variance at each exposure level to the control using a χ^2 test. Because variance is heterogenous, criteria for use of ANOVA (i.e., equal variance, normal data distribution) are not met, so there are no comparisons between doses. Studies have identified that differences in variance relative to controls may occur only in some exposure concentrations (e.g., **Fig. 2** of the manuscript), but still recognize heterogenous variance as an early toxicological indicator.

In this study, heterogenous variance indicates that exposure to bis-FMeSI has a significant effect on the variance of swimming velocity at concentrations as low as 10 ng/L. The means of swimming velocity did not differ significantly, but the outcomes of variance testing still demonstrate an effect of exposure. This is similar to prior *D. magna* studies that observed no effect of 48-hr contaminant exposure on mean oxygen consumption, but significant changes in the variance of oxygen consumption. Their results, results of additional prior work, and results herein highlight that testing homogeneity of variance has utility beyond its traditional use as criteria for use of ANOVA. The impacted endpoint (velocity) indicates an effect on swimming behavior, which is controlled in *D. magna* by the central nervous system. Because of this, prior studies have noted that changes in swimming behavior may be indicative of a neuroactive effect. Additional study would be required to confirm the mechanism of action of bis-FMeSI exposure.”

23. Lines 237: The distance/velocity figures here with just mean +/- SD are not very useful in their current format for publication. A box plot, a violin plot, and/or a plot with the individual replicates displayed, would do a better job visually displaying the change in variance the authors are attributing to bis-FASI exposure.

Thank you for the opportunity to improve the utility of the *D. magna* and zebrafish figures. **Fig. 2** of the manuscript was **changed to a box and whisker format** for data on percent time swimming, distance, and velocity, as shown below.

24. Lines 225-227: The lack of dose response, or effect to changes in the distance travelled, swimming duration or velocity, indicates to me a lack of toxicological effect.

Thank you for this comment. We have addressed this in our response to **Comment 22**.

25. Lines 253-255: What do decreases in these parameters mean for overall mitochondrial function? According to Chacko et al. 2014 Clinical Science (doi: 10.1042/CS20140101), damaged/stressed mitochondria tend to have increased basal respiration and proton leak, along with reduced reserve capacity and ATP linked respiration. In this paper, the results seem to be decreased across the board which makes it challenging to interpret what they mean for overall mitochondrial health.

Thank you for the question. Chemical exposure can impair mitochondrial function by multiple mechanisms and a decreases or increases in function compared to the control group indicate mitochondrial dysregulation or compromised mitochondrial health. However, extensive further studies are needed to interpret the mechanisms underlying altered function. It is widely accepted in the field that interpretation and inferences of these data have to be made in the context of the control samples, given the complexity of whole organisms and the ability to induce adaptive responses following exposure¹⁹. Please note that we have thoroughly reviewed the discussion by Chacko *et al.* (2014)²⁰, where they primarily derive conclusions based on cell culture studies and not whole animal studies, where adaptive responses are likely to significantly differ.

Chacko *et al.* (2014) do rightly say that this parameter should be interpreted with information from the rest of the mitochondrial profile²⁰. As the reviewer stated, we observed decreases in multiple parameters. This is likely due to decreased substrate availability or compromised mitochondrial mass or integrity. This is suggested by Chacko *et al.* (2014) whether it be linked to ATP synthesis or other mitochondrial parameters²⁰. Brand *et al.* (2011) also cites substrate supply as a reason for changes in basal respiration and maximal mitochondrial respiration²¹. These authors state that a decrease in maximum mitochondrial respiration is a strong indicator of potential mitochondrial disfunction. Together, these data show that zebrafish embryos exposed to high concentrations of bis-FMeSI do not produce as much energy in a resting state and are unable to make up that deficiency even with an artificial energy demand. We have added some of this information to the supplementary text as described below. However, we refrained from elaborating on the mitochondrial data in the current study, without in-depth mechanistic analyses that our laboratory is setup to do and look forward to conducting in the future.

The following was added to the supplementary text (see **Toxicity: Zebrafish**):

“Decreases in basal respiration indicate that embryos exposed to high concentrations of bis-FMeSI do not produce as much energy in a resting state. These differences are reflected in decreased maximum mitochondrial respiration and spare capacity, which show that these embryos are unable to make up this energy deficit even with an artificial energy demand. Such decreases in energy may be due to decreased substrate availability or compromised mitochondrial mass or integrity.”

Additionally, the supplementary text (see **Toxicity: Zebrafish**) now states that:

“While zebrafish behavioral changes has been reported following exposure to a variety of PFAS, bioenergetics is considerably less well studied. This is interesting because behavior and energy production are, in many ways, linked. Many mitochondrial defects have been shown to affect the nervous system. For example, Huang et al. (2023) found that exposure to PFHpA decreased ATP-linked respiration in zebrafish embryos and reduced locomotor activity in larvae. Patel et al. (2022) reported changes in expression levels of mitochondrial-related genes but no changes in mitochondrial function or larval locomotion. The decrease that we observed in both the behavior test as well as the mitochondrial assay suggests that exposure to bis-FMeSI is energetically costly and provides information on potential modes of action of bis-FMeSI. Increases that we observed at the lowest concentrations may be related to more than one mechanism (e.g., bioenergetics and neurological function). Additional experiments are needed to resolve these questions.”

26. 256-259: Are these behavioral results toxicity or just random variance? Are non-monotonic responses in LPR typical for zebrafish exposed to PFAS? The effect at 25 ng/L looks within the realm of control fish since the controls on the field panel looks to be higher than the fish exposed to 25 ng/L. Demonstrating that this effect is reproducible, along with testing a few concentrations below this “threshold” would convince readers that there is something going on here.

Thank you for the question and the opportunity to clarify questions surrounding reproducibility. To address these concerns, **the project team repeated the behavioral assay experiment, and as part of this effort, introduced a lower test concentration (2.5 ng/L). This more than doubled our sample size.** In this type of assay, we expect inter-individual variability. Fitzgerald et al. (2019) has stated that variability is lowest during dark phases of this type of test²². In order to determine the variance within our data, we calculated coefficients of variation (CV) for each experimental group in each light/dark phase for both bis-FMeSI and found this was the case. Additionally, CVs were similar within phase. We have included the tables below in the supplementary data (**Data S12-S13**). We are also including the figure below showing means for each experimental group across the three runs/biological replicates for bis-FMeSI. Within each run, we observed very similar trends as that of the combined data now shown in the main manuscript (revised **Fig. 3**).

There are a wide range of patterns in behavioral responses following exposure to different PFAS as reported by Gaballah et al. (2020), Menger et al. (2020), and Rericha et al. (2021)²³⁻²⁵. We are not able to say if non-monotonic responses to PFAS are “typical” for zebrafish as there are not enough studies to determine this with a high level of confidence yet. What we can say there is some data available in zebrafish that shows low concentrations can cause effects when high concentrations do not. For example, figures in Ulhaq *et al.* (2013) and Wasel *et al.* (2022) showed some greater effects at lower concentrations of some, but not all, of the PFAS tested^{26,27}. Ulhaq et al. (2013) correlates behavioral changes to PFAS chain length²⁶. Other studies, such as Rericha et al. (2022) provide some evidence that it may be related to the functional group²⁸. A systematic review of epidemiological studies in humans indicates that PFAS have the potential for non-monotonic dose response curves, particularly in response to endocrine disrupting compounds, with low concentrations/exposures exhibiting more disruptive effect than high concentrations/exposures²⁹. For example, non-monotonic dose

response curves in serum lipid concentrations have been associated with prenatal exposures to PFOS and PFOA³⁰.

The following table was added to the supplementary data file (tab **Data S12**)

Coefficients of variation for each experimental bis-FMeSI group in each phase of the zebrafish locomotion assay

	Control	2.5 ng/L	25 ng/L	250 ng/L	2,500 ng/L	25,000 ng/L	250,000 ng/L
Light 1	228.87	161.55	197.74	234.17	252.94	205.09	210.02
Dark 1	66.39	43.88	64.00	62.95	71.97	79.88	64.90
Light 2	292.02	176.45	207.92	209.89	316.73	258.69	275.76
Dark 2	69.04	50.95	69.52	75.98	72.55	74.00	62.74

Additionally, we added the following table to the supplementary data file as tab **Data S13**).

Coefficients of variation for each experimental water sample group in each phase of the zebrafish locomotion assay

	Control	Vial Blank	MN 15	MN 22	MN 4
Light 1	178.20	150.73	138.84	115.54	126.38
Dark 1	70.77	55.27	39.21	40.29	47.26
Light 2	180.64	165.70	190.84	124.29	148.63
Dark 2	76.82	61.50	52.47	42.00	51.64

The following was added here for the reviewer’s benefit to demonstrate that within each run, trends were similar to the combined data shown in the manuscript (revised **Fig. 3**)

Lastly, **Fig. 3** of the main manuscript has been revised as follows:

Fig. 3. Zebrafish larval locomotion.

Mean distance traveled (mm) during light/dark phases of the assay (mm/40min) by larvae exposed to 25-250,000 ng/L bis-FMeSI (A) and field-collected aqueous samples MN 4 (including duplicate), MN 22, and MN 15 (B). Bars represent means ± standard error of the mean. Different letters represent statistical differences within phase (P<0.05; n=30).

27. 267-272: Why is the zebrafish behavior displayed like this? The materials and methods listed this assay as a 50-minute LPR. It is unclear what this data is showing. Is this light and dark cycle data averaged together? If so, why was this done? Did exposure to these compounds affect behavior in light or dark phases, did it affect how the fish transition between phases? Overall, this is an unusual way of evaluating the LPR assay and I would have to ask, why perform light/dark transitions if averaging it all together?

Thank you for the feedback on the figure format. We originally displayed behavior data this way to simplify the results. The reviewer is correct that this is a 50-min assay. However, we typically do not include data from the 10-min habituation phase in our analyses. We have updated the figure in the main manuscript to separate out light and dark phases. The revised figure is included in the manuscript (**Fig. 3**), and is also shown in our response to **Comment 26**. This separation with additional replicates shows an even greater increase in hyperactivity at low concentrations, both in the light and dark phases.

28. 357-360: there is insufficient evidence in this study to make such strong conclusions. Based on the evidence in this study there seems to be no dose-dependent effects on behavior and potential mitochondrial dysfunction at 250,000 ng/L.

Thank you for the feedback. We agree with the reviewer that the effects of concentration that we observed for zebrafish locomotion did not follow a typical dose-response. However, non-monotonic dose responses or inverse dose response effects are emerging in sub-lethal assays (e.g. prolactin level changes with DDE and BPA are classic examples), and warrants significant attention in the field of toxicology, as also evident from our assays. More details are provided in our response to **Comment 26**, and our response to that feedback also demonstrates through addressing questions about reproducibility that non-monotonic response is not, as the reviewer had questioned, “random noise.”

This compound behaves differently than what we might expect for typical lipophilic or hydrophilic compounds, given that very low concentrations have the greatest effect on zebrafish locomotion. The magnitude of this effect may be up to interpretation, but there are statistical differences that demonstrate bis-FMeSI is showing strong effects at very low concentrations in both zebrafish and *D. magna*. This leads us to the conclusion that this compound is in need of further study is needed with this compound to identify the target(s) and mechanisms by which bis-FMeSI is acting upon.

29. Methods/Results: a summary table of measured bis-FMeSI for the daphnia and zebrafish exposures should be presented somewhere to allow better interpretation of the nominal values presented in this study.

Thank you for the feedback. The measured bis-FMeSI concentrations for *D. magna* exposure were provided in the original submission (**Table S6**), so no edits were made in response to this feedback for that organism model. The equivalent data for zebrafish have been added to the revised manuscript.

We added the following table to the supplementary data file as tab Data S16.

Nominal Concentration (ng/L)	Measured Concentration (ng/L)	Standard Deviation
Day 0		
Control	ND	ND
2.5	2.72	0.26
25	22.9	4.5
250	200	13
2,500	1,870	210
25,000	17,000	2,000
250,000	152,000	9,000
Day 6		
Control	ND	ND
2.5	3.06	0.16
25	23.0	3.6
250	214	9
2,500	1,890	230
25,000	17,000	1,000
250,000	181,000	13,000
2,500,000	2,490,000	270,000
25,000,000	23,500,000	2,100,000
250,000,000	257,000,000	16,000,000

Reviewer #3 (Remarks to the Author):

30. The paper present unexpected and worrying data showing the release of PFAS into the environment near lithium-ion battery related production facilities. The work is original and extremely important with respect to the "green credentials" of lithium-ion batteries and the race to nett zero. The paper is entirely self-contained and the data presented does support the conclusions drawn. I can see no flaws in the analyses-but I must confess to not being an expert in analysis. As far as I can see, the work is methodical and precise and good science. There is very great experimental detail provided in the paper which would allow the tests to be reproduced. I recommend publication.

We would like to thank Reviewer 3 for their time and effort in reviewing our work, and we would also like to express our appreciation for their favorable view of the manuscript.

References Cited

1. UNEP. Proposal to list long-chain perfluorocarboxylic acids, their salts and related compounds in Annexes A, B and/or C to the Stockholm Convention on Persistent Organic Pollutants.
2. US EPA. Drinking Water Health Advisory for Perfluorooctanoic Acid (PFOA). (US Environmental Protection Agency, 2016).
3. US EPA. Drinking Water Health Advisory for Perfluorooctane Sulfonate (PFOS). (US Environmental Protection Agency, 2016).
4. US EPA. IRIS Toxicological Review of Perfluorohexanoic Acid [PFHxA, CASRN 307-24-4] and Related Salts. (US Environmental Protection Agency, 2023).

5. US EPA. Maximum Contaminant Level Goal (MCLG) Summary Document for a Mixture of Four Per- and Polyfluoroalkyl Substances (PFAS): HFPO-DA and its Ammonium Salt (also known as GenX Chemicals), PFBS, PFNA, and PFHxS. (US Environmental Protection Agency, 2023).
6. BAuA. Annex XV Restriction Report, Proposal for a Restriction, Substances name: Per and polyfluoroalkyl substances (PFASs). <https://echa.europa.eu/documents/10162/f605d4b5-7c17-7414-8823-b49b9fd43aea> (2023).
7. Wang, Z. et al. A New OECD Definition for Per- and Polyfluoroalkyl Substances. *Environ. Sci. Technol.* **55**, 15575–15578 (2021).
8. Arp, H. P. H. & Hale, S. E. Assessing the Persistence and Mobility of Organic Substances to Protect Freshwater Resources. *ACS Environ. Au* **2**, 482–509 (2022).
9. Hale, S. E., Arp, H. P. H., Schliebner, I. & Neumann, M. Persistent, mobile and toxic (PMT) and very persistent and very mobile (vPvM) substances pose an equivalent level of concern to persistent, bioaccumulative and toxic (PBT) and very persistent and very bioaccumulative (vPvB) substances under REACH. *Environ Sci Eur* **32**, 155 (2020).
10. European Commission. Directorate General for the Environment., Milieu Ltd., Ökopol., Risk & Policy Analysts (RPA)., & RIVM. Study for the strategy for a non-toxic environment of the 7th Environment Action Programme: final report. (Publications Office, 2017).
11. Guelfo, J. L. et al. Evaluation and Management Strategies for Per- and Polyfluoroalkyl Substances (PFASs) in Drinking Water Aquifers: Perspectives from Impacted U.S. Northeast Communities. *Environmental Health Perspectives* **126**, 065001 (2018).
12. Rensmo, A. et al. Lithium-ion battery recycling: a source of per- and polyfluoroalkyl substances (PFAS) to the environment? *Environ. Sci.: Processes Impacts* 10.1039/D2EM00511E (2023) doi:10.1039/D2EM00511E.
13. Arthur, S. D. et al. Nonaqueous electrolyte compositions comprising lithium oxalato phosphates. (2022).
14. 360 Research Reports. Global LiTFSI Market Growth 2023-2029. 92 (2023).
15. Arkema USA. ARKEMA HPP, PVDF Electrode Binders & Separator Coatings. <https://hpp.arkema.com/en/markets-and-applications/renewable-energy/lithium-ion-battery/> (2022).
16. 3M. 3M™ Battery Electrolyte HQ-115. https://www.3m.com/3M/en_US/p/d/b00005989/ (2022).
17. Solvay. LiTFSi: Lithium Salt for Safe and Performing Batteries. <https://www.google.com/url?sa=t&rct=j&q=&esrc=s&source=web&cd=&ved=2ahUKEwjLisbPppH9AhWifDABHWgSCtIQFnoECBMQAQ&url=https%3A%2F%2Fwww.solvay.com%2Fen%2FdownloadDocument%3FfileId%3DT5Yk9C04xcifrkl8Yo%26fileName%3D24998%2520LiTFSI%2520for%2520Safe%2520and%2520Performing%2520Batteries%2520v2%26base%3DFAST&usg=AOvVaw1FZa-96nMjQ8KDJ3tSgkWa> (NA).
18. Gaber, N., Bero, L. & Woodruff, T. J. The Devil they Knew: Chemical Documents Analysis of Industry Influence on PFAS Science. *Annals of Global Health* **89**, 37 (2023).
19. Meyer, J. N., Hartman, J. H. & Mello, D. F. Mitochondrial Toxicity. *Toxicological Sciences* **162**, 15–23 (2018).
20. Chacko, B. K. et al. The Bioenergetic Health Index: a new concept in mitochondrial translational research. *Clinical Science* **127**, 367–373 (2014).
21. Brand, M. D. & Nicholls, D. G. Assessing mitochondrial dysfunction in cells. *Biochemical Journal* **435**, 297–312 (2011).

22. Fitzgerald, J. A., Kirla, K. T., Zinner, C. P. & Vom Berg, C. M. Emergence of consistent intra-individual locomotor patterns during zebrafish development. *Sci Rep* **9**, 13647 (2019).
23. Gaballah, S. et al. Evaluation of Developmental Toxicity, Developmental Neurotoxicity, and Tissue Dose in Zebrafish Exposed to GenX and Other PFAS. *Environ Health Perspect* **128**, 047005 (2020).
24. Menger, F., Pohl, J., Ahrens, L., Carlsson, G. & Örn, S. Behavioural effects and bioconcentration of per- and polyfluoroalkyl substances (PFASs) in zebrafish (*Danio rerio*) embryos. *Chemosphere* **245**, 125573 (2020).
25. Rericha, Y. et al. Behavior Effects of Structurally Diverse Per- and Polyfluoroalkyl Substances in Zebrafish. *Chem. Res. Toxicol.* **34**, 1409–1416 (2021).
26. Ulhaq, M., Örn, S., Carlsson, G., Morrison, D. A. & Norrgren, L. Locomotor behavior in zebrafish (*Danio rerio*) larvae exposed to perfluoroalkyl acids. *Aquatic Toxicology* **144–145**, 332–340 (2013).
27. Wasel, O., Thompson, K. M. & Freeman, J. L. Assessment of unique behavioral, morphological, and molecular alterations in the comparative developmental toxicity profiles of PFOA, PFHxA, and PFBA using the zebrafish model system. *Environment International* **170**, 107642 (2022).
28. Rericha, Y. et al. Sulfonamide functional head on short-chain perfluorinated substance drives developmental toxicity. *iScience* **25**, 103789 (2022).
29. Rappazzo, K., Coffman, E. & Hines, E. Exposure to Perfluorinated Alkyl Substances and Health Outcomes in Children: A Systematic Review of the Epidemiologic Literature. *IJERPH* **14**, 691 (2017).
30. Maisonet, M., Näyhä, S., Lawlor, D. A. & Marcus, M. Prenatal exposures to perfluoroalkyl acids and serum lipids at ages 7 and 15 in females. *Environment International* **82**, 49–60 (2015).

REVIEWER COMMENTS

Reviewer #1 (Remarks to the Author):

R: In my previous review of the manuscript submitted by Guelfo et al., I provided some critical but constructive comments that I thought would help improve the manuscript. I was disappointed with the response by the authors. They decided to rebut most of my comments rather than to address them. I therefore decided it was necessary to reemphasize and further clarify some of the points I made previous in this re-review. I also conducted a brief review to determine what was already known about the environmental behavior and toxicity of bisFASIs prior to this submission.

Below are the authors' responses to the comments (labeled A) of my first review and my responses (labeled R).

A1. Thank you for the summary of our manuscript and favorable view of our data quality. We would like to emphasize that our abstract does not, “conclude that bis-FASIs are comparable to PFAS that are now prohibited...” Rather we specifically state (lines 28-30) that the, “...occurrence, ecotoxicity, and treatability of this novel class of PFAS are comparable to PFAS that are now prohibited and highly regulated.”

R1: In the abstract the authors now write:

“Here we demonstrate that occurrence, ecotoxicity, and treatability of this novel class of PFAS are comparable to PFAS such as perfluorooctanoic acid that are now prohibited and highly regulated worldwide.”

I disagree strongly with this. PFOA is detected in the human blood of nearly all Americans so I don't see how the occurrence of bis-FASIs in surface waters near point sources is comparable to the global occurrence of PFOA. I could go on regarding the widespread of occurrence of PFOA in multiple environmental media and biological tissues, which is fundamentally different to what the authors present in their manuscript, but I hope the authors will get my point here. The statement is too broad. Later in the manuscript the authors are more careful about making specific statements.

A: We view this as a fundamental distinction to point out since this manuscript contains original data on occurrence, ecotoxicity, and treatability of bis-perfluoroalkyl sulfonimides (bis-FASIs). Subsequent comments by Reviewer 1 (e.g., Comments 2-4), indicate that we have stated bis-FASIs pose the same “level of concern” as perfluorooctane sulfonic acid (PFOS), perfluorohexane sulfonic acid (PFHxS), and

perfluorooctanoic acid (PFOA). In fact, we have only drawn comparisons between bisFASIs and other per- and polyfluoroalkyl substances (PFAS) in areas where we have original data to compare directly to that of prior studies (i.e., occurrence, ecotoxicity, and treatability).

R: I see this distinction, but I don't see how the occurrence in water samples near manufacturing sites is comparable to global distribution of other PFAS. However, I agree that the monitoring data reported in the manuscript are novel and interesting. I wanted to check if the other parts of the manuscript were also novel. Therefore, during the 2-week window I was given for this rereview, I conducted a brief review (only using Google) on what is already known about the bisFASIs.

I found a dossier on the toxicity on the lithium salt of bis-FMeSI (published in July 2023):

<https://nepis.epa.gov/Exe/ZyPDF.cgi/P10188AG.PDF?Dockey=P10188AG.PDF>

I found considerable information in the supporting information of this paper (indicating the ecotoxicity, persistence and sorption of bis-FMeSI):

<https://www.sciencedirect.com/science/article/pii/S0304389420318859#bib0390>

There were also sorption studies done in this paper:

<https://www.sciencedirect.com/science/article/pii/S0045653508012083>

The European Chemicals Agency also have a dossier on the lithium salt of bis-FMeSI indicating it to be ecotoxic:

<https://echa.europa.eu/sk/registration-dossier/-/registered-dossier/18080/6/1>

It appears that there is already a considerable amount of information available on bis-FMeSI. The available information is sufficient to conclude that bis-FMeSI is persistent, mobile and ecotoxic. Why didn't the authors review the literature prior to submitting their study? Do they consider the lithium salt of bis-FMeSI to be distinct in its behavior from bis-FMeSI itself?

A: To be clear, we have focused our comparisons on those supported directly by our own original data and results reported in the literature. We did not make generalized conclusions that bisFASIs are equivalent to other PFAS (i.e., that they are the same “level of concern”), and feel it is important to clarify this for the reviewers and the editor. More details regarding our comparisons of occurrence, ecotoxicity, and treatability of bis-FASIs to other PFAS are provided in our response to Comment 4.

R: I don't see how the comparison of occurrence of bis-FASIs and legacy PFAS is valid. In my opinion, it's misleading. I understand that it makes the manuscript more appealing. I wish the authors would just stick to the results being reported and not over-extrapolate the importance of their work. The novel aspect is the monitoring. The additional test data are an important addition to what was already known about bisFASIs. However, the authors misled me and the editors by not including previously published information mentioned above. I did not have time to conduct a thorough review and I may have missed additional information. The authors should have conducted a more thorough review before commencing their research, and certainly when writing the manuscript.

A2. We disagree with this assessment. It is crucial to note that we do not use the word “risk” within our manuscript. Risk assessment was not an objective of this study, so this feedback misrepresents our work. Notably, Reviewer 1's view that we have overstated risks associated with bis-FASIs is not shared by Reviewers 2 or 3, who are also experts in their fields. More details are provided in our responses to Comments 4, and 6-10.

R: Reviewer 1 is also an expert in their field. It's true that the word risk is not used in the manuscript and the authors use “concern” instead. I used the term “risk” rather loosely and opened myself up for this criticism.

I stand by my comments regarding the overselling of the manuscript, however. My concern regarding the manuscript started when I read the title (“The dirty side of clean energy: Lithium ion batteries as a source of PFAS in the environment”). This title is intentionally designed to draw attention and will antagonize those in the green energy sector. While I recognize the concern with PFAS in LiBs, there are also a number of other potentially more important environmental issues associated with LiBs (e.g. how to recycle critical metals).

A3. Thank you for the feedback. We agree with the reviewer's assessment that PFOS, PFOA, PFHxS, and compounds related to each of these are listed in the Stockholm Convention. We also note that long-chain perfluoroalkyl carboxylic acid (PFCAs; e.g., those with 8-20 carbons) are candidate persistent organic pollutants (POPs) under the Stockholm Convention¹. However, we disagree that the Stockholm Convention should be the only metric of gauging worldwide regulation of PFAS. Prominent examples of regulating additional PFAS include regulations in the United States (US) and Europe.

R: I agree with this, and why I pointed about the regulation of persistent and mobile substances in Europe.

A: The US has signed, but not ratified, the Stockholm Convention. However, the US Environmental Protection Agency (USEPA) has proposed federal drinking water maximum contaminant levels (MCLs) for PFAS on the Stockholm Convention list as well as 2 additional PFAS. Specifically, the proposed MCLs are 4 ng/L for PFOS and PFOA^{2,3}, and a hazard index (HI)-based MCL for perfluorononanoic acid (PFNA), perfluorobutane sulfonate (PFBS), PFHxS, and hexafluoropropylene oxide dimer acid (HFPO-DA, trade name GenX)^{4,5}.

Additionally, the German, Swedish, Norwegian, and Danish governments have collectively proposed to regulate PFAS as a class⁶, where the Organization for Economic Cooperation and Development (OECD) definition of PFAS (which includes bis-FASIs) is used to bracket the scope of this regulation⁷. These proposed regulations demonstrate that the scope of international focus on PFAS extends far beyond the individual compounds listed in the Stockholm Convention.

R: The PFAS Restriction Proposal is a regional proposal which is not yet enforced. There is particular push back from the green energy sector with long derogations in place, or sought. The Stockholm Convention is the most relevant global regulation for PFAS even if certain countries have not ratified the Convention.

A4. Thank you for the feedback. The reviewer here casts our toxicity tests as “nonstandardized.” We address this directly in our response to Comment 11. Bioaccumulation is further addressed in Comment 8, but measurement of bis-FASi bioaccumulation was beyond the scope of the current study and will be addressed in future work.

The reviewer’s feedback states that we have “claimed that the level of concern...” for bisFASIs, “is the same as for PFOS, PFOA, and PFHxS.” However, our manuscript does not contain statements that are this generalized. References to PFOS, PFOA, and/or PFHxS are points of comparison of specific results of our work to results of prior studies. These include:

- Occurrence: A comparison of the measured aquatic concentrations of bis-FASIs near manufacturing facilities to those of PFOS and PFOA (e.g., lines 122-124),
- Ecotoxicity: A comparison of zebrafish ecotoxicity of the bis-FASi homologue bis(trifluoromethylsulfonyl)imide (bis-FMeSI) evaluated in our study to the effects of PFOS in similar studies (e.g., lines 277-278); and

- Treatability: A comparison of bis-FMeSI treatability in granular activated carbon and ion exchange (both evaluated in our study) to removal of previously studied PFAS such as PFOS and PFOA (e.g., Fig. 4 of the manuscript).

Since it is necessary to compare study results to those of prior work, we have not made any edits related to these statements. Additionally, in Comment 5 the reviewer concurs with our assessment of mobility and persistence.

The following, general statements were also included in the manuscript:

A general statement in the summary paragraph (lines 28-30) that mentions occurrence, ecotoxicity, and treatability (see response to Comment 1).

Our conclusions state that, “In general, the challenges associated with bis-FASI occurrence, mobility, toxicity, and recalcitrance are similar to those that have been realized for other PFAS.”

No edits were made to the summary paragraph statement since again, it is related to original data generated in our study. To ensure we have limited our comparisons of bis-FASIs vs. other PFAS to areas that are specific to our study, we have made a minor clarification in the conclusions. Line 443 of the manuscript now states that: “In general, the challenges associated with bis-FASI occurrence, mobility, ecotoxicity, and recalcitrance...”.

R: I appreciate the clarifications. The statement in the summary is still too strong regarding occurrence as it's rather vague and edits are necessary (see comments above). PFOA is globally occurring in multiple media and not just at ng/L levels in the aquatic environment near manufacturing facilities. Levels of legacy PFAS in surface waters near to manufacturing facilities were also higher than the ng/L levels reported for bis-FASIs.

A5. Thank you for the suggestion. We assume the reviewer is referring to the European Commission's introduction of a very persistent very mobile (vPvM) hazard class⁸. We agree that bis-FMeSI meets criteria for designation as vPvM, and our revisions are discussed below. We also note that we find this comment combined with feedback in Comment 4 and elsewhere to be contradictory. Reviewer 1 disagrees with the concept that bis-FASIs pose the same “level of concern” as other PFAS (e.g., Comment 4). Here the reviewer seems to argue the opposite. European Commission reports and peer reviewed studies argue that persistent and mobile compounds represent an equivalent level of concern as persistent, bioaccumulative, and toxic (PBT) compounds^{9,10}, and this argument has been used to designate PFBS (a PFAS with persistence and mobility similar to bis-FMeSI) as a substance of very high

concern. To be clear, this means that vPvM PFAS are considered to carry the same “level of concern” as PFAS, such as PFOA that are considered PBT. So although we did not make generalized statements that bis-FASIs represent the same level of concern as other PFAS, this comment suggests that we would have been justified in doing so.

R: In several places the authors suggest that my comments are contradictory. That is untrue. I was trying to be constructive and provide improved arguments for the authors. The vPvM hazard classification is a regional regulation that was only implemented very recently. My own personal views regarding how PFAS should be regulated are not agreed by all. Indeed, there is a strong disagreement globally on how/if PFAS should be grouped for regulation. Currently, there is more general global agreement that substances which are P, B and T should be regulated. In summary:

- Bis-FASIs are clearly very persistent (vP).
- This study, and other studies that I found, mentioned their ecotoxicity (T), but I’m not sure if that is sufficient for global regulation. The US EPA have conducted a more thorough review of T but the authors missed this (see above).
- B in humans is not well studied, although my brief review (above) uncovered some information.

Therefore, the authors are welcome to suggest that bis-FASIs are of the same level of concern as other legacy PFAS based on their vPvM properties but that is unlikely to convince many outside of Europe. The authors certainly need to conduct a proper review of the literature and consider all information before making their conclusions.

A: Nevertheless, we agree that bis-FMeSI meets criteria for designation as vPvM, and have edited the manuscript as described below.

R: Great to hear!

A6. Thank you for the feedback. The reviewer states that monitoring is not considered sufficient for publication in high impact journals, and we agree. However, many studies published in high impact journals included monitoring as part of a multi-faced study, as presented here. To be clear, we do not rely exclusively on our monitoring results to establish the novelty or importance of our work. This is the first study to consider bis-FASI occurrence in consumer products and disposal scenarios, ecotoxicity, or treatability. Additionally, we argue that the very limited prior environmental measurements of bis-FASIs in environmental samples (i.e., detections of ≤ 2 g/L bis-FMeSI in 12 aqueous samples across two studies) do not diminish the novelty of our monitoring data. Our monitoring data are international, and unlike the previous work, we include multiple bis-FASI homologues and data for soil and sediment. The novelty

and need for our work is also supported by statements made in these prior studies about the urgent need for additional investigations of bis-FASIs. Lastly, the importance and quality of the monitoring work are supported by Reviewer 2 (see Comments 15, 17).

The reviewer's feedback suggests that further description of the prior study results vs. the contributions of the monitoring data in our study is warranted.

R: The additions are good. Unfortunately, the novelty can be questioned further now that I have uncovered a body of literature not considered by the authors.

A7. Thank you for the feedback. We disagree with the reviewer's description of the sampling as "biased" since a primary objective of this study was to evaluate release near sites of PFAS manufacturing, as stated in Lines 80-81. Use of the word bias implies that our study design prevented our ability to adequately address one of our study objectives, but our objectives did not include completion of a "national/global" inventory of bis-FASIs in the environment. In order to evaluate release near sites of PFAS manufacturing, it was necessary to sample near sites of PFAS manufacturing. We evaluated PFAS release from multiple manufacturers in order to meet this study objective.

Specifically, we sampled near 3M (2 locations), Solvay (1 location), and Arkema (2 locations) facilities. Thus, the assessment that sampling was biased towards only 3M facilities, fails to recognize a significant portion of the study scope and led us to wonder whether the study design and results were carefully considered by the reviewer.

This feedback also conflicts with comments the reviewer provided in Comment 5, where they suggested that bis-FMeSI may meet criteria for classification as vPvM. According to published studies, the new vPvM classification was developed partly in recognition that relatively small releases of vPvM compounds may cause global impacts to water resources that persist long after release has stopped⁸. Reviewer 1 suggested we make a case that bis-FMeSI may meet criteria for designation as vPvM (i.e., high potential for global distribution), and we agreed. However, here the reviewer argues inconsistently that bis-FASIs may not be a global issue.

R: I think the authors understood what I meant but decided to interpret bias in their own way. I meant that the samples were only taken near to point sources (admittedly by design). This differs from legacy PFAS where I can take a random sample from any river in populated regions and would detect PFOA or PFOS in ng/L levels. I am not being contradictory here. The authors have not demonstrated the global impacts of bis-FASIs in this manuscript. Their sampling strategy was never intended to demonstrate the global impacts of bis-FASIs as all samples were taken in the proximity of point sources. If bis-FASIs were detectable in all rivers, the Arctic, global rainwater, etc. then their global impact would be more certain.

A: The reviewer's feedback suggested it would be beneficial to clarify our objective of monitoring near manufacturing facilities. The supplementary text (see Background) now states that:

R: I appreciate the clarifications.

A: To ensure that the implications of our work are clear, we have edited the manuscript conclusions (Lines 423-427) to state that:

R: This is also an improvement.

A8. Thank you for the feedback. Use of the term "missing" in this feedback implies that data needed to support our objectives were not included in the study. However, our study objectives did not include an assessment of bioaccumulation. Additionally, this feedback is contradictory to Comment 5 provided by the reviewer, which suggests that assessments can be made based solely on persistence and mobility, factors that were evaluated in our study. As noted in our earlier response, vPvM compounds have been found to have the same level of concern as PBT compounds, and have been designated as substances of very high concern, equivalent to PBT compounds. This suggests that persistence and mobility can be used in the absence of bioaccumulation as "typical assessment criteria" for completing "chemical assessment."

R: See response to Comment 5. I'm not being contradictory. B is a much more established regulatory criterion than M. The lack of evidence for B does make the study a little weaker if the authors claim similar levels of concern to legacy PFAS. By the way, I expect that B is unlikely for bis-FASIs unless there are 6 or more fully fluorinated carbons in the structures. I found some information for B in my brief literature review (see information shared above). By the way, a figure with the structures would be very useful for readers.

A: We would also like to address the reviewer's feedback that "treatability is not a criterion used in the chemical assessment." As noted in Comment 5, PFBS has been designated as a substance of very high concern based on its vPvM classification. Specific characteristics that were considered were a high potential for exposure because PFBS can pass through drinking water treatment systems as a result of its persistence and mobility⁹. So the vPvM classification suggested by the reviewer, appears to be an approach to chemical assessment that considers treatability.

R: Good point. It is therefore great that the authors accepted my recommendation to include mention of the vPvM criteria.

A: However, most critical to this response, is that we are unaware of a requirement to always use a uniform approach for assessing chemicals and would argue that constant use of such approaches without consideration of the data that would be most informative, runs the risk of hindering scientific creativity and novelty. In this study, an objective was to evaluate treatability, so we generated the data needed to address this objective. To provide improved context regarding the inclusion of treatability studies, the following was added to the supplementary text (see Background):

R: Point taken. I agree with the added text.

A: As noted in previous work¹¹, screening treatability of PFAS also informs potential for exposure. If exposure potential is low, then additional studies of bioaccumulation and toxicity may not need immediate prioritization in future studies. If treatability studies highlight treatment challenges, exposure potential will be higher (as noted for vPvM compounds), and this highlights a more pressing need for studies of bioaccumulation. To better emphasize this point, we have added the following statement to the conclusions (Lines 436-443):

R: Good addition.

A10. Thank you for the feedback. We agree that Rensmo et al. (2021) state that, “Bis(trifluoromethylsulfonyl)imide and similar ILs have mostly been investigated for their use as corrosion inhibitors¹²,” but we disagree with their conclusion. They provide a single citation to support their statement, which we suggest that the reviewer consider more carefully. The referenced study by Assenine et al. (2021) investigates use of bis-FMeSI as a corrosion inhibitor; however, the study does not provide any evidence that bis-FMeSI is used primarily as a corrosion inhibitor vs. other applications (e.g., electrolytes). We conducted a search on Web of Science for references containing the terms “LiTFSI” (a trade name for bis-FMeSI) and “corrosion inhibitor.” This search resulted in a total of 5 publications between 2014 (none earlier) and 2023 that were cited 116 times. In contrast, the search for “LiTFSI” and “electrolyte” identified 3190 publications between 1992 and 2023 that were cited a total of 153,693 times. Thus it appears that bis-FMeSI has been far more extensively examined as an electrolyte than as a corrosion inhibitor. Because of the scarcity of information related to use of bis-FASIs as corrosion inhibitors, we have declined to make edits in response to this feedback. Additional information on the use of bis-FASIs in LiBs is provided in our response to Comment 10.

R: I wanted the authors to check that bis-FASIs were principally used in battery electrolytes and didn't have other more important uses. They are certainly only used in certain LIBs and then at much lower

quantities than the principal ingredients in the electrolytes (PF6- and BF4- which have been detected in Shanghai drinking water, <https://pubs.acs.org/doi/10.1021/acs.est.3c02718>).

A10. Thank you for the feedback. Although these insights are interesting, the reviewer has requested we implement modifications to this manuscript based on unpublished and personal conversations with industry representatives, which we cannot verify. The primary question of relevance in this feedback appears to be the use of bis-FASIs as electrolytes relative to other LiB electrolytes. First, we note that our manuscript describes multiple uses of bis-FASIs including uses as primary or secondary electrolytes and as a component of polyvinylidene fluoride (PVDF) electrode binders and separators used in LiBs (Lines 59-61, and in the supplementary text, Background). These uses are supported by information obtained from electrolyte manufacturers and our own data as summarized below.

The reviewer has suggested that we need to speak with electrolyte manufacturers; however, information in our manuscript was obtained directly from literature generated by electrolyte manufacturers and manufacturers of battery binders (e.g., 3M, Solvay, and Arkema). As noted in the supplementary text, 3M and Solvay collectively account for 70% of the global production of bis-FMeSI^{13–15}, and both companies advertise bis-FMeSI for use as an electrolyte and polymer additive on their web pages^{16,17}.

I am disappointed that the authors did not make an attempt to reach out to electrolyte manufacturers. There are several electrolyte manufacturers who are making PFAS-free alternatives and would be welcome to discuss the use of bis-FASIs in electrolytes. Their opinion is that bis-FASIs are entirely unnecessary and can be rapidly phased out and replaced with PFAS-free alternatives. I still recommend that the authors make contact with manufacturers to get a better understanding of why bis-FASIs are used.

For the reviewer's benefit we note the following excerpts from the manufacturers web pages that unambiguously indicate marketing of bis-FASIs as electrolytes and polymer additives:

- Solvay describes bis-FMeSI under the trade name LiTFSI and states that, "LiTFSI (or TFSILi) is a key component for Li-ion batteries in liquid electrolyte as well as solid electrolyte (LMP batteries)..."
- "Solvay lithium bis-trifluoromethanesulfonimide can be used as electrolyte additive or main salt."
- Lastly, Solvay states that "LiTFSI and its derivatives, such as ionic liquid, acts as an antistatic agent in polymer matrix..."
- Similarly, 3M markets their product as 3MTM Battery Electrolyte HQ-115, and states that, "3M™ Battery Electrolyte HQ-115 (lithium bis-trifluoromethanesulfonimide) is a high purity electrolyte salt for use in lithium batteries. HQ-115 is widely used as an additive and as the primary salt to enhance battery performance and life."

R: Thanks for pointing this information out to me. Bis-FASIs are not a component of all LIB electrolytes, however. It's perfectly possible to not include bis-FASIs in electrolytes. If the authors had discussed with electrolyte manufacturers they would have obtained some useful additional information about what bisFASIs are used.

A: Peer-reviewed studies demonstrate that information sourced directly from PFAS manufacturers may not always be trustworthy¹⁸. So although we considered available information from the primary manufacturers of bis-FASIs, we disagree with the reviewer's assessment that information from industry should be considered the only and/or most reliable source of information on bis-FASI use.

R: I didn't suggest that industry is the only reliable source of information! However, why not try and obtain all available information before making strong conclusions?

A: To verify the information from manufacturers and the literature, we included product analysis in the study. We identified bis-FMeSI and/or homologues in 11 of 17 batteries tested (Table S7), and hypothesize that varied levels of detection are reflective of use as a primary electrolyte vs. secondary electrolyte or additive (Lines 316-320). These data confirm publicly-available, published information from electrolyte manufacturers, and we view the combination of manufacturer information and original data as more valid than personal conversations relayed by the reviewer since, as mentioned, this is information we cannot verify.

R: I personally prefer to use all available information and the manufacturers are very knowledgeable. You have to be aware that they are biased toward their products, but that does not mean you should not try and talk to them. The fact that some batteries did not contain bis-FASIs supports my view that they are not needed in the electrolytes of LiBs. This is also the view of several electrolyte manufacturers.

A: Information in this response was included in the original submission, but we have edited the introduction to ensure we are absolutely clear that the companies we are describing market bisFASIs for use in LiBs. As in the original submission, more details on the patents and advertised information are provided in the supporting text (see Background). Lines 61-64 now state:

R: Thanks for the addition.

A: To ensure, the full extent of bis-FASI detection in consumer products is clear, we also made edits to the "Consumer products" section. Lines 313-317 now state:

R: Thanks for the addition. From talking to electrolyte manufacturers I understand that Samsung have strong patents for using bis-FASIs.

A: Lastly, we note that this comment is another instance where the reviewer's feedback is inconsistent with their suggestion to describe bis-FMeSI as a vPvM compound. As noted, this classification was created in recognition of the potential environmental impacts that could be caused by releases of relatively small amounts of persistent and mobile compounds along with the challenges in treating such constituents (see responses to Comments 5, 8). So even if LiPF₆ is a more common electrolyte relative to bis-FMeSI, our data demonstrate clearly that bisFASIs occur in LiBs. As a vPvM compound, even small release volumes associated with use in LiBs are of concern. The frequency or amount of use relative to other electrolytes such as LiPF₆ does not change the conclusions of this work or the concerns associated with bis-FASI use in LiBs.

R: The point I was trying to make is that bis-FASIs are often minor additives or not added at all. They are also not necessary to use as there are alternatives. I agree that they are of concern but I want that concern to be in the proper perspective.

A11. Thank you for the feedback and for the positive assessment of our expertise in PFAS treatment and analytical chemistry. We note that work by the lead authors of the *Daphnia magna* and *Danio rerio* testing (Gray, Jayasundara) has been cited over 2500 times, which supports their expertise in their respective fields. We also note that the reviewer does not claim expertise in this field but describes the experimental approach used as "gaining acceptance." This is in direct contrast to Reviewer 2, who is an expert in the field, and who states that, "...behavioral and mitochondrial tests conducted in the animals are standard (see Comment 16)." Because this feedback otherwise appears to be primarily for the editor, we have not provided additional response. Additional responses focused on results of *Daphnia magna* and *Danio rerio* testing can be found in responses to comments provided by Reviewer 2.

R: When I said "not standard" I was wondering if they would be considered part of a regulatory assessment or would other tests be more typically used? I'm not an expert on this topic and clearly the editor has found other experts to review this part of the manuscript.

A12. Thank you for the feedback. The trends with time are supported by studies that document increasing use of LiBs and our own data showing bis-FASI occurrence in LiBs. The line the reviewer references is in the Summary Paragraph, and the support for this statement becomes more evident in the main manuscript. Nevertheless, we agree that the language should be softened for the Summary Paragraph. Line 31 now states: "...and confirm the clean energy sector as an unrecognized and potentially growing source of global PFAS release."

R: This is a good modification.

A13. Thank you for the feedback. We have addressed this in our response to Comment 10. Here we will additionally note that Table S7, which is unchanged from the original submission, documents the source of batteries including common consumer products such as iPad, iPhone, Tesla, and others). We agree that more additional assessment of bis-FASI occurrence in commercial LiBs would be valuable, but such a wide survey of occurrence beyond the 17 LiBs that we considered was outside the scope of this study.

R: As noted in my above responses, the authors could have followed my recommendation to have a discussion with electrolyte manufacturers. I cannot share all the information I have personally gathered through such an exercise, although I want to be constructive.

A14. Lines 353: I don't see how global release is proven by this study? The monitoring was regional and biased to an area near to manufacturing. These statements are an overextrapolation. Thank you for the feedback. We have addressed this in our response to Comment 7.

R: Thanks for this response, and all your other responses!

Reviewer #2 (Remarks to the Author):

I've completed the review of the revised version of this manuscript by Guelfo et al. as well as read through their thorough responses to comments. The authors have responded and addressed the questions and concerns of the reviewers. I have a few minor comments/suggestions for the revised figures. I would like to applaud the authors for strengthening the ecotoxicological data set by expanding the range of concentrations tested. This has substantially strengthened the manuscript. I think the revised figure edits look great and that the revised figures will be more helpful and comprehensive for readers.

Comments:

General figure comments: The figure captions need to be updated (fig 2 for example is still describing the old plots and not what a box plot is) or fig 3 (what is the y axis, mm/min? the caption says mm/40 min but that can't be right).

Figure 3: I appreciate the authors sharing the three runs of data they pooled to create Figure 3. The first run looks to be statistically significantly different (controls for example are almost half of the other two runs) from the other two for all concentrations tested? Were the runs as assessed to determine if they were different before they were pooled, and if they are significantly different why would you pool them? It might make more sense to exclude the hypoactive run as a “bad run/animals not behaving in a normal manner due to any number of factors” and pool the two “normal” runs instead. There are so many factors that could affect the behavior of zebrafish from time of day to who is handling the animals. It also looks like the first run is missing a data point. Overall, these three runs really indicate just how variable behavior data is in zebrafish.

Line 288: I disagree with this line stating low concentrations may be more harmful. The data certainly indicate lower concentrations may elicit hyperactivity in the fish, but it’s a big jump to say that hyperactivity equals harm. Also, the mitochondrial dysfunction seemed to follow a dose response with higher doses causing lower mitochondrial function.

RESPONSE TO COMMENTS

Lithium-ion battery components are at the nexus of sustainable energy and environmental PFAS release

For the purposes of continuity and clarity this document integrates all comments and responses to the original submission and the revision. Centered titles (i.e., Original Submission: Comment X) and comments/responses in grey are associated with the original submission. Comments shown with numbers and **bold, black** font are new comments about the revised manuscript. Responses to these comments are in regular, black font, in a point-by-point format. Text shown in blue highlight edits that were made in this revision of the manuscript.

REVIEWER COMMENTS

Reviewer #1 (Remarks to the Author):

- 1. R: In my previous review of the manuscript submitted by Guelfo et al., I provided some critical but constructive comments that I thought would help improve the manuscript. I was disappointed with the response by the authors. They decided to rebut most of my comments rather than to address them. I therefore decided it was necessary to reemphasize and further clarify some of the points I made previous in this re-review. I also conducted a brief review to determine what was already known about the environmental behavior and toxicity of bisFASIs prior to this submission.**

Below are the authors' responses to the comments (labeled A) of my first review and my responses (labeled R).

We would again like to thank Reviewer 1 for their time and effort in reviewing our manuscript. Additional responses to Reviewer 1's reemphasis and clarification are provided in a point by point format below.

Original Submission: Comment 1

Summary: The manuscript by Guelfo et al. presents information on a class of per- and polyfluoroalkyl substances (PFASs) known as bis-perfluoroalkyl sulfonamides (bis-FASIs). They present monitoring data, conduct ecotoxicity testing and test their removal efficiency in water treatment processes. The authors conclude that bis-FASIs are “comparable to PFAS that are now prohibited and highly regulated worldwide” (lines 29-30). The analytical data are of good quality and the testing also appears to have been performed with rigor.

Thank you for the summary of our manuscript and favorable view of our data quality. We would like to emphasize that our abstract does not, “conclude that bis-FASIs are comparable to PFAS that are now prohibited...” Rather we specifically state (lines 28-30) that the, “...occurrence, ecotoxicity, and treatability of this novel class of PFAS are comparable to PFAS that are now prohibited and highly regulated.”

2. In the abstract the authors now write:

“Here we demonstrate that occurrence, ecotoxicity, and treatability of this novel class of PFAS are comparable to PFAS such as perfluorooctanoic acid that are now prohibited and highly regulated worldwide.”

I disagree strongly with this. PFOA is detected in the human blood of nearly all Americans so I don’t see how the occurrence of bis-FASIs in surface waters near point sources is comparable to the global occurrence of PFOA. I could go on regarding the widespread of occurrence of PFOA in multiple environmental media and biological tissues, which is fundamentally different to what the authors present in their manuscript, but I hope the authors will get my point here. The statement is too broad. Later in the manuscript the authors are more careful about making specific statements.

Since the author’s feedback is focused on occurrence, we presume the reviewer has no issue with our statement in the summary paragraph with respect to ecotoxicity and treatability. We agree with the reviewer’s assessment of PFOA occurrence in the blood of the world population and concur that there is no data to support (or refute) whether bis-FASIs occur in the blood of humans or any biological tissues. The matrices in which we evaluated occurrence are clear from the summary paragraph and the manuscript. For example, in the summary paragraph where we state that, “U.S. and European surface water, soil, and sediment measurements confirmed bis-FASI release internationally...” Later in our manuscript, we cite references that demonstrate these concentrations of bis-FASIs are within ranges of PFOA concentrations measured in the same media near manufacturing releases, which is equivalent to this study. Resultingly, we are uncertain why the reviewer chooses to extend the definition of occurrence beyond our defined meaning to additional media such as blood.

We feel that the following edit is as explicit as is reasonable within the word limits of a summary paragraph. We have made the following edits to the summary paragraph.

In the summary paragraph (**line 29**) we have made the following edit:

“Here we demonstrate that environmental concentrations proximal to manufacturers...”

We view this as a fundamental distinction to point out since this manuscript contains original data on occurrence, ecotoxicity, and treatability of bis-perfluoroalkyl sulfonimides (bis-FASIs).

Subsequent comments by Reviewer 1 (e.g., **Comments 2-4**), indicate that we have stated bis-FASIs pose the same “level of concern” as perfluorooctane sulfonic acid (PFOS), perfluorohexane sulfonic acid (PFHxS), and perfluorooctanoic acid (PFOA). In fact, we have only drawn comparisons between bis-FASIs and other per- and polyfluoroalkyl substances (PFAS) in areas where we have original data to compare directly to that of prior studies (i.e., occurrence, ecotoxicity, and treatability).

- 3. R: I see this distinction, but I don’t see how the occurrence in water samples near manufacturing sites is comparable to global distribution of other PFAS. However, I agree that the monitoring data reported in the manuscript are novel and interesting. I wanted to check if the other parts of the manuscript were also novel. Therefore, during the 2-week window I was given for this rereview, I conducted a brief review (only using Google) on what is already known about the bisFASIs.**

I found a dossier on the toxicity on the lithium salt of bis-FMeSI (published in July 2023): <https://nepis.epa.gov/Exe/ZyPDF.cgi/P10188AG.PDF?Dockey=P10188AG.PDF>

I found considerable information in the supporting information of this paper (indicating the ecotoxicity, persistence and sorption of bis-FMeSI): <https://www.sciencedirect.com/science/article/pii/S0304389420318859#bib0390>

There were also sorption studies done in this paper: <https://www.sciencedirect.com/science/article/pii/S0045653508012083>

The European Chemicals Agency also have a dossier on the lithium salt of bis-FMeSI indicating it to be ecotoxic: <https://echa.europa.eu/sk/registration-dossier/-/registered-dossier/18080/6/1>

It appears that there is already a considerable amount of information available on bis-FMeSI. The available information is sufficient to conclude that bis-FMeSI is persistent, mobile and ecotoxic. Why didn't the authors review the literature prior to submitting their study? Do they consider the lithium salt of bis-FMeSI to be distinct in its behavior from bis-FMeSI itself?

Sentence one of this comment states that, “I see the distinction, but I don’t see how the occurrence near manufacturing facilities is comparable to global distribution of other PFAS.” **Comment 2** also addresses the comparison of bis-FMeSI to other PFAS. In that comment, the reviewer stated that, “Later in the manuscript the authors are more careful about making specific statements.” So, we assume the similar feedback here still pertains to the sentence in the summary paragraph (no line numbers referencing other portions of the revised manuscript were provided). Since we have already addressed the feedback to the summary paragraph in our response to **Comment 2** and made associated edits, no additional response is provided to that point here. This response instead focuses on the reviewer’s feedback regarding existing bis-FASI literature. We have considered the literature suggested by the reviewer and conducted an additional search. Although a number of these studies were published after the time of our manuscript submittal, we concur that there were also some earlier studies that support outcomes of our work, and we appreciate the reviewer catching this.

We have relied primarily on peer-reviewed, published papers, and we do not consider gray literature such as the European Chemicals Agency (ECHA) dossier cited by the reviewer¹ to

be reliable as a primary source of information. This is because it is impossible to evaluate the primary data sources used for most of the values in this dossier, as those are derived from unpublished studies conducted by industry. To further demonstrate this point, we note that the United States Environmental Protection Agency (USEPA) reviewed the ECHA dossier as part of the toxicity assessment report the reviewer also mentioned. More information is provided on that below, but in their assessment, the USEPA says the following about the ECHA dossier.² “Details on methods of endpoint assessment and results were limited. Additionally, most data were provided qualitatively, and the data that were provided qualitatively were incomplete...²”

The USEPA toxicity assessment report mentioned by the reviewer was released the same month as our manuscript was originally submitted and was unavailable for evaluation before initial submission of this manuscript. We became aware of the release of this document after we submitted our work; however, it focuses on reference doses for human health and is based on mammalian toxicity data – thus it is not directly comparable to the ecotoxicological investigations we conducted. The assessment relies heavily on the ECHA dossier described above. Further, the USEPA gave the chronic reference dose established in their assessment a rating of “low confidence.” Contributing factors to this designation include “...observational bias for some endpoints... selective reporting, chemical administration, and exposure characterization... outcome assessment, and results presentation,” of the ECHA studies that formed the basis of the assessment². Nevertheless, the release of this document demonstrates enhanced regulatory scrutiny of bis-FMeSI, and we have included a short passage citing this and the ECHA dossier used in part for development of a chronic reference dose. **Lines 75 – 76** have been edited to state that:

“A limited number of studies indicate that bis-FMeSI may not be removed during conventional treatment, and **only recently has regulatory scrutiny of this compound emerged.**”

Further details have also been added to the Supplementary Material (see **Background**), which now states that:

“Occurrence of bis-FMeSI in drinking water in other regions is unknown, but the United States Environmental Protection Agency (USEPA) released a chronic reference dose (RfD) of 0.3 µg/kg_{bw}-day for bis-FMeSI based on studies in a European Chemicals Agency (ECHA) bis-FMeSI dossier. This RfD was rated as “low confidence” by USEPA based on concerns with the quality of the underlying data.”

The following references were also added in support of the above edits:

USEPA. ORD Human Health Toxicity Value for Lithium Bis[(Trifluoromethyl)Sulfonyl]Azanide (HQ-115).

ECHA. Lithium bis(trifluoromethylsulfonyl)imide Ecotoxicological Summary. <https://echa.europa.eu/sk/registration-dossier/-/registered-dossier/18080/6/1> (2020).

Reviewer 1 referenced the supporting information of <https://www.sciencedirect.com/science/article/pii/S0304389420318859#bib0390>, which is a paper by Kowalska et al. 2021.³ The supporting information of Kowalska et al (2021) included information on ready biodegradability, bioaccumulation, and toxicity in Table 1S but did not include a references cited section. There was a list of author-year citations below Table 1S, but full references were not provided. The list contained citations beyond those cited in the main

manuscript, and the information in the supporting information was often wrong or insufficient to locate those references. For example:

- A number of EC₅₀ values in Table 1S are linked to Reference 27, and the Table 1S footnote is lacking a Reference 27.
- Reference 31 (Stolte et al. 2007b) does not have a match in the main manuscript. There is a reference for Stolte et al. 2008 in the main manuscript, but it appears to be a different reference.
- Reference 34 (Torrecila et al. 2009) also was not cited in the main manuscript. We found a paper by the same author on the correct year entitled “Effect of Cationic and Anionic Chain Lengths on Volumetric, Transport, and Surface Properties of 1-Alkyl-3-methylimidazolium Alkylsulfate Ionic Liquids at (298.15 and 313.15) K,”⁴ which clearly is not the correct citation to support toxicity studies in IPC-81.
- Reference 37 (Montalban et al., 2018) was also not in the main manuscript, although there was reference in the main manuscript to Montalban et al. 2016, which *may* be the correct paper.⁵ It assesses the toxicity of ionic liquids to *Vibrio fischeri*, which was considered beyond the scope of this work.
- Reference 39 (Diaz et al. 2018) which was also used to support toxicity values was not cited in the main manuscript, and a search for this reference did not identify a match.

The above are just select examples of challenges encountered in trying to locate the primary data cited in the supporting information of Kowalska et al. (2021). So, while we appreciate the apparent value of information summarized in Table 1S of their work, errors and omissions in citations of the original studies limit use of the information. We reviewed the reference list for the main manuscript of Kowalska et al. (2021), and we also conducted a comprehensive, molecule-specific literature review for environmental fate, toxicity, and treatability of bis-FASIs. Lastly, we also reviewed the study provided by Reviewer 1 in their third link.

Importantly, our study does not comprise a comprehensive review paper on bis-FASI environmental behavior – such a manuscript might be of great interest but is outside the scope of our goals for this paper. Edits made as a result of this literature review do not capture every peer-reviewed work related to bis-FMeSI. Instead, they focus on prior studies related to the key areas of our work (i.e., occurrence, *D. Magna* and zebrafish toxicity, treatability). A summary of key outcomes of this literature search is provided below.

- We identified three relevant peer-reviewed studies that were published since the time of our initial submission.

The following edit was made to the manuscript to reflect additional detections of bis-FMeSI in aqueous samples (**lines 73-74**):

“The occurrence of bis-FMeSI at low ng/L levels in European and Chinese **environmental water, wastewater, and drinking water** was recently confirmed...”

The following edit was made to the manuscript to reflect a new study that evaluated treatability of sub ng/L bis-FMeSI concentrations (**lines 75-76**):

“A **limited number of studies indicate that bis-FMeSI may not be removed during conventional treatment...**”

Additional details of the prior treatment study were added to the Supplementary Material (see **Background**):

“Nevertheless, the USEPA RfD points to increased regulatory scrutiny of bis-FMeSI exposure and an associated need to understand treatability. A single study of powdered activated carbon treatment at a wastewater treatment plant (WWTP) found no measurable removal of bis-FMeSI, but both influent and effluent bis-FMeSI concentrations (~0.8 ng/L) were ~50% of the limit of quantitation (1.5 ng/L). Further assessment of bis-FMeSI removal using adsorption (e.g., activated carbon, ion exchange), and advanced treatment approaches (e.g., oxidation) is warranted.”

Lastly, The following citations were added to support these statements:

Hu, J. et al. Integration of target, suspect, and nontarget screening with risk modeling for per- and polyfluoroalkyl substances prioritization in surface waters. *Water Research* 233, 119735 (2023).

Jiao, E. et al. Further Insight into Extractable (Organo)fluorine Mass Balance Analysis of Tap Water from Shanghai, China. *Environ. Sci. Technol.* 57, 14330–14339 (2023).

Neuwald, I. J. et al. Efficacy of activated carbon filtration and ozonation to remove persistent and mobile substances – A case study in two wastewater treatment plants. *Science of The Total Environment* 886, 163921 (2023).

- The reviewer suggested we discuss studies related to sorption, specifically Matzke et al. 2009. We concur that this study included a compound the authors reference as IM14 (CF₃SO₂)₂N).⁶ This compound is an ionic liquid (IL) comprised of a 1-butyl-3-methyl-imidazolium cation (i.e., IM14), and bis-FMeSI (i.e., (CF₃SO₂)₂N) as the counterion. In the cited paper, the authors only evaluated sorption of the IM14 cation but did not evaluate sorption of the bis-FMeSI anion.
- In addition to Matzke et al. 2009, additional sorption studies were cited within the work by Kowalska et al. 2021³. No other sorption studies were identified using molecule-specific literature review, and additional studies cited in Kowalska et al. 2021 were not cited for reasons described below. As a result, no edits were made to our discussion of bis-FMeSI sorption. Reasons are:
 - Stepnowski 2007 studied IL sorption, but these studies did not include bis-FMeSI.⁷
 - Kowalska et al. 2021 cite studies by Li et al. 2017a and 2017b in Table 1 of their work that are supposed to include bis-FMeSI. However, the citations for Li et al. 2017a and 2017b are duplicates of each other. Nevertheless, there should be at least one paper by Li et al. 2017 entitled “Effects of Soil Characteristics on Sorption-Desorption of 1-butyl-3-methyl-imidazolium-based Ionic Liquids.” We could not locate this paper, but the associated journal is not in English, so we were unable to pursue it further.
- We found one additional, relevant study cited by Kowalska et. al related to bis-FMeSI toxicity to *Daphnia Manga* (*D. Magna*). Steudte et al. 2012 evaluated the half maximal effective concentration (EC₅₀) of bis-FMeSI in *D. magna* based on immobilization. Although we monitored immobilization, our primary focus was on more sensitive sublethal endpoints (e.g., swimming behaviors). Nevertheless, the resulting EC₅₀ of 130 mg/L supports a lack of significant trends in immobilization observed in our work, so we have added the following to the Supplementary Material (see **Toxicity: D.magna**)

“A single study measured an immobilization effective concentration (EC_{50}) of 130 mg/L, which is consistent with our observation that immobilization did not correlate with dose up to the maximum evaluated concentration of 5000 ng/L (Figure S16).”

The following reference was also added to the supporting information:

Steudte, S., Stepnowski, P., Cho, C.-W., Thöming, J. & Stolte, S. (Eco)toxicity of fluoro-organic and cyano-based ionic liquid anions. *Chem. Commun.* 48, 9382 (2012).

- Dołzonek et al. 2017 evaluated the water-phospholipid membrane partitioning coefficients (K_{MW}) of ILs including bis-FMeSI⁸. As noted in the manuscript, we agree information about bis-FASI bioaccumulation is an area of need even though it was not an objective of this work. Dołzonek et al. 2017 discuss the relationship between $\log K_{MW}$ and bioconcentration factors (BCFs). However, they indicate that the relationship should be considered a “very rough and preliminary estimate of bioconcentration potential.” In our view, a more interesting point is the similarity between the $\log K_{MW}$ of bis-FMeSI and that of PFBS (evaluated in a separate study⁹) since it further supports the similarities we discussed between bis-FMeSI and PFBS in the discussion of very persistent, very mobile (vPvM) criteria. This discussion was incorporated into the manuscript in the last revision in response to earlier comments from Reviewer 1. In the current revision, we added the following to the vPvM discussion in the Supplementary Materials (see **Treatment: Oxidation**):

“Interestingly, bis-FMeSI also has a similar phospholipid-membrane water partitioning coefficient ($\log K_{MW}$) as PFBS (PFBS $\log K_{MW}=2.63$; bis-FMeSI $\log K_{MW} = 2.5$).”

We also added the following references:

Droge, S. T. J. Membrane–Water Partition Coefficients to Aid Risk Assessment of Perfluoroalkyl Anions and Alkyl Sulfates. *Environ. Sci. Technol.* 53, 760–770 (2019).

Dołzonek, J. et al. Membrane partitioning of ionic liquid cations, anions and ion pairs – Estimating the bioconcentration potential of organic ions. *Environmental Pollution* 228, 378–389 (2017).

- Kowalska et al. 2021 cite a study of bis-FMeSI hydrolysis by Steudte et al. 2012 that also supports the vPvM discussion since it is indicative of compound persistence. The following was added to the relevant discussion in the Supplementary Material (see **Treatment: Oxidation**):

“Studies have also found that bis-FMeSI does not hydrolyze over a pH range of 1-13 and does not degrade under aerobic activated sludge or anaerobic denitrifying conditions.”

The following reference was added to support this statement:

Steudte, S. et al. Hydrolysis study of fluoroorganic and cyano-based ionic liquid anions – consequences for operational safety and environmental stability. *Green Chem.* 14, 2474 (2012).

We also added the following statement to the main manuscript (**lines 368-370**):

“Lastly, as further described in the supplementary text, these results are consistent with limited prior studies of bis-FASI transformation and have implications...”

None of these papers identified sources of bis-FASIs to the environment, and none of them were focused on the link between LiB manufacture, use, and disposal and bis-FASI release.

Such studies only add to the importance of our findings and do not diminish the novelty of our results.

To be clear, we have focused our comparisons on those supported directly by our own original data and results reported in the literature. We did not make generalized conclusions that bis-FASIs are equivalent to other PFAS (i.e., that they are the same “level of concern”), and feel it is important to clarify this for the reviewers and the editor. More details regarding our comparisons of occurrence, ecotoxicity, and treatability of bis-FASIs to other PFAS are provided in our response to **Comment 4**.

4. R: I don’t see how the comparison of occurrence of bis-FASIs and legacy PFAS is valid. In my opinion, it’s misleading. I understand that it makes the manuscript more appealing. I wish the authors would just stick to the results being reported and not over-extrapolate the importance of their work. The novel aspect is the monitoring. The additional test data are an important addition to what was already known about bisFASIs. However, the authors misled me and the editors by not including previously published information mentioned above. I did not have time to conduct a thorough review and I may have missed additional information. The authors should have conducted a more thorough review before commencing their research, and certainly when writing the manuscript.

Thank you for the feedback. We have subdivided this comment for the purposes of clarity.

- Part 1: “I don’t see how the comparison of occurrence of bis-FASIs and legacy PFAS is valid. In my opinion, it’s misleading. I understand that it makes the manuscript more appealing. I wish the authors would just stick to the results being reported and not over-extrapolate the importance of their work.”

In Comment 1, the reviewer disagrees with wording used in our Summary Paragraph and states that, “The statement is too broad. Later in the manuscript the authors are more careful about making specific statements.” In our view, Comment 1 feedback implies that wording in the summary paragraph related to bis-FASIs vs. legacy PFAS is problematic, and that issues in the main manuscript were resolved. Accordingly, we made edits to the summary paragraph as summarized in our response to **Comment 2**.

- Part 2: “The novel aspect is the monitoring. The additional test data are an important addition to what was already known about bisFASIs.”

We disagree with this statement. While we appreciate Reviewer 1’s favorable view of the monitoring effort, our additional test data are also novel. Key examples include: 1) no prior studies have confirmed bis-FMeSI in batteries associated with consumer products; 2) no prior studies have demonstrated bis-FMeSI occurrence in landfill leachate; 3) no prior studies have monitored the same sublethal *D. Magna* endpoints used in our work and none have evaluated exposures in the low ng/L; and 4) similarly, no prior studies have monitored the same sublethal endpoints on zebrafish embryos or exposures in the low ng/L. It is also important to note that at the time of our original submittal, no studies had evaluated fate of bis-FMeSI during treatment with activated carbon. Even with the publication of a single study that evaluated removal of <1 ng/L bis-FMeSI, no prior studies have evaluated removal in ion exchange processes. We’d like to provide one final and very specific example to underscore our point here. Our work is the first to demonstrate changes in

behavior of key test organisms (i.e., *D. Magna* and zebrafish) at single digit ng/L exposures. The potential for ecotoxic impacts resulting from low ng/L exposures was not previously known. It is not an addition to existing knowledge. It is an important and novel finding that was not superseded by any of the literature that the reviewer refers to, and this is just one such specific example.

- Part 3: “However, the authors misled me and the editors by not including previously published information mentioned above. I did not have time to conduct a thorough review and I may have missed additional information. The authors should have conducted a more thorough review before commencing their research, and certainly when writing the manuscript.”

In our response to **Comment 3**, we expressed our appreciation to Reviewer 1 for catching additional, prior studies that support the outcomes of our work. We refer Reviewer 1 to that response for the associated edits, but would like note that of 8 citations added that focus on bis-FMeSI, 4 (1 USEPA report and 3 peer-reviewed studies) were published after our initial submittal, 1 is gray literature (i.e., the ECHA dossier) for which original methods and data are not available, and only 3 were pre-2023 publications that constitute literature that Reviewer 1 could reasonably argue that we intentionally omitted. Of those 3, one monitors immobilization of *D. Magna* at mg/L concentrations, one studied bis-FMeSI hydrolysis, and the last studied K_{mw} of bis-FMeSI. While we agree that addition of these citations to our work is useful, we hardly think that these 3 peer reviewed publications support Reviewer 1’s opinion that we intentionally omitted literature in order to overstate the importance of our work. This implication seems especially outrageous given that the outcomes of these prior studies are supportive of, but not directly overlapping with, our results. We would also note that *Nature Communications* has a limit on the number of publications that can be cited in a main manuscript, which means that focus is on the most relevant list of prior studies. The remainder have been cited in the Supplementary Material for those reasons. Lastly, we repeat our earlier reminder that the objective of this study was not to conduct a comprehensive literature review of bis-FASIs, so only the literature pertinent to our work has been included.

Original Submission: Comment 2

My main criticism of the manuscript is that the authors oversell the risks posed by bis-FASIs based on limited data. This overselling of the significance of the work is a shame because it undermines their otherwise good work.

We disagree with this assessment. It is crucial to note that we do not use the word “risk” within our manuscript. Risk assessment was not an objective of this study, so this feedback misrepresents our work. Notably, Reviewer 1’s view that we have overstated risks associated with bis-FASIs is not shared by Reviewers 2 or 3, who are also experts in their fields. More details are provided in our responses to **Comments 4**, and **6-10**.

5. R: Reviewer 1 is also an expert in their field. It’s true that the word risk is not used in the manuscript and the authors use “concern” instead. I used the term “risk” rather loosely and opened myself up for this criticism.

I stand by my comments regarding the overselling of the manuscript, however. My concern regarding the manuscript started when I read the title (“The dirty side of clean energy: Lithium ion batteries as a source of PFAS in the environment”). This title is intentionally designed to draw attention and will antagonize those in the green energy sector. While I recognize the concern with PFAS in LiBs, there are also a number of other potentially more important environmental issues associated with LiBs (e.g. how to recycle critical metals).

Our original title was not intended to “antagonize those in the green energy sector.” Instead, our intent was to draw attention to the disadvantages of conducting science in a silo. What a Materials Engineer may see as an electrolyte with superior performance, an Environmental Engineer may recognize as an environmental hazard, and this raises important questions about how sustainability is evaluated. As for the reviewer’s view that there are “a number of other potentially more important environmental issues associated with LiBs,” it is our view that studies are needed (e.g., lifecycle analyses) that directly compare and contrast these issues. Until such evaluations are conducted, we agree that both issues are important (and both would deserve titles that draw appropriate attention), but we maintain their relative importance is a matter of Reviewer 1’s opinion. Notably, we have not claimed within our work that one issue is more important than the other.

In response to concerns that our title is intended to “antagonize,” we have edited our manuscript title as follows:

“Lithium-ion battery components are at the nexus of sustainable energy and environmental PFAS release”

In response to Reviewer 1’s comment regarding resource recovery, we have edited **Lines 424-427** of the manuscript to state that:

“Results of this study provide a clear indication that the impacts of manufacturing, use, and end-of-life management associated with infrastructure components such as LiBs require additional consideration along with other issues such as resource recovery.”

Original Submission: Comment 3

The PFAS which was prohibited and highly regulated worldwide are those PFAS listed on the Stockholm Convention and are: 1) perfluorooctane sulfonic acid (PFOS) its salts and perfluorooctane sulfonyl fluoride (PFOSF), 2) perfluorooctanoic acid (PFOA) its salts and PFOA-related compounds, and 3) perfluorohexane sulfonic acid (PFHxS) its salts and PFHxS-related compounds.

Thank you for the feedback. We agree with the reviewer's assessment that PFOS, PFOA, PFHxS, and compounds related to each of these are listed in the Stockholm Convention. We also note that long-chain perfluoroalkyl carboxylic acid (PFCAs; e.g., those with 8-20 carbons) are candidate persistent organic pollutants (POPs) under the Stockholm Convention¹⁰. However, we disagree that the Stockholm Convention should be the only metric of gauging worldwide regulation of PFAS. Prominent examples of regulating additional PFAS include regulations in the United States (US) and Europe.

6. R: I agree with this, and why I pointed about the regulation of persistent and mobile substances in Europe.

Thank you for your original comment- we are glad it is resolved!

The US has signed, but not ratified, the Stockholm Convention. However, the US Environmental Protection Agency (USEPA) has proposed federal drinking water maximum contaminant levels (MCLs) for PFAS on the Stockholm Convention list as well as 2 additional PFAS. Specifically, the proposed MCLs are 4 ng/L for PFOS and PFOA^{11,12}, and a hazard index (HI)-based MCL for perfluorononanoic acid (PFNA), perfluorobutane sulfonate (PFBS), PFHxS, and hexafluoropropylene oxide dimer acid (HFPO-DA, trade name GenX)^{13,14}. Additionally, the German, Swedish, Norwegian, and Danish governments have collectively proposed to regulate PFAS as a class¹⁵, where the Organization for Economic Cooperation and Development (OECD) definition of PFAS (which includes bis-FASIs) is used to bracket the scope of this regulation¹⁶. These proposed regulations demonstrate that the scope of international focus on PFAS extends far beyond the individual compounds listed in the Stockholm Convention.

7. R: The PFAS Restriction Proposal is a regional proposal which is not yet enforced. There is particular push back from the green energy sector with long derogations in place, or sought. The Stockholm Convention is the most relevant global regulation for PFAS even if certain countries have not ratified the Convention.

Our paper does not cover global PFAS regulations to this level of detail, and our understanding is that this is informational and not a request for modifications to the work. So, no edits have been made, but we appreciate the information.

Original Submission: Comment 4

PFOS, PFOA and PFHxS are known to be bioaccumulative in humans and also to cause a range of (eco)toxicological effects to humans and wildlife. Guelfo et al. have demonstrated the presence of bis-FASIs in Minnesota at parts per trillion to low parts per million levels (particularly near manufacturing point sources) and shown that they cause ecotoxicological effects in non-standardized toxicity tests. They have not demonstrated bioaccumulation potential and cannot claim that the level of concern is the same as for PFOS, PFOA and PFHxS.

Thank you for the feedback. The reviewer here casts our toxicity tests as “non-standardized.” We address this directly in our response to **Comment 11**. Bioaccumulation is further addressed in **Comment 8**, but measurement of bis-FASI bioaccumulation was beyond the scope of the current study and will be addressed in future work.

The reviewer’s feedback states that we have “claimed that the level of concern...” for bis-FASIs, “is the same as for PFOS, PFOA, and PFHxS.” However, our manuscript **does not** contain statements that are this generalized. References to PFOS, PFOA, and/or PFHxS are points of comparison of specific results of our work to results of prior studies. These include:

- Occurrence: A comparison of the measured aquatic concentrations of bis-FASIs near manufacturing facilities to those of PFOS and PFOA (e.g., **lines 122-124**),
- Ecotoxicity: A comparison of zebrafish ecotoxicity of the bis-FASI homologue bis(trifluoromethylsulfonyl)imide (bis-FMeSI) evaluated in our study to the effects of PFOS in similar studies (e.g., **lines 277-278**); and
- Treatability: A comparison of bis-FMeSI treatability in granular activated carbon and ion exchange (both evaluated in our study) to removal of previously studied PFAS such as PFOS and PFOA (e.g., **Fig. 4** of the manuscript).

Since it is necessary to compare study results to those of prior work, we have not made any edits related to these statements. Additionally, in **Comment 5** the reviewer concurs with our assessment of mobility and persistence.

The following, general statements were also included in the manuscript:

- A general statement in the summary paragraph (**lines 28-30**) that mentions occurrence, ecotoxicity, and treatability (see response to **Comment 1**).
- Our conclusions state that, “In general, the challenges associated with bis-FASI occurrence, mobility, toxicity, and recalcitrance are similar to those that have been realized for other PFAS.”

No edits were made to the summary paragraph statement since again, it is related to original data generated in our study. To ensure we have limited our comparisons of bis-FASIs vs. other PFAS to areas that are specific to our study, we have made a minor clarification in the conclusions. **Line 443** of the manuscript now states that:

“In general, the challenges associated with bis-FASI occurrence, mobility, ecotoxicity, and recalcitrance...”

8. R: I appreciate the clarifications. The statement in the summary is still too strong regarding occurrence as it’s rather vague and edits are necessary (see comments above).

PFOA is globally occurring in multiple media and not just at ng/L levels in the aquatic environment near manufacturing facilities. Levels of legacy PFAS in surface waters near to manufacturing facilities were also higher than the ng/L levels reported for bis-FASIs.

Thank you for the feedback. We have already made edits to the Summary paragraph in response to earlier comments. Here we would like to note that our maximum detections were not “just at ng/L levels.” Our maximum detections were in the $\mu\text{g/L}$ levels in the aquatic environment, and notably were among the highest concentrations of any PFAS detected within the range of 1.04 (PFMBA) - 5,501 (PFBA) ng/L (see manuscript **line 116**). Notably, these concentrations are also similar to historical concentrations of legacy PFAS from the same facility (3M, Cottage Grove) detected near our location MN-4. In historical data, samples were collected from the wastewater treatment outfall, which is upstream from MN-4 on 3M’s private property. Dilution occurs between the outfall and MN-4 because the outfall releases into a natural creek before flowing under a railroad track and merging with the Mississippi River. In our study, sampling location MN-4 was located downstream of the outfall (on public property) and just upstream of the confluence with the Mississippi River.¹⁷ Despite this dilution, our results of 2.4 $\mu\text{g/L}$ are only one to two orders of magnitude lower than average concentrations of key legacy PFAS reported by 3M from 1996-2001. Notably, this was a period of peak PFAS production and little, if any, mitigation of PFAS outfall concentrations. Specifically, 3M reported average outfall concentrations of perfluorohexane sulfonate (PFHxS; 11 $\mu\text{g/L}$), perfluorobutane sulfonate (PFBS; 64 $\mu\text{g/L}$), perfluoroheptanoic acid (PFHpA; 14 $\mu\text{g/L}$), and perfluorohexanoic acid (PFHxA; 29 $\mu\text{g/L}$) in the tens of ug/L and concentrations of perfluorooctane sulfonate (PFOS; 262 $\mu\text{g/L}$) and perfluorooctanoic acid (PFOA; 216 $\mu\text{g/L}$) in the 100s of ug/L.¹⁷ It is quite possible, and even likely, that dilution corrected bis-FMeSI concentrations at MN-4 would be well within these ranges. Regardless, we have made edits to clarify the Summary Paragraph, as summarized in our response to **Comment 2**.

Original Submission: Comment 5

In Europe, regulators are using a new paradigm for regulation based on if a substance is persistent, mobile and toxic (PMT) or very persistent (vP) and very mobile (vM). Bis-FASIs seem to meet the PMT/vPvM criteria (M indicated by low log K_d value) which would them candidates for listing as Substances of Very High Concern in the European Union. I would have preferred arguments along these lines.

Thank you for the suggestion. We assume the reviewer is referring to the European Commission's introduction of a very persistent very mobile (vPvM) hazard class¹⁸. We agree that bis-FMeSI meets criteria for designation as vPvM, and our revisions are discussed below. We also note that we find this comment combined with feedback in **Comment 4** and elsewhere to be contradictory. Reviewer 1 disagrees with the concept that bis-FASIs pose the same "level of concern" as other PFAS (e.g., Comment 4). Here the reviewer seems to argue the opposite. European Commission reports and peer reviewed studies argue that persistent and mobile compounds represent an equivalent level of concern as persistent, bioaccumulative, and toxic (PBT) compounds^{19,20}, and this argument has been used to designate PFBS (a PFAS with persistence and mobility similar to bis-FMeSI) as a substance of very high concern. To be clear, this means that vPvM PFAS are considered to carry the same "level of concern" as PFAS, such as PFOA that are considered PBT. **So, although we did not make generalized statements that bis-FASIs represent the same level of concern as other PFAS, this comment suggests that we would have been justified in doing so.**

9. R: In several places the authors suggest that my comments are contradictory. That is untrue. I was trying to be constructive and provide improved arguments for the authors. The vPvM hazard classification is a regional regulation that was only implemented very recently. My own personal views regarding how PFAS should be regulated are not agreed by all. Indeed, there is a strong disagreement globally on how/if PFAS should be grouped for regulation. Currently, there is more general global agreement that substances which are P, B and T should be regulated. In summary:

- **Bis-FASIs are clearly very persistent (vP).**
- **This study, and other studies that I found, mentioned their ecotoxicity (T), but I'm not sure if that is sufficient for global regulation. The US EPA have conducted a more thorough review of T but the authors missed this (see above).**
- **B in humans is not well studied, although my brief review (above) uncovered some information.**

Therefore, the authors are welcome to suggest that bis-FASIs are of the same level of concern as other legacy PFAS based on their vPvM properties but that is unlikely to convince many outside of Europe. The authors certainly need to conduct a proper review of the literature and consider all information before making their conclusions

Thank you for the feedback. Comments regarding the available bis-FMeSI literature have already been addressed in our responses to **Comments 3** and **4**. We have also pointed out concerns regarding the literature that would support an assessment of toxicity (T) and bioaccumulation (B) in our response to **Comment 3**. However, it is worth summarizing here that the USEPA gave their own toxicity assessment a ranking of "low confidence," and this is because it relied heavily on the ECHA dossier.² The EPA had many concerns with the quality

of the information presented in the ECHA dossier that mirrored our own. These include a lack of method reporting, qualitative (vs. quantitative reporting of data), partial and selective reporting of data, observational bias, and questions regarding chemical administration, characterization of exposure, and assessment and presentation of the results.² Based on this, in our view it would be imprudent to rely on this information for an assessment of T. Similarly, in their study of bis-FMeSI K_{mw} Dołzonek et al. 2017 caution that their relationship between that K_{MW} and BCFs is a “very rough and preliminary estimate of bioconcentration potential.” Based on the advice of the authors, it appears it would be preliminary to use K_{mw} to make conclusions about B characteristics of bis-FMeSI, especially since that was not an objective of our study. As such, we have maintained the original vPvM discussion as suggested by Reviewer 1 in their original comments, but we have added edits and references to the relevant information on T and B (see **Comment 3**).

Nevertheless, we agree that bis-FMeSI meets criteria for designation as vPvM and have edited the manuscript as described below.

10. R: Great to hear!

No additional response or edits are required in response to this feedback. Thank you to the reviewer for providing it.

Lines 409-415 state that:

“High persistence and high mobility (i.e., weak sorption; **Fig. S9**) of bis-FMeSI may meet criteria for classification as a very persistent very mobile (vPvM) compound under recently proposed European Commission hazard classifications. Bis-FMeSI has similar persistence and mobility as PFBS, which is a vPvM PFAS recently designated under European guidelines as a substance with an equal level of concern as persistent, bioaccumulative, and toxic substances. More details are in the supplementary text.”

The following has been added to the supplementary text (see **Treatment: Oxidation**):

“The European Commission’s (EC’s) recently proposed addition of a very persistent very mobile (vPvM) hazard class to multiple regulations. Under the vPvM paradigm, biodegradation $\frac{1}{2}$ -life is used to evaluate persistence and mobility is evaluated using the organic carbon-water partitioning coefficient (K_{oc}). Specifically, compounds with $\log K_{oc} < 2.0$ and biodegradation $\frac{1}{2}$ -life > 180 days meet criteria for designation as vPvM. Field K_d values for bis-FMeSI were similar to PFPeA and PFBS (**Fig. S9**), both of which have $\log K_{oc}$ values < 2 , suggesting that bis-FMeSI is very mobile. As observed for PFOA and PFOS in prior studies, bis-FMeSI was recalcitrant under alkaline, oxidative conditions (**Fig. S19**). Although confirmation is needed, these results suggest that bis-FASIs also will not undergo degradation in the environment, which is also true of PFOA and PFOS. Persistence and mobility data presented here suggest that bis-FMeSI meets criteria for classification as vPvM under the new EC hazard classification.

The vPvM classification recognizes the potential for compounds to impact aqueous systems over large (e.g., global) regions for extended periods of time, even if production volumes are low relative to other contaminants. In fact, the European Commission has recognized that the concerns about contaminant mobility should be considered equivalent to concerns about bioaccumulation. PFBS is designated as a vPvM compound, and in 2019 it was identified as a substance of very high concern (SVHC) based on its equivalent level of concern (ELoC) to persistent, bioaccumulative,

and toxic (PBT) substances. The decision to designate PFBS as a SVHC was based on characteristics including difficulty in removing PFBS during drinking water treatment, continuous exposure as a result of persistence and mobility, accumulation in organisms at an equilibrium level as a result of continuous exposure, high global transport potential based on low volatility, high solubility, and low sorption to soils and sediments. A recent study used systematic case studies to demonstrate that vPvM substances across contaminant classes pose ELoC to PBT substances under REACH. Based on their outcomes, it is reasonable to hypothesize that bis-FMeSI, which has persistence and mobility similar to PFBS, also poses ELoC as PBT substances. Importantly this does not supersede the value in additional assessments of bis-FASI bioaccumulation and toxicity.”

Original Submission: Comment 6

A few other major concerns are listed below:

This was not the first report of bis-FASIs in the environment as the authors admit, and this further reduces the significance of the work. Although monitoring of bis-FASIs is needed monitoring is not usually considered sufficient for publication in high-impact journals.

Thank you for the feedback. The reviewer states that monitoring is not considered sufficient for publication in high impact journals, and we agree. However, many studies published in high impact journals included monitoring as part of a multi-faced study, as presented here. To be clear, we do not rely exclusively on our monitoring results to establish the novelty or importance of our work. This is the first study to consider bis-FASI occurrence in consumer products and disposal scenarios, ecotoxicity, or treatability. Additionally, we argue that the very limited prior environmental measurements of bis-FASIs in environmental samples (i.e., detections of ≤ 2 g/L bis-FMeSI in 12 aqueous samples across two studies) do not diminish the novelty of our monitoring data. Our monitoring data are international, and unlike the previous work, we include multiple bis-FASI homologues and data for soil and sediment. The novelty and need for our work are also supported by statements made in these prior studies about the urgent need for additional investigations of bis-FASIs. Lastly, the importance and quality of the monitoring work are supported by Reviewer 2 (see **Comments 15, 17**).

The reviewer's feedback suggests that further description of the prior study results vs. the contributions of the monitoring data in our study is warranted. **Lines 77-78** of the manuscript now state:

“More information is available in the supplementary text.”

The following was added to the supplementary text (see **Background**):

“There are limited prior reports of bis-FMeSI in surface water and riverbank filtration samples. Neuwald et al. (2022) detected bis-FMeSI in 9 samples at concentrations of ≤ 2 ng/L (median 0.8 ng/L) in surface water and riverbank filtration samples, and Wang et al (2023) detected bis-FMeSI in 3 sea water samples at concentrations of 0.296-1.5 ng/L. There are no reports of bis-FMeSI in soil or sediment, and no reports of bis-FMeSI homologues such as bisperfluoroethanesulfonimide (bis-FEtSI) and bisperfluorobutanesulfonimide (bis-FBSI) in any environmental media. These prior studies, which referred to bis-FMeSI as NTF2, note the urgent need for additional bis-FASI investigation. Wang et al. (2023) noted that, “...further field data on the environmental behavior of NTF2 is urgently needed.” In reference to bis-FMeSI, Neuwald et al. (2022) stated, “Currently, the lack of occurrence data makes it impossible to evaluate if its use in energy storage leads to its environmental release.” Conclusions by these authors further highlight the novelty and immediacy of our study objectives and results. Importantly, we will demonstrate detections of bis-FMeSI three orders of magnitude higher than those of previous studies, the first bis-FMeSI detections in soil and sediment, and the first detections of bis-FBSI or bis-FEtSI in any environmental media.”

11. R: The additions are good. Unfortunatelt, the novelty can be questioned further now that I have uncovered a body of literature not considered by the authors.

Thank you for the feedback. Since Reviewer 1 is satisfied with additions made in response their original feedback, no further response or manuscript edits are provided here. However,

we would like to address the question of novelty, which has been raised in multiple comments. The reviewer has stated that they support the novelty of our monitoring work, so we assume questions of novelty pertain to other aspects of our work. We addressed this in our responses to **Comments 3** and **4** but will briefly reiterate here that our additional test data are also novel. As we have noted, no prior studies have confirmed bis-FMeSI in batteries associated with consumer products; no prior studies have demonstrated bis-FMeSI occurrence in landfill leachate; no prior studies have monitored the same sublethal *D. Magna* endpoints used in our work and none have evaluated exposures in the low ng/L; and similarly, no prior studies have monitored the same sublethal endpoints on zebrafish embryos or exposures in the low ng/L. It is also important to note that at the time of our original submittal, no studies had evaluated fate of bis-FMeSI during treatment with activated carbon. Even with the publication of a single study that evaluated removal of <1 ng/L bis-FMeSI, no prior studies have evaluated removal in ion exchange processes. These areas of novelty remain true despite additional literature considered by Reviewer 1 and the author team in the current submission of our work.

Original Submission: Comment 7

Also, the sampling was biased toward areas close to 3M manufacturing sites so it's difficult to get a representative picture of the national/global environmental contamination with bis-FASIs.

Thank you for the feedback. We disagree with the reviewer's description of the sampling as "biased" since a primary objective of this study was to evaluate release near sites of PFAS manufacturing, as stated in **Lines 80-81**. Use of the word bias implies that our study design prevented our ability to adequately address one of our study objectives, but our objectives did not include completion of a "national/global" inventory of bis-FASIs in the environment. In order to evaluate release near sites of PFAS manufacturing, it was necessary to sample near sites of PFAS manufacturing. We evaluated PFAS release from multiple manufacturers in order to meet this study objective. Specifically, we sampled near 3M (2 locations), Solvay (1 location), and Arkema (2 locations) facilities. Thus, the assessment that sampling was biased towards only 3M facilities, fails to recognize a significant portion of the study scope and led us to wonder whether the study design and results were carefully considered by the reviewer.

This feedback also conflicts with comments the reviewer provided in **Comment 5**, where they suggested that bis-FMeSI may meet criteria for classification as vPvM. According to published studies, the new vPvM classification was developed partly in recognition that relatively small releases of vPvM compounds may cause global impacts to water resources that persist long after release has stopped¹⁸. Reviewer 1 suggested we make a case that bis-FMeSI may meet criteria for designation as vPvM (i.e., high potential for global distribution), and we agreed. However, here the reviewer argues inconsistently that bis-FASIs may not be a global issue.

12. R: I think the authors understood what I meant but decided to interpret bias in their own way. I meant that the samples were only taken near to point sources (admittedly by design). This differs from legacy PFAS where I can take a random sample from any river in populated regions and would detect PFOA or PFOS in ng/L levels. I am not being contradictory here. The authors have not demonstrated the global impacts of bis-FASIs in this manuscript. Their sampling strategy was never intended to demonstrate the global impacts of bis-FASIs as all samples were taken in the proximity of point sources. If bis-FASIs were detectable in all rivers, the Arctic, global rainwater, etc. then their global impact would be more certain.

The word "bias" has a clear definition, so we did not need to interpret it "in our own way." Bias is defined as, "prejudice in favor of or against one thing, person, or group compared with another, usually in a way considered to be unfair."²¹ If the reviewer didn't intend to convey this meaning, then another word should have been used in their feedback. Since Reviewer 1 did use this word, we felt it necessary to clarify in our original response that the word "bias" does not apply to our study. Failure to provide this clarification risks that an editor or another reviewer would assume that our study contains some form of bias.

Nevertheless, in this round of revisions, the issue appears to be the use of the word "global." We removed the instance of "global" we felt the reviewer disagreed with during our last revision and edited the language to read "international." Moreover, Reviewer 1 states in their feedback that they agree these edits are an improvement (see **Comment 14**) and do not request anything further. So, we are not sure where the issue remains. In the current revision, we have

searched our manuscript for all instances of the word “global” to ensure no further uses are problematic. One additional edit was made. The Summary paragraph, **line 32**, has been edited to say:

“...confirm the clean energy as an unrecognized and potentially growing source of international...”

The reviewer’s feedback suggested it would be beneficial to clarify our objective of monitoring near manufacturing facilities. The supplementary text (see **Background**) now states that:

“PFAS releases have occurred from primary manufacturers of PFAS (e.g., Chemours, Parkersburg, WV) and from sites where PFAS are used during the manufacturing process (e.g., ChemFab performance plastics in Bennington, VT). For example, atmospheric emissions from secondary manufacturer ChemFab leached into groundwater after surface deposition and caused perfluorooctanoic acid (PFOA) concentrations up to 600 ng/L in public supply and residential wells over an area >200 km². By the time this release was discovered in 2015, use of PFOA was phased out, but it had been replaced with other PFAS including perfluoro-2-propoxypropanoic acid (HFPO-DA; trade name GenX). Information about replacement PFAS is considered confidential business information in the US, so releases of compounds such as HFPO-DA were initially discovered through monitoring efforts in the Cape Fear River near a Chemours primary manufacturing facility. This led to subsequent discovery of concentrations of HFPO-DA up to 631 ng/L in raw water at a drinking water treatment plant in the same watershed. Collectively, these studies demonstrate potential for widespread environmental impacts of PFAS as a result of both primary and secondary manufacturers, and the value of independent monitoring of manufacturing sites to elucidate undisclosed releases of PFAS. Primary manufacturers have a documented history of failing to disclose information on the human health impacts of PFAS, which adds a layer of importance to independent studies of PFAS occurrence as well as toxicity and treatability.”

13. R: I appreciate the clarifications.

Thank you, we are glad the reviewer finds them helpful.

To ensure that the implications of our work are clear, we have edited the manuscript conclusions (**Lines 423-427**) to state that:

“This study demonstrates for the first time ~~the global~~ **international** and heretofore unrecognized release of novel, LiB-associated PFAS (bis-FASIs, particularly bis-FMeSI) to soil, sediment, and surface water, and that concentrations of these compounds in the parts per billion are common, near manufacturing areas. When coupled with low-level detections in three Chinese sea water samples and characteristics consistent with vPvM classification, this suggests bis-FASI release is global.”

14. R: This is also an improvement.

Thank you. This edit (see **grey font** in above paragraph) was made in response to prior concerns regarding the use of the word “global.” This response suggests that we have addressed Reviewer 1’s concerns while feedback provided in other comments (e.g., **Comment 12**), suggests we have not. As a result, it is unclear to us what specific uses of the word “global” remain concerning to Reviewer 1. Our response to **Comment 12** summarizes our efforts to find and address any remaining concerns with use of the word “global.”

Original Submission: Comment 8

The addition of the ecotoxicity tests and water treatment tests improves novelty. However, treatability is not a criterion used in the chemical assessment, and other more typical assessment criteria (e.g. bioaccumulation potential) are missing.

Thank you for the feedback. Use of the term “missing” in this feedback implies that data needed to support our objectives were not included in the study. However, our study objectives did not include an assessment of bioaccumulation. Additionally, this feedback is contradictory to **Comment 5** provided by the reviewer, which suggests that assessments can be made based solely on persistence and mobility, factors that were evaluated in our study. As noted in our earlier response, vPvM compounds have been found to have the same level of concern as PBT compounds and have been designated as substances of very high concern, equivalent to PBT compounds. This suggests that persistence and mobility can be used in the absence of bioaccumulation as “typical assessment criteria” for completing “chemical assessment.”

15. R: See response to Comment 5. I’m not being contradictory. B is a much more established regulatory criterion than M. The lack of evidence for B does make the study a little weaker if the authors claim similar levels of concern to legacy PFAS. By the way, I expect that B is unlikely for bis-FASIs unless there are 6 or more fully fluorinated carbons in the structures. I found some information for B in my brief literature review (see information shared above). By the way, a figure with the structures would be very useful for readers.

Thank you. We have addressed incorporation of additional literature in our response to **Comments 3 and 4**, and we have addressed use of that literature for the potential assessment of B in our response to **Comment 9**. A figure with structures has been added to the Supplementary Material. Thank you for the suggestion! Additionally, the main manuscript has been edited to state that (**line 61**):

“Companies that hold patents for and/or advertise production or use of bis-perfluoroalkyl sulfonimide (bis-FASI) salts (**Fig. S1**)...”

We would also like to address the reviewer’s feedback that “treatability is not a criterion used in the chemical assessment.” As noted in **Comment 5**, PFBS has been designated as a substance of very high concern based on its vPvM classification. Specific characteristics that were considered were a high potential for exposure because PFBS can pass through drinking water treatment systems as a result of its persistence and mobility¹⁹. So, the vPvM classification suggested by the reviewer, appears to be an approach to chemical assessment that considers treatability.

16. R: Good point. It is therefore great that the authors accepted my recommendation to include mention of the vPvM criteria.

Thanks, and we again extend our appreciation for the suggestion.

However, most critical to this response, is that we are unaware of a requirement to always use a uniform approach for assessing chemicals and would argue that constant use of such approaches without consideration of the data that would be most informative, runs the risk of hindering scientific creativity and novelty. In this study, an objective was to evaluate treatability, so we generated the data needed to address this objective. To provide improved context regarding the

inclusion of treatability studies, the following was added to the supplementary text (see **Background**):

“Collectively, studies of PFAS occurrence and toxicity indicate a need for effective PFAS treatment approaches, but most conventional treatment approaches for media such as drinking water are ineffective for complete removal of PFAS. Highly recalcitrant PFAS such as PFOA, perfluorooctanesulfonic acid (PFOS), and their homologues are not readily mineralized during oxidation. Under some conditions, concentrations of perfluoroalkyl carboxylates (PFCAs; i.e., PFOA and homologues) may increase during oxidation because they are terminal daughter products of oxidizable PFAS. Even PFAS that can be degraded during oxidation are recalcitrant because their terminal daughter products are still PFAS. Since oxidative approaches are routinely employed for disinfection during treatment, it is important to understand how PFAS may behave in these systems. Although removal is unlikely, parent PFAS may have different properties than their terminal oxidative transformation products, which may impact the approach used to remove those products during subsequent treatment steps (e.g., adsorption). Researchers are investigating destructive techniques for PFAS, but adsorption-based removal using granular activated carbon (GAC) and/or ion exchange (IX) resin is more readily implemented in full-scale systems. Collectively, screening novel PFAS for fate during oxidation and adsorption-based treatment will inform their recalcitrance and treatability relative to PFAS for which there are already regulatory drivers for removal (e.g., PFOA, PFOS).”

17. R: Point taken. I agree with the added text.

Thank you for the feedback.

As noted in previous work²², screening treatability of PFAS also informs potential for exposure. If exposure potential is low, then additional studies of bioaccumulation and toxicity may not need immediate prioritization in future studies. If treatability studies highlight treatment challenges, exposure potential will be higher (as noted for vPvM compounds), and this highlights a more pressing need for studies of bioaccumulation. To better emphasize this point, we have added the following statement to the conclusions (**Lines 436-443**):

“Where these treatment approaches are not in place or are not engineered for bis-FASI removal, potential for exposure may be elevated. This underscores a need for additional studies of bioaccumulation and human health impacts.”

18. R: Good addition.

Thank you for the feedback.

Original Submission: Comment 9

It is assumed that bis-FASIs are solely derived from lithium ion batteries (LIBs), but in the recent review by Rensmo et al. (ref. 21 in the manuscript) it is suggested that bis(trifluoromethylsulfonyl)imide and similar ionic liquids have mostly been investigated for their use as corrosion inhibitors. A more thorough review of the possible sources and uses of bis-FASIs is warranted.

Thank you for the feedback. We agree that Rensmo et al. (2021) state that, “Bis(trifluoromethylsulfonyl)imide and similar ILs have mostly been investigated for their use as corrosion inhibitors²³,” but we disagree with their conclusion. They provide a single citation to support their statement, which we suggest that the reviewer consider more carefully. The referenced study by Assenine et al. (2021) investigates use of bis-FMeSI as a corrosion inhibitor; however, the study does not provide any evidence that bis-FMeSI is used primarily as a corrosion inhibitor vs. other applications (e.g., electrolytes). We conducted a search on Web of Science for references containing the terms “LiTFSI” (a trade name for bis-FMeSI) and “corrosion inhibitor.” This search resulted in a total of 5 publications between 2014 (none earlier) and 2023 that were cited 116 times. In contrast, the search for “LiTFSI” and “electrolyte” identified 3190 publications between 1992 and 2023 that were cited a total of 153,693 times. Thus it appears that bis-FMeSI has been far more extensively examined as an electrolyte than as a corrosion inhibitor. Because of the scarcity of information related to use of bis-FASIs as corrosion inhibitors, we have declined to make edits in response to this feedback. Additional information on the use of bis-FASIs in LIBs is provided in our response to **Comment 10**.

19. R: I wanted the authors to check that bis-FASIs were principally used in battery electrolytes and didn't have other more important uses. They are certainly only used in certain LIBs and then at much lower quantities than the principal ingredients in the electrolytes (PF6- and BF4- which have been detected in Shanghai drinking water, <https://pubs.acs.org/doi/10.1021/acs.est.3c02718>).

At the suggestion of Reviewer 1, we investigated the use of bis-FASIs as corrosion inhibitors in our original response. In this revision we added the search combination “NtF2” and “corrosion inhibitor” and found only 17 references compared to 337 for “NtF2” and “electrolyte.” We also searched “LitFSI” and “NtF2” with the following uses for ionic liquids: “hydrosilylation” (12 references), “hydroformylation” (3 references), “carbon dioxide capture” (72 references), and “solar cell” (163, although this constitutes another electrolyte use). The numbers of literature citations summarized here still suggest that other uses of bis-FASIs are minor relative to those we have discussed in our work (i.e., LiB electrolytes and binders). However, to be sure readers are aware that other uses exist, we have added the following to the Supplementary Material (see **Background**):

“Other lesser uses for bis-FASIs are documented in the literature including use as corrosion inhibitors, in hydrosilylation, hydroformylation, carbon dioxide capture, and electrolytes in dye-sensitized solar cells. Notably, these other uses often involve bis-FMeSI with counterions other than lithium, including organic counterions.”

We have also added the following to the main manuscript conclusions (**lines 393-394**):

“Further, other uses of bis-FASIs (e.g., CO₂ capture, solar cells; see supplementary text) should be evaluated to determine their contribution to ongoing release.”

We agree with Reviewer 1’s assessment that there are other LiB electrolytes and other detections of those electrolytes in the environment; however, our objective was to assess occurrence of bis-FASIs. We cannot fully address occurrence of alternate electrolytes such as PF₆⁻ and BF₄⁻ within the scope of this study, but have added the following to the main manuscript (**lines 427-428**):

“This includes environmental impacts associated with other fluorinated, but non-PFAS LiB electrolytes PF₆⁻ and BF₄⁻ detected in drinking water.”

Original Submission: Comment 10

Based on extensive discussions with electrolyte manufacturers, it is my understanding that most LIB electrolytes are based on LiPF₆, organic cyclic and linear carbonates, and various additives. The same is true for Na-ion batteries. I have been informed that it only makes sense, for very specific applications such as Li/S-cells, to use lithium bis(trifluoromethanesulfonyl)imide (LiTFSI) and PFAS-based ethers, such as 1,1,2,2-tetrafluoroethyl 2,2,3,3-tetrafluoropropyl ether. In Bolloré cells, LiTFSI/PEO was used in the past, whereas Li-Triflate (CAS 33454-82-9), LiBETI (CAS 132843-44-8), LiFAP (LiPF₃(CF₂CF₃)₃n, Tris(2,2,2-trifluoroethyl)borate (TFEB CAS 659-18-7) and trifluorotoluene (TFT CAS No. 98-08-8) have no current commercial relevance. In summary, according to electrolyte manufactures fluorinated ionic liquids are only used in very limited applications in LIBs, e.g. as additives. The manuscript may exaggerate the current situation, based on the discussions that I have had. A more thorough review and some discussion with electrolyte manufacturers is needed to back up the strong claims made in the manuscript. LIB manufacturing is an important industry for the green energy transition. Scientific accuracy is therefore important so as to properly weigh the risks and benefits of LIBs.

Thank you for the feedback. Although these insights are interesting, the reviewer has requested we implement modifications to this manuscript based on *unpublished and personal conversations* with industry representatives, which we cannot verify. The primary question of relevance in this feedback appears to be the use of bis-FASIs as electrolytes relative to other LiB electrolytes. First, we note that **our manuscript describes multiple uses** of bis-FASIs including uses as primary or secondary electrolytes and as a component of polyvinylidene fluoride (PVDF) electrode binders and separators used in LiBs (Lines 59-61, and in the supplementary text, **Background**). These uses are supported by information obtained from electrolyte manufacturers and our own data as summarized below.

The reviewer has suggested that we need to speak with electrolyte manufacturers; however, **information in our manuscript was obtained directly from literature generated by electrolyte manufacturers** and manufacturers of battery binders (e.g., 3M, Solvay, and Arkema). As noted in the supplementary text, 3M and Solvay collectively account for 70% of the global production of bis-FMeSI²⁴⁻²⁶, and both companies advertise bis-FMeSI for use as an electrolyte and polymer additive on their web pages^{27,28}.

20. I am disappointed that the authors did not make an attempt to reach out to electrolyte manufacturers. There are several electrolyte manufacturers who are making PFAS-free alternatives and would be welcome to discuss the use of bis-FASIs in electrolytes. Their opinion is that bis-FASIs are entirely unnecessary and can be rapidly phased out and replaced with PFAS-free alternatives. I still recommend that the authors make contact with manufacturers to get a better understanding of why bis-FASIs are used.

We are glad to hear that manufacturers are considering the use of non-PFAS alternatives in LiBs. However, the use of PFAS-free electrolytes does not change the outcomes of our study which detect bis-FASIs in the environment and evaluates their treatability, ecotoxicity, and occurrence in consumer products and disposal scenarios. We agree that PFAS-free alternatives are compelling, but this is only beneficial if such alternatives are carefully vetted by both industry and independent scientists from multiple disciplines to avoid regrettable substitution

(i.e., PFAS free does not guarantee better). Review of PFAS-free electrolytes represents an effort well beyond the scope of this study, and as noted above, we have provided more evidence that use as electrolytes and binders constitute the primary uses of bis-FASIs, which are the focus of this study. As mentioned in our response to **Comment 19**, we have added reference to environmental occurrence of other fluorinated (but non-PFAS) electrolytes in the environment. In order to acknowledge efforts in developing fluorine-free alternatives and the need for careful evaluation of the safety of these alternatives, we have added the following to the manuscript conclusions (**lines 428-430**):

“Researchers are in the early stages of studying fluorine-free electrolytes, and although promising, these alternative materials also merit close evaluation of potential environmental and human health risks.”

We also added the following reference to support development of fluorine free alternatives:

Hernández, G., Mogensen, R., Younesi, R. & Mindemark, J. Fluorine-Free Electrolytes for Lithium and Sodium Batteries. *Batteries & Supercaps* 5, e202100373 (2022).

Lastly, we note that it would be challenging to include correspondence with manufacturers in this work. Nature journals require that all parties that participate in personal communication provide permission to cite that information via written communication from each party to the editor. So even if we had a mechanism by which to obtain viable information from manufacturers, this would likely pose a barrier to using it. So, we have cited evidence for concepts presented by Reviewer 1 (e.g., fluorine free alternatives) from the peer-reviewed literature.

For the reviewer’s benefit we note the following excerpts from the manufacturers web pages that unambiguously indicate marketing of bis-FASIs as electrolytes and polymer additives:

- Solvay describes bis-FMeSI under the trade name LiTFSI and states that, “LiTFSI (or TFSILi) is a key component for Li-ion batteries in liquid electrolyte as well as solid electrolyte (LMP batteries)...”
- “Solvay lithium bis-trifluoromethanesulfonimide can be used as electrolyte additive or main salt.”
- Lastly, Solvay states that “LiTFSI and its derivatives, such as ionic liquid, acts as an antistatic agent in polymer matrix...”
- Similarly, 3M markets their product as 3M™ Battery Electrolyte HQ-115, and states that, “3M™ Battery Electrolyte HQ-115 (lithium bis-trifluoromethanesulfonimide) is a high purity electrolyte salt for use in lithium batteries. HQ-115 is widely used as an additive and as the primary salt to enhance battery performance and life.”

21. R: Thanks for pointing this information out to me. Bis-FASIs are not a component of all LIB electrolytes, however. It’s perfectly possible to not include bis-FASIs in electrolytes. If the authors had discussed with electrolyte manufacturers they would have obtained some useful additional information about what bisFASIs are used.

Thank you for the feedback, but we have not claimed that bis-FASIs are a component of all LiB electrolytes. Manufacturers may reinforce that point, but it is also evident from the literature. To ensure we are clear on this point, we have added reference to other electrolytes (see response to **Comment 19**). We apologize, but the meaning of Reviewer 1’s last statement

wasn't clear to us. It states that, "If the authors had discussed with electrolyte manufacturers they would have obtained some useful additional information about what bis-FASIs are used." Because we are unclear, we have not provided additional response to that feedback.

Peer-reviewed studies demonstrate that information sourced directly from PFAS_manufacturers may not always be trustworthy²⁹. So, although we considered available information from the primary manufacturers of bis-FASIs, we disagree with the reviewer's assessment that information from industry should be considered the only and/or most reliable source of information on bis-FASI use.

22. R: I didn't suggest that industry is the only reliable source of information! However, why not try and obtain all available information before making strong conclusions?

Thank you for the question. We have provided feedback to an equivalent comment in our response to **Comments 20 and 21**.

To verify the information from manufacturers and the literature, we included product analysis in the study. We identified bis-FMeSI and/or homologues in 11 of 17 batteries tested (**Table S7**) and hypothesize that varied levels of detection are reflective of use as a primary electrolyte vs. secondary electrolyte or additive (**Lines 316-320**). These data confirm publicly available, published information from electrolyte manufacturers, and we view the combination of manufacturer information and original data as more valid than personal conversations relayed by the reviewer since, as mentioned, this is information we cannot verify.

23. R: I personally prefer to use all available information and the manufacturers are very knowledgeable. You have to be aware that they are biased toward their products, but that does not mean you should not try and talk to them. The fact that some batteries did not contain bis-FASIs supports my view that they are not needed in the electrolytes of LiBs. This is also the view of several electrolyte manufacturers.

Thank you for the feedback. We have responded and made edits in response to similar feedback in **Comments 20-22**. Briefly, we would like to reiterate that use of bis-FASI-free and even more broadly, fluorine-free alternatives does not change the fact that we have detected bis-FASIs in the environment. Additionally, we want to emphasize our earlier statement that PFAS- or fluorine-free does not automatically equate to an improvement. Alternative should be vetted by both industry and independent scientists from multiple disciplines to avoid regrettable substitution. Lastly, as noted above, it is very unlikely that we would be able to use personal communications with manufacturers within this manuscript. So, we have instead looked to the peer reviewed literature for information on non bis-FASI and fluorine-free alternatives.

Information in this response was included in the original submission, but we have edited the introduction to ensure we are absolutely clear that the companies we are describing market bis-FASIs for use in LiBs. As in the original submission, more details on the patents and advertised information are provided in the supporting text (see **Background**). **Lines 61-64** now state:

"Companies that hold patents for and/or advertise production or use of bis-perfluoroalkyl sulfonimide (bis-FASI) salts including bis-FMeSI and its longer-chain homologues (e.g., bis(pentafluoroethylsulfonyl)imide; bis-FEtSI) for use as an electrolyte or polymer additive include 3M, Solvay, and Arkema."

24. R: Thanks for the addition.

Thank you for the feedback.

To ensure, the full extent of bis-FASI detection in consumer products is clear, we also made edits to the “Consumer products” section. **Lines 313-317** now state:

“Eleven of 17 batteries contained bis-FASIs at mass loadings above the analytical detection limit (5 ng per battery). Mass loadings were variable, ranging from 7.2 ng to 35.6 mg, across a range of battery types and sizes (**Table S7**). The presence of 21.8 µg bis-FMeSI in a 21 g Ultrafire 14,500-type battery and >35 mg bis-FMeSI in a 47 g Samsung 18650-type battery suggests its use as a primary electrolyte.”

25. R: Thanks for the addition. From talking to electrolyte manufacturers I understand that Samsung have strong patents for using bis-FASIs.

Thank you for the feedback.

Lastly, we note that this comment is another instance where the reviewer’s feedback is inconsistent with their suggestion to describe bis-FMeSI as a vPvM compound. As noted, this classification was created in recognition of the potential environmental impacts that could be caused by releases of relatively small amounts of persistent and mobile compounds along with the challenges in treating such constituents (see responses to **Comments 5, 8**). So even if LiPF₆ is a more common electrolyte relative to bis-FMeSI, our data demonstrate clearly that bisFASIs occur in LiBs. As a vPvM compound, even small release volumes associated with use in LiBs are of concern. **The frequency or amount of use relative to other electrolytes such as LiPF₆ does not change the conclusions of this work or the concerns associated with bis-FASI use in LiBs.**

26. R: The point I was trying to make is that bis-FASIs are often minor additives or not added at all. They are also not necessary to use as there are alternatives. I agree that they are of concern but I want that concern to be in the proper perspective.

Thank you for the feedback. We have provided more detailed feedback and edits to similar feedback in **Comments 20-22**. Briefly, we have clarified the alternate but lesser uses of bis-FASIs including corrosion inhibitors, in hydrosilylation, hydroformylation, carbon dioxide capture, and electrolytes in dye-sensitized solar cells. We have also mentioned detections of other electrolytes highlighted by Reviewer 1 (i.e., PF₆⁻ and BF₄⁻) in the environment. Lastly, we have highlighted that there is promising research focused on fluorine free alternatives for LiB components, but with emphasis on the need for environmental screening of such alternatives. This particular study is focused on bis-FASIs, but our message is bigger than bis-FASIS or even PFAS. Specifically, we hope that drawing attention to bis-FASI release will underscore the need for heightened consideration of the impacts of manufacturing, use, and end-of-life management associated sustainable energy infrastructure in order to ensure development of truly sustainable solutions.

Original Submission: Comment 11

I am not an ecotoxicologist but, as I understand, swimming behavior tests are not standardized tests, although they appear to be gaining acceptance in the scientific community. I'd strongly recommend that the editor reaches out to an expert ecotoxicologist to review this section of the manuscript. The authors are experts on PFAS treatment and analytical chemistry and I feel confident that this part of the manuscript is well done.

Thank you for the feedback and for the positive assessment of our expertise in PFAS treatment and analytical chemistry. We note that work by the lead authors of the *Daphnia magna* and *Danio rerio* testing (Gray, Jayasundara) has been cited over 2500 times, which supports their expertise in their respective fields. We also note that the reviewer does not claim expertise in this field but describes the experimental approach used as "gaining acceptance." This is in direct contrast to Reviewer 2, who is an expert in the field, and who states that, "...behavioral and mitochondrial tests conducted in the animals are standard (see **Comment 16**)." Because this feedback otherwise appears to be primarily for the editor, we have not provided additional response. Additional responses focused on results of *Daphnia magna* and *Danio rerio* testing can be found in responses to comments provided by **Reviewer 2**.

27. R: When I said "not standard" I was wondering if they would be considered part of a regulatory assessment or would other tests be more typically used? I'm not an expert on this topic and clearly the editor has found other experts to review this part of the manuscript.

Thank you for clarifying the intent of your original feedback. As the reviewer is aware, in regulatory toxicology, animal tests are used to identify toxicity of chemicals as part of the hazard assessment process. There has been harmonization within Europe under OECD Test Guidelines and the United States under EPA Guidelines, but there are several differences between these in species and methodology and neither include bioenergetics or behavior as specific endpoints. That said, government agencies typically take many studies into account, not just those run with their guidelines, in order to make recommendations and regulations as evidenced by the links the reviewer provided at the beginning of the review. Studies like ours should be viewed as a major study that such agencies can include in their evaluations rather than one that will dictate the regulation itself. Should government organizations such as the EPA and ECHA find compelling evidence that a chemical is toxic, different countries often have very different ways of regulating a chemical. Therefore, sufficient and uniform regulation on a global level is not realistic at this point in time.

Original Submission: Comment 12

Some more minor points:

Line 31: No temporal trend data are included so how can the authors conclude that the clean energy sector is a growing source of PFAS release? It seems likely that this is true, but the study does not confirm an increasing trend.

Thank you for the feedback. The trends with time are supported by studies that document increasing use of LiBs and our own data showing bis-FASI occurrence in LiBs. The line the reviewer references is in the Summary Paragraph, and the support for this statement becomes more evident in the main manuscript. Nevertheless, we agree that the language should be softened for the Summary Paragraph. **Line 31** now states:

“...and confirm the clean energy sector as an unrecognized and potentially growing source of global PFAS release.”

28. R: This is a good modification.

Thank you for confirming.

Original Submission: Comment 13

Lines 278-279: It is not clear how representative these LIBs are of products on the market? The authors should have made efforts to discuss this with the LIB industry. As I mentioned above, my discussions have indicated that bis-FASIs are not widely used in LIBs. Is the LIB industry being deliberately misleading?

Thank you for the feedback. We have addressed this in our response to **Comment 10**. Here we will additionally note that **Table S7**, which is unchanged from the original submission, documents the source of batteries including common consumer products such as iPad, iPhone, Tesla, and others). We agree that more additional assessment of bis-FASI occurrence in commercial LiBs would be valuable, but such a wide survey of occurrence beyond the 17 LiBs that we considered was outside the scope of this study.

29. R: As noted in my above responses, the authors could have followed my recommendation to have a discussion with electrolyte manufacturers. I cannot share all the information I have personally gathered through such an exercise, although I want to be constructive.

Thank you for the feedback. We have addressed similar comments in our responses to **Comments 20-22**.

Original Submission: Comment 14

Lines 353: I don't see how global release is proven by this study? The monitoring was regional and biased to an area near to manufacturing. These statements are an overextrapolation.

Thank you for the feedback. We have addressed this in our response to **Comment 7**.

30. R: Thanks for this reponse, and all your other responses!

Thank you for the time and effort you invested in reviewing our work.

Original Submission: Comment 15

Note: Reviewer 2 did not provide point-by-point feedback regarding our responses to their original feedback, but the feedback and responses are provided below in gray for continuity. Their feedback to the revised manuscript begins following all comments to the original submission.

Reviewer #2 (Remarks to the Author):

This paper by Guelfo et al. examines the effects of bis-perfluoroalkyl sulfonimides (bis-FASIs), a novel class of PFAS found in lithium-ion batteries, to *Daphnia magna* and zebrafish. This paper is a comprehensive evaluation of bis-FASIs and may be the first paper to examine the environmental impacts of this class of PFAS compounds from manufacturing until disposal. Their environmental sampling efforts are particularly notable and useful to risk assessors, since they include data from many countries (USA, Antwerp, France) and matrixes (surface water, soil, sediment etc.). The experimental design is also strong, since the paper includes a wide range of samples and concentrations (5 concentrations and a control) that span levels found in the environment.

We would like to thank Reviewer 2 for their time and effort in reviewing our manuscript, and for the favorable view of our work. We really appreciate the value that this feedback brought to the work and have addressed each comment on a point by point basis below. We hope that we have alleviated remaining concerns regarding the manuscript.

Original Submission: Comment 16

The behavioral and mitochondrial tests conducted in the animals are standard. However, this reviewer was unable to find any data provided on growth and survival, which are critical toxicological endpoints, and this information should be included within the publication.

Thank you for the feedback. Information on lethality and immobilization in *Daphnia magna* (*D. Magna*) were provided in the original submission in **Figure S15**. No significant lethality was found through the exposure period. The experimental design for *D. Magna* exposures was based on United States Environmental Protection Agency Method EPA-821-R-02-012, which uses neonates and evaluates impacts of a 48-hour exposure. Neonate exposure over a 48-hour duration precludes evaluation of growth and reproduction.

We also measured mortality, hatching rate, and development in *Danio rerio* (zebrafish) during exposure to bis-FMeSI and field-collected water samples, but a description of these data was unintentionally omitted from the supplementary text. There were no significant differences from controls for these endpoints in any of the concentrations of bis-FMeSI or water samples tested. As noted in the edits summarized below, we have also tested three additional, higher concentrations (2,500,000, 25,000,000, and 250,000,000 ng/L) for acute endpoints (i.e., survival hatching rate, and development) in response to this reviewer feedback. These additional experiments confirm and extend findings reported in the previous version of the manuscript. Specifically, they verify lack of acute mortality even at very high, environmentally unrealistic concentrations.

The scope of this study was limited to very early life stages. Therefore, reproduction was not measured. However, we measured the standard length of larvae after hatching when we repeated the behavior assay experiments (see **Comment 26**). There were no differences between groups at 6 days post fertilization (dpf). To examine potential effects of developmental exposure on later life growth, we measured standard length again at 6 weeks post-fertilization, in clean (no chemical exposure) conditions. While this does not address long term changes in growth that may be attributable to early life exposure to bis-FMeSI, it does provide additional information to address the reviewer's concern regarding growth.

We have added this information as text and new figures to the supplementary text (see Toxicity: zebrafish) as well as the main manuscript. Specific edits are described below.

The following was added to the main manuscript (**lines 261-262**):

“Survival, growth, developmental teratogenicity, mitochondrial function in embryos at 30 hours post fertilization (hpf) and larval locomotion at 6 days post fertilization (dpf) were used to evaluate impacts of bis-FMeSI exposure...”

The following were added to the Materials and Methods (see **Zebrafish Exposures**):

“Three additional concentrations – 2500000, 25000000, 250000000 ng/L – were tested for acute endpoints of survival, hatching rate, and development at 144 hpf. There were 3 replicate dishes per treatment group. Larval locomotion assays had 9 replicate dishes across three biological replicates. Mitochondrial assays had 3 replicate dishes per experimental group (n=30).”

“Mortality was determined by lack of heartbeat, and those individuals were removed from the dish. Larvae were assessed for developmental deformities at 6 days post-fertilization (dpf), before larval locomotion assays (see description of assay below).”

“A subset of thirty larvae per bis-FMeSI experimental group were also imaged in lateral orientation with a Nikon SMZ1500 microscope with a Nikon DXM1200 digital camera and NIS-Elements 3.10 software. Standard lengths (mm) of these larvae were measured with ImageJ 1.52a software. Thirty larvae per group were placed in 10L tanks and placed on the AHAB system and fed Ziegler’s dry larval diet under the colony regimen for grow-out under clean conditions. At 6 weeks post-fertilization (wpf), these larvae were anesthetized with cold water, measured for standard length (mm), and returned to their tanks.”

The following text and figures were added to the supplementary text (see **Toxicity: zebrafish**):

“There were no differences from controls in any of the concentrations of bis-FMeSI or water samples tested for survival, hatching, or developmental deformities in zebrafish used for mitochondrial and locomotion assays (**Fig. S16**). In the additional three highest concentrations tested for acute endpoints only, the highest concentration (250,000,000 ng/L) had >80% of larvae with uninflated swim bladders and \approx 20% of larvae with pericardial and yolk sac edema. Body lengths were the same between larvae in all groups used for mitochondrial and locomotion assays (**Fig. S17A**). At 6 wpf, body lengths were mostly similar between experimental groups. There were statistically significant differences between fish that had been exposed to 25 ng/L bis-FMeSI compared to control and 25,000 ng/L (**Fig. S17B**). Likewise, fish in the 250 ng/L were different than those in control. This is very likely due to the lower tank density in 25 ng/L (n=8) and 250 ng/L (n=9) compared to other tanks such as control (n=17). It is well established in zebrafish that fish naturally grow larger when tank densities are lower. Therefore, this is very likely due to this rather than a treatment effect.”

Fig. S16. Acute toxicity endpoints in zebrafish larvae exposed to bis-FMeSI (A-C) or field collected water samples (D-F) including survival (A,D), hatching rate (B,E), developmental deformities (C,F), and standard lengths (D,H). Points and bars represent means \pm SEM. Abbreviations for types of developmental deformities: Spine – curvature of the spine; Tail – bending shortening, or alteration of the caudal tail; PE – pericardial edema; YSE – yolk sac edema; CF – craniofacial deformity; Fin – alteration of the pectoral fin or fin fold; SB – uninflated or less inflated swim bladder. No statistical differences between any groups ($P > 0.05$).

Fig. S17. Growth of zebrafish exposed to bis-FMeSI. Standard lengths (mm) at 6 days post-fertilization (A) and 6 weeks post-fertilization (B). Bars represent means \pm SEM. Statistical differences between groups ($P < 0.05$).

Original Submission: Comment 17

Furthermore, it is unclear to the reader why behavior and mitochondrial function were selected to evaluate toxicity of bis-FASIs and no other standard endpoints like growth and survival. There is no hypothesis presented in the introduction for why these endpoints were selected and there is no link to why behavior and/or mitochondrial respiration are expected to be impacted. It is also unclear how mitochondrial function at 48 hpf relates to behavior measured at 144 hpf. Overall, the field data and environmental sampling are the major strength of this paper, while the toxicological endpoints seem to offer very little evidence of dose response effect and need clarification.

Thank you for the feedback and the opportunity to clarify the rationale for the zebrafish experimental approach. Mitochondrial function was measured between 30-34 hours post fertilization (hpf) to examine developmental toxicity of bis-FMeSI to fundamental energy metabolic processes. Changes in mitochondrial dynamics can affect neurodevelopment and fish behavior via multiple mechanisms. For example, mitochondrial integrity is critical for neuronal function and formation and sustaining energetic needs for fish movement. However, without further studies, it is difficult to discern the precise link between mitochondrial function and behavior and our data indicate the need for future analyses. The link between behavior and energy production are also addressed in more detail in our response to **Comment 25**.

We have addressed questions about growth, survival, and reproduction in our response to **Comment 16**. Additionally, we have added rationale for the selected assays to the main manuscript, and in more detail in the supplementary text:

Lines 75-78 of the manuscript now state:

“When coupled with past and current challenges associated with PFAS such as perfluorooctanoic acid (PFOA) this illustrates the need for studies of bis-FASI occurrence, toxicity, and treatability. More information is available in the supplementary text.”

The supplementary text (see **Background**) states:

“Zebrafish larval locomotion is a widely used method for identification of neurobehavioral effects and an indicator of sub-lethal and sub-teratogenic toxicity resulting from chemical exposure. Behavioral alterations have been reported in zebrafish larvae at non-teratogenic concentrations of several types of PFAS. Chemical exposure has been shown to alter cellular energy metabolism directly by causing mitochondrial dysfunction. Mitochondrial function is also considered to be a biomarker for energy metabolism, and it is one that can be studied in a vertebrate, whole organism by use of embryonic zebrafish. While behavioral alterations have been shown following exposure to PFAS of a variety of structural subclasses, mitochondrial effects are considerably less well studied for these types of compounds particularly in zebrafish. Hagenaaers et al. (2013) reported PFOA caused an increase in mitochondrial permeability as well as a decrease in electron transport chain activity likely resulting from a decrease in ATP production. Similar evaluations of bis-FASIs are not available.

Original Submission: Comment 18

Direct comments:

Line 29-30: list the PFAS that bis-FASIs are comparable to.

Thank you for the feedback. We have edited **Line 30** of the manuscript. It now states that:

“Here we demonstrate that occurrence, ecotoxicity, and treatability of this novel class of PFAS are comparable to PFAS such as perfluorooctanoic acid...”

Original Submission: Comment 19

Lines 31-33: please include the median concentration. Giving just the max with no other context is not helpful to the reader.

Thank you for the feedback. We have edited **Lines 32-34** to state that:

“U.S. and European surface water, soil, and sediment measurements confirmed bis-FASI release internationally at a median concentration of 53 parts per trillion (ppt) and concentrations as high as 2,437 ppt.”

Original Submission: Comment 20

Lines 47-84: This main section does a great job of explaining how bis-FASIs end up in the environment, as well as the rationale for the different areas sampled. What is missing from this section is why the only way aquatic toxicity was assessed was through behavioral assays in zebrafish and daphnia and a mitochondrial function assay in zebrafish. What was the hypothesis of the test? Why were normal parameters of toxicity, such as growth reproduction or survival, not considered or measured?

Thank you for the feedback. Based on this **Comment 17** by Reviewer 2 and feedback from Reviewer 1 (e.g., **Comments 7 and 8**), Edits to the main manuscript and the rationale for the zebrafish experimental approach were described in our response to **Comment 17**. We also added additional rationale for *D. magna* testing to the supplementary text.

The supplementary text (see **Background**) now states that:

“*Daphnia magna* (*D. Magna*) have been used to evaluate the toxicity of individual toxicants and effluent wastes for more than 90 years. These organisms are excellent indicators of toxicity because they are sensitive to low concentrations of different toxicants (1mg/L – 100 µg/L). Further, they are easily maintained in lab cultures, reproduce asexually, eliminating genetic variability in the test population, and are a representative species at the bottom freshwater food chains. Most published *D. magna* toxicity testing data uses lethality as the main test endpoint. However, there has been a paradigm shift from lethality based toxicity testing to developing sublethal methods capable of identifying effects to environmentally relevant exposure. *D. magna* are an excellent candidate for sublethal toxicity testing as there are currently 48 sublethal effects identified across four classes (reproduction, swimming behavior, biochemical, and physiological changes). Perturbations to *D. magna* swimming behavior as a result of toxicant exposure is a consistently sensitive endpoint class with respect to dose.

Much of the sublethal endpoints identified in *D. magna* were identified using exposures to a variety of pharmaceuticals (e.g. antibiotics, beta blockers). Effects to swimming parameters have been observed as low as 500 ng/L. This effect level underscores the utility of *D. magna* as a sensitive indicator of toxicity. Further, this growing body of pharmaceutical effects data allow conclusions to be drawn between known human mechanism of action and the effects observed in *D. magna*. Compared to pharmaceuticals, there is a limited knowledge of the toxicity of PFAS to *D. magna*, none of which evaluate sublethal effects on swimming. Therefore, the aim of this work is to evaluate the effects of bis-FMeSI on *D. magna* at environmentally relevant concentrations reported in this study.”

Lastly, additional information on text that was added related to PFAS occurrence and treatability can be found in our responses to **Comments 7 and 8**.

Original Submission: Comment 21

Line 207: Did exposure to bis-FASIs affect the growth or survival of daphnia?

Thank you for the question. We have addressed this in our response to **Comment 16**.

Original Submission: Comment 22

Lines 222-225: There was no change in average velocity, just a change in the variance of velocity. Is this how one would define a neuroactive effect?

Thank you for the question. Based on this feedback, we recognize that the manuscript would benefit from additional clarification regarding use of variance as an indicator of toxicological effects and the implications of the results of bis-FMeSI exposure. Because the manuscript length needs to be consistent with the journal format, we have edited the main manuscript for clarity and provided additional discussion regarding use of variance testing in toxicological assessments in the supplementary text.

Lines 228-234 now state that:

“Prior studies have found heterogenous variance in the absence of significant changes to the mean can be an earlier and more sensitive indicator of toxicological effects in 48-hr *D. magna* exposures. Prior studies have noted that impacts to *D. magna* swimming behavior often indicate a neuroactive effect because swimming in *D. magna* is controlled by the nervous system. Although mechanism of action would require further confirmation, results here indicate that bis-FMeSI has an effect on swimming velocity at concentrations as low as 10 ng/L, consistent with levels broadly detected in monitoring regions included in this study.”

The following was added to the supplementary text (see **Toxicity: *D. magna***):

“As noted in the main manuscript, there is recognition in the literature that heterogeneity of variance can be an earlier and more sensitive indicator of toxicological effects relative to changes in the mean. Studies attribute changes in variance to experimental factors (e.g., analytical variability), genetic variability, and non-genetic phenotypic response. In this study, bis-FMeSI exposure concentration was the only change in experimental condition, and *D. magna* are genetically identical to each other. Thus, changes in variance of a monitored endpoint relative to the control are attributable to changes in exposure concentration. As noted above, changes in variance are evaluated by comparing variance at each exposure level to the control using a Chi² test. Because variance is heterogenous, criteria for use of ANOVA (i.e., equal variance, normal data distribution) are not met, so there are no comparisons between doses. Studies have identified that differences in variance relative to controls may occur only in some exposure concentrations (e.g., **Fig. 2** of the manuscript), but still recognize heterogenous variance as an early toxicological indicator.

In this study, heterogenous variance indicates that exposure to bis-FMeSI has a significant effect on the variance of swimming velocity at concentrations as low as 10 ng/L. The means of swimming velocity did not differ significantly, but the outcomes of variance testing still demonstrate an effect of exposure. This is similar to prior *D. magna* studies that observed no effect of 48-hr contaminant exposure on mean oxygen consumption, but significant changes in the variance of oxygen consumption. Their results, results of additional prior work, and results herein highlight that testing homogeneity of variance has utility beyond its traditional use as criteria for use of ANOVA. The impacted endpoint (velocity) indicates an effect on swimming behavior, which is controlled in *D. magna* by the central nervous system. Because of this, prior studies have noted that changes in swimming behavior may be indicative of a neuroactive effect. Additional study would be required to confirm the mechanism of action of bis-FMeSI exposure.”

Original Submission: Comment 23

Lines 237: The distance/velocity figures here with just mean +/- SD are not very useful in their current format for publication. A box plot, a violin plot, and/or a plot with the individual replicates displayed, would do a better job visually displaying the change in variance the authors are attributing to bis-FASI exposure.

Thank you for the opportunity to improve the utility of the *D. magna* and zebrafish figures. Fig. 2 of the manuscript was changed to a box and whisker format for data on percent time swimming, distance, and velocity, as shown below.

Original Submission: Comment 24

Lines 225-227: The lack of dose response, or effect to changes in the distance travelled, swimming duration or velocity, indicates to me a lack of toxicological effect.

Thank you for this comment. We have addressed this in our response to **Comment 22**.

Original Submission: Comment 25

Lines 253-255: What do decreases in these parameters mean for overall mitochondrial function? According to Chacko et al. 2014 Clinical Science (doi: 10.1042/CS20140101), damaged/stressed mitochondria tend to have increased basal respiration and proton leak, along with reduced reserve capacity and ATP linked respiration. In this paper, the results seem to be decreased across the board which makes it challenging to interpret what they mean for overall mitochondrial health.

Thank you for the question. Chemical exposure can impair mitochondrial function by multiple mechanisms and a decrease or increases in function compared to the control group indicate mitochondrial dysregulation or compromised mitochondrial health. However, extensive further studies are needed to interpret the mechanisms underlying altered function. It is widely accepted in the field that interpretation and inferences of these data have to be made in the context of the control samples, given the complexity of whole organisms and the ability to induce adaptive responses following exposure³⁰. Please note that we have thoroughly reviewed the discussion by Chacko *et al.* (2014)³¹, where they primarily derive conclusions based on cell culture studies and not whole animal studies, where adaptive responses are likely to significantly differ.

Chacko *et al.* (2014) do rightly say that this parameter should be interpreted with information from the rest of the mitochondrial profile³¹. As the reviewer stated, we observed decreases in multiple parameters. This is likely due to decreased substrate availability or compromised mitochondrial mass or integrity. This is suggested by Chacko *et al.* (2014) whether it be linked to ATP synthesis or other mitochondrial parameters³¹. Brand *et al.* (2011) also cites substrate supply as a reason for changes in basal respiration and maximal mitochondrial respiration³². These authors state that a decrease in maximum mitochondrial respiration is a strong indicator of potential mitochondrial disfunction. Together, these data show that zebrafish embryos exposed to high concentrations of bis-FMeSI do not produce as much energy in a resting state and are unable to make up that deficiency even with an artificial energy demand. We have added some of this information to the supplementary text as described below. However, we refrained from elaborating on the mitochondrial data in the current study, without in-depth mechanistic analyses that our laboratory is setup to do and look forward to conducting in the future.

The following was added to the supplementary text (see **Toxicity: Zebrafish**):

“Decreases in basal respiration indicate that embryos exposed to high concentrations of bis-FMeSI do not produce as much energy in a resting state. These differences are reflected in decreased maximum mitochondrial respiration and spare capacity, which show that these embryos are unable to make up this energy deficit even with an artificial energy demand. Such decreases in energy may be due to decreased substrate availability or compromised mitochondrial mass or integrity.”

Additionally, the supplementary text (see **Toxicity: Zebrafish**) now states that:

“While zebrafish behavioral changes have been reported following exposure to a variety of PFAS, bioenergetics is considerably less well studied. This is interesting because behavior and energy production are, in many ways, linked. Many mitochondrial defects have been shown to affect the nervous system. For example, Huang et al. (2023) found that exposure to PFHpA decreased ATP-linked respiration in zebrafish embryos and reduced locomotor activity in larvae. Patel et al. (2022) reported changes in expression levels of mitochondrial-related genes but no changes in

mitochondrial function or larval locomotion. The decrease that we observed in both the behavior test as well as the mitochondrial assay suggests that exposure to bis-FMeSI is energetically costly and provides information on potential modes of action of bis-FMeSI. Increases that we observed at the lowest concentrations may be related to more than one mechanism (e.g., bioenergetics and neurological function). Additional experiments are needed to resolve these questions.”

Original Submission: Comment 26

256-259: Are these behavioral results toxicity or just random variance? Are non-monotonic responses in LPR typical for zebrafish exposed to PFAS? The effect at 25 ng/L looks within the realm of control fish since the controls on the field panel looks to be higher than the fish exposed to 25 ng/L. Demonstrating that this effect is reproducible, along with testing a few concentrations below this “threshold” would convince readers that there is something going on here.

Thank you for the question and the opportunity to clarify questions surrounding reproducibility. To address these concerns, **the project team repeated the behavioral assay experiment, and as part of this effort, introduced a lower test concentration (2.5 ng/L). This more than doubled our sample size.** In this type of assay, we expect inter-individual variability. Fitzgerald et al. (2019) has stated that variability is lowest during dark phases of this type of test³³. In order to determine the variance within our data, we calculated coefficients of variation (CV) for each experimental group in each light/dark phase for both bis-FMeSI and found this was the case. Additionally, CVs were similar within phase. We have included the tables below in the supplementary data (**Data S12-S13**). We are also including the figure below showing means for each experimental group across the three runs/biological replicates for bis-FMeSI. Within each run, we observed very similar trends as that of the combined data now shown in the main manuscript (revised **Fig. 3**).

There are a wide range of patterns in behavioral responses following exposure to different PFAS as reported by Gaballah et al. (2020), Menger et al. (2020), and Rericha et al. (2021)³⁴⁻³⁶. We are not able to say if non-monotonic responses to PFAS are “typical” for zebrafish as there are not enough studies to determine this with a high level of confidence yet. What we can say there is some data available in zebrafish that shows low concentrations can cause effects when high concentrations do not. For example, figures in Ulhaq *et al.* (2013) and Wasel *et al.* (2022) showed some greater effects at lower concentrations of some, but not all, of the PFAS tested^{37,38}. Ulhaq et al. (2013) correlates behavioral changes to PFAS chain length³⁷. Other studies, such as Rericha et al. (2022) provide some evidence that it may be related to the functional group³⁹. A systematic review of epidemiological studies in humans indicates that PFAS have the potential for non-monotonic dose response curves, particularly in response to endocrine disrupting compounds, with low concentrations/exposures exhibiting more disruptive effect than high concentrations/exposures⁴⁰. For example, non-monotonic dose response curves in serum lipid concentrations have been associated with prenatal exposures to PFOS and PFOA⁴¹.

The following table was added to the supplementary data file (tab **Data S12**)

Coefficients of variation for each experimental bis-FMeSI group in each phase of the zebrafish locomotion assay							
	Control	2.5 ng/L	25 ng/L	250 ng/L	2,500 ng/L	25,000 ng/L	250,000 ng/L
Light 1	228.87	161.55	197.74	234.17	252.94	205.09	210.02
Dark 1	66.39	43.88	64.00	62.95	71.97	79.88	64.90
Light 2	292.02	176.45	207.92	209.89	316.73	258.69	275.76
Dark 2	69.04	50.95	69.52	75.98	72.55	74.00	62.74

Additionally, we added the following table to the supplementary data file as tab **Data S13**).

Coefficients of variation for each experimental water sample group in each phase of the zebrafish locomotion assay

	Control	Vial Blank	MN 15	MN 22	MN 4
Light 1	178.20	150.73	138.84	115.54	126.38
Dark 1	70.77	55.27	39.21	40.29	47.26
Light 2	180.64	165.70	190.84	124.29	148.63
Dark 2	76.82	61.50	52.47	42.00	51.64

The following was added here for the reviewer’s benefit to demonstrate that within each run, trends were similar to the combined data shown in the manuscript (revised **Fig. 3**)

Lastly, **Fig. 3** of the main manuscript has been revised as follows:

Fig. 3. Zebrafish larval locomotion.

Mean distance traveled (mm) during light/dark phases of the assay (mm/40min) by larvae exposed to 25-250,000 ng/L bis-FMeSI (A) and field-collected aqueous samples MN 4 (including duplicate), MN 22, and MN 15 (B). Bars represent means ± standard error of the mean. Different letters represent statistical differences within phase ($P < 0.05$; $n = 30$).

Original Submission: Comment 27

267-272: Why is the zebrafish behavior displayed like this? The materials and methods listed this assay as a 50-minute LPR. It is unclear what this data is showing. Is this light and dark cycle data averaged together? If so, why was this done? Did exposure to these compounds affect behavior in light or dark phases, did it affect how the fish transition between phases? Overall, this is an unusual way of evaluating the LPR assay and I would have to ask, why perform light/dark transitions if averaging it all together?

Thank you for the feedback on the figure format. We originally displayed behavior data this way to simplify the results. The reviewer is correct that this is a 50-min assay. However, we typically do not include data from the 10-min habituation phase in our analyses. We have updated the figure in the main manuscript to separate out light and dark phases. The revised figure is included in the manuscript (**Fig. 3**) and is also shown in our response to **Comment 26**. This separation with additional replicates shows an even greater increases in hyperactivity at low concentrations, both in the light and dark phases.

Original Submission: Comment 28

357-360: there is insufficient evidence in this study to make such strong conclusions. Based on the evidence in this study there seems to be no dose-dependent effects on behavior and potential mitochondrial dysfunction at 250,000 ng/L.

Thank you for the feedback. We agree with the reviewer that the effects of concentration that we observed for zebrafish locomotion did not follow a typical dose-response. However, non-monotonic dose responses or inverse dose response effects are emerging in sub-lethal assays (e.g. prolactin level changes with DDE and BPA are classic examples), and warrants significant attention in the field of toxicology, as also evident from our assays. More details are provided in our response to **Comment 26**, and our response to that feedback also demonstrates through addressing questions about reproducibility that non-monotonic response is not, as the reviewer had questioned, “random noise.”

This compound behaves differently than what we might expect for typical lipophilic or hydrophilic compounds, given that very low concentrations have the greatest effect on zebrafish locomotion. The magnitude of this effect may be up to interpretation, but there are statistical differences that demonstrate bis-FMeSI is showing strong effects at very low concentrations in both zebrafish and *D. magna*. This leads us to the conclusion that this compound is in need of further study is needed with this compound to identify the target(s) and mechanisms by which bis-FMeSI is acting upon.

Original Submission: Comment 29

Methods/Results: a summary table of measured bis-FMeSI for the daphnia and zebrafish exposures should be presented somewhere to allow better interpretation of the nominal values presented in this study.

Thank you for the feedback. The measured bis-FMeSI concentrations for *D. magna* exposure were provided in the original submission (**Table S6**), so no edits were made in response to this feedback for that organism model. The equivalent data for zebrafish have been added to the revised manuscript.

We added the following table to the supplementary data file as tab Data S16.

Nominal Concentration (ng/L)	Measured Concentration (ng/L)	Standard Deviation
Day 0		
Control	ND	ND
2.5	2.72	0.26
25	22.9	4.5
250	200	13
2,500	1,870	210
25,000	17,000	2,000
250,000	152,000	9,000
Day 6		
Control	ND	ND
2.5	3.06	0.16
25	23.0	3.6
250	214	9
2,500	1,890	230
25,000	17,000	1,000
250,000	181,000	13,000
2,500,000	2,490,000	270,000
25,000,000	23,500,000	2,100,000
250,000,000	257,000,000	16,000,000

31. Reviewer #2 (Remarks to the Author):

I've completed the review of the revised version of this manuscript by Guelfo et al. as well as read through their thorough responses to comments. The authors have responded and addressed the questions and concerns of the reviewers. I have a few minor comments/suggestions for the revised figures. I would like to applaud the authors for strengthening the ecotoxicological data set by expanding the range of concentrations tested. This has substantially strengthened the manuscript. I think the revised figure edits look great and that the revised figures will more helpful and comprehensive for readers.

Thank you for the time and effort you invested in reviewing both the original submission and the revised manuscript. We are glad to hear that our revisions addressed the majority of Reviewer 2's concerns about the work and have addressed additional feedback on a point-by-point basis below.

32. Comments:

General figure comments: The figure captions need to be updated (fig 2 for example is still describing the old plots and not what a box plot is) or fig 3 (what is the y axis, mm/min? the caption says mm/40 min but that cant be right).

We thank the reviewer for this comment. The reviewer is correct that a modification is needed to the Fig. 2 caption. Therefore, the caption now reads (**lines 244-249**):

“Swimming track density for *D. magna* exposed to 0 (top left) and 5000 (top right) ng/L bis-FMeSI. Swimming parameters of distance and percent time swimming (bottom left), and swimming velocity (bottom right). **In box and whisker plots, the lower and upper extent of each box represent the 25th and 75th percentiles and whisker represent the 1.5 interquartile range (1.5 IQR). Data beyond the 1.5 IQR are considered outliers.** Different letters reflect statistical differences in the variability between treatments ($P \leq 0.002$ $n=10$).”

We also see how the caption for Fig. 3 was confusing. Each alternating light/dark phase is 10 minutes long, with two light and two dark phases making up a total of 40 min for the whole assay. The y-axis of Fig. 3 is labelled as mm and our original caption referred to mm/40 min assay. We have revised the figure caption to be clearer. **Lines 284-287** now state that:

“Mean distance traveled (mm) during **four, 10 min** phases of the assay (**two light and two dark**) by larvae exposed to 25-250,000 ng/L bis-FMeSI (A) and field-collected aqueous samples MN 4 (including duplicate), MN 22, and MN 15 (B). Bars represent means \pm standard error of the mean. Different letters represent statistical differences within phase ($P < 0.05$; $n=30$). “

33. Figure 3: I appreciate the authors sharing the three runs of data they pooled to create Figure 3. The first run looks to be statistically significantly different (controls for example are almost half of the other two runs) from the other two for all concentrations tested? Were the runs as assessed to determine if they were different before they were pooled, and if they are significantly different why would you pool them? It might make more sense to exclude the hypoactive run as a “bad run/animals not behaving in a normal manner due to any number of factors” and pool the two “normal” runs instead. There are so many factors that could affect the behavior of zebrafish from time of day to who

is handling the animals. It also looks like the first run is missing a data point. Overall, these three runs really indicate just how variable behavior data is in zebrafish.

We thank the reviewer for highlighting the differences between the control groups from each run. We conduct many exposure assays with zebrafish, and we do find there is variability from run to run (batch effects). However, we primarily focus on the treatment effect from the exposure and consider the consistency in the treatment effect despite the variability within a run. As indicated by the data presented here, despite the run variability, we see that the treatment effects are consistent and therefore we have decided to include these data to show the persistent effects of this compound across multiple experimental runs.

34. Line 288: I disagree with this line stating low concentrations may be more harmful. The data certainly indicate lower concentrations may elicit hyperactivity in the fish, but it's a big jump to say that hyperactivity equals harm. Also, the mitochondrial dysfunction seemed to follow a dose response with higher doses causing lower mitochondrial function.

We respectfully disagree with the reviewer's notion that hyperactivity does not equate to harm. Hyperactivity could be indicative of multiple neurological deficits such as anxiety and may have significant ecological consequences such as reduced capacity to find food or increased metabolic demand overtime. Essentially, we consider deviation from the control levels in behavior is likely to affect the organismal fitness in the natural environment but do agree with the reviewer that we will need more targeted studies to ascertain ecological impacts.

Regarding the statistical differences we found in the behavioral assay compared to the mitochondrial assay, there are a few factors that may contribute to this. The first is the age of the animal. The behavioral assay was done with larvae 6 days post-fertilization and the mitochondrial assay was done with embryos 30 hours post-fertilization. It is possible that the presence of a chorion may have impacted the actual dose that embryos received in that only a fraction of the high dose made it across the chorion. Another is the sensitivity of the endpoint, or the amount of chemical needed to elicit a response. While, mitochondria are particularly vulnerable to environmental toxicants, their dysfunction involves multiple cellular processes.

Nevertheless, we have removed the statement that, "low concentrations may be more harmful than higher concentrations. An earlier statement in the same paragraph (**lines 267-268**) states that, "Larva exposed to the lowest concentrations (i.e., 2.5 and 25 ng/L) were the most hyperactive." In our view, this conveys the key message, and will hopefully alleviate concerns regarding use of the word "harmful."

Original Submission: Comment 30

Reviewer #3 (Remarks to the Author):

The paper present unexpected and worrying data showing the release of PFAS into the environment near lithium-ion battery related production facilities. The work is original and extremely important with respect to the "green credentials" of lithium-ion batteries and the race to nett zero. The paper is entirely self-contained and the data presented does support the conclusions drawn. I can see no flaws in the analyses-but I must confess to not being an expert in analysis. As far as I can see, the work is methodical and precise and good science. There is very great experimental detail provided in the paper which would allow the tests to be reproduced. I recommend publication.

We would like to thank Reviewer 3 for their time and effort in reviewing our work, and we would also like to express our appreciation for their favorable view of the manuscript.

References Cited

1. ECHA. Lithium bis(trifluoromethylsulfonyl)imide Ecotoxicological Summary. <https://echa.europa.eu/sk/registration-dossier/-/registered-dossier/18080/6/1> (2020).
2. USEPA. *ORD Human Health Toxicity Value for Lithium Bis[(Trifluoromethyl)Sulfonyl]Azanide (HQ-115)*. <https://nepis.epa.gov/Exe/ZyPDF.cgi/P10188AG.PDF?Dockey=P10188AG.PDF> (2023).
3. Kowalska, D., Maculewicz, J., Stepnowski, P. & Dołżonek, J. Ionic liquids as environmental hazards – Crucial data in view of future PBT and PMT assessment. *Journal of Hazardous Materials* **403**, 123896 (2021).
4. Torrecilla, J. S., Palomar, J., García, J. & Rodríguez, F. Effect of Cationic and Anionic Chain Lengths on Volumetric, Transport, and Surface Properties of 1-Alkyl-3-methylimidazolium Alkylsulfate Ionic Liquids at (298.15 and 313.15) K. *J. Chem. Eng. Data* **54**, 1297–1301 (2009).
5. Montalbán, M. G., Hidalgo, J. M., Collado-González, M., Díaz Baños, F. G. & Villora, G. Assessing chemical toxicity of ionic liquids on *Vibrio fischeri*: Correlation with structure and composition. *Chemosphere* **155**, 405–414 (2016).
6. Matzke, M., Thiele, K., Müller, A. & Filser, J. Sorption and desorption of imidazolium based ionic liquids in different soil types. *Chemosphere* **74**, 568–574 (2009).
7. Stepnowski, P., Mrozik, W. & Nichthauser, J. Adsorption of Alkylimidazolium and Alkylpyridinium Ionic Liquids onto Natural Soils. *Environ. Sci. Technol.* **41**, 511–516 (2007).
8. Dołżonek, J. *et al.* Membrane partitioning of ionic liquid cations, anions and ion pairs – Estimating the bioconcentration potential of organic ions. *Environmental Pollution* **228**, 378–389 (2017).
9. Droge, S. T. J. Membrane–Water Partition Coefficients to Aid Risk Assessment of Perfluoroalkyl Anions and Alkyl Sulfates. *Environ. Sci. Technol.* **53**, 760–770 (2019).
10. UNEP. *Proposal to List Long-Chain Perfluorocarboxylic Acids, Their Salts and Related Com-Pounds in Annexes A, B and/or C to the Stockholm Convention on Persistent Organic Pollutants*.
11. US EPA. *Drinking Water Health Advisory for Perfluorooctanoic Acid (PFOA)*. (US Environmental Protection Agency, 2016).
12. US EPA. *Drinking Water Health Advisory for Perfluorooctane Sulfonate (PFOS)*. (US Environmental Protection Agency, 2016).
13. US EPA. *IRIS Toxicological Review of Perfluorohexanoic Acid [PFHxA, CASRN 307-24-4] and Related Salts*. (US Environmental Protection Agency, 2023).
14. US EPA. *Maximum Contaminant Level Goal (MCLG) Summary Document for a Mixture of Four Per- and Polyfluoroalkyl Substances (PFAS): HFPO-DA and Its Ammonium Salt (Also Known as GenX Chemicals), PFBS, PFNA, and PFHxS*. (US Environmental Protection Agency, 2023).
15. BAuA. *Annex XV Restriction Report, Proposal for a Restriction, Substances Name: Per and Polyfluoroalkyl Substances (PFASs)*. <https://echa.europa.eu/documents/10162/f605d4b5-7c17-7414-8823-b49b9fd43aea> (2023).
16. Wang, Z. *et al.* A New OECD Definition for Per- and Polyfluoroalkyl Substances. *Environ. Sci. Technol.* **55**, 15575–15578 (2021).

17. ASTDR. *3M Chemolite: Perfluorochemical Releases at the 3M Cottage Grove Facility*. <https://www.health.state.mn.us/communities/environment/hazardous/docs/sites/washington/3mcg0205.pdf> (2005).
18. Arp, H. P. H. & Hale, S. E. Assessing the Persistence and Mobility of Organic Substances to Protect Freshwater Resources. *ACS Environ. Au* **2**, 482–509 (2022).
19. Hale, S. E., Arp, H. P. H., Schliebner, I. & Neumann, M. Persistent, mobile and toxic (PMT) and very persistent and very mobile (vPvM) substances pose an equivalent level of concern to persistent, bioaccumulative and toxic (PBT) and very persistent and very bioaccumulative (vPvB) substances under REACH. *Environ Sci Eur* **32**, 155 (2020).
20. European Commission. Directorate General for the Environment., Milieu Ltd., Ökopol., Risk & Policy Analysts (RPA)., & RIVM. *Study for the Strategy for a Non-Toxic Environment of the 7th Environment Action Programme: Final Report*. (Publications Office, LU, 2017).
21. Oxford University Press. Oxford English Dictionary. <https://www.oed.com/?tl=true> (2023).
22. Guelfo, J. L. *et al.* Evaluation and Management Strategies for Per- and Polyfluoroalkyl Substances (PFASs) in Drinking Water Aquifers: Perspectives from Impacted U.S. Northeast Communities. *Environmental Health Perspectives* **126**, 065001 (2018).
23. Rensmo, A. *et al.* Lithium-ion battery recycling: a source of per- and polyfluoroalkyl substances (PFAS) to the environment? *Environ. Sci.: Processes Impacts* 10.1039/D2EM00511E (2023) doi:10.1039/D2EM00511E.
24. Arthur, S. D. *et al.* Nonaqueous electrolyte compositions comprising lithium oxalato phosphates. (2022).
25. 360 Research Reports. *Global LiTFSI Market Growth 2023-2029*. 92 (2023).
26. Arkema USA. ARKEMA HPP, PVDF Electrode Binders & Separator Coatings. <https://hpp.arkema.com/en/markets-and-applications/renewable-energy/lithium-ion-battery/> (2022).
27. 3M. 3M™ Battery Electrolyte HQ-115. https://www.3m.com/3M/en_US/p/d/b00005989/ (2022).
28. Solvay. *LiTFSi: Lithium Salt for Safe and Performing Batteries*. <https://www.google.com/url?sa=t&rct=j&q=&esrc=s&source=web&cd=&ved=2ahUKEwjLisbPppH9AhWifDABHWgSCtIQFnoECBMQAQ&url=https%3A%2F%2Fwww.solvay.com%2Fen%2FdownloadDocument%3FfileId%3DT5Yk9C04xcifrkl8Yo%26fileName%3D24998%2520LiTFSI%2520for%2520Safe%2520and%2520Performing%2520Batteries%2520v2%26base%3DFAST&usg=AOvVaw1FZa-96nMjQ8KDJ3tSgkWa> (NA).
29. Gaber, N., Bero, L. & Woodruff, T. J. The Devil they Knew: Chemical Documents Analysis of Industry Influence on PFAS Science. *Annals of Global Health* **89**, 37 (2023).
30. Meyer, J. N., Hartman, J. H. & Mello, D. F. Mitochondrial Toxicity. *Toxicological Sciences* **162**, 15–23 (2018).
31. Chacko, B. K. *et al.* The Bioenergetic Health Index: a new concept in mitochondrial translational research. *Clinical Science* **127**, 367–373 (2014).
32. Brand, M. D. & Nicholls, D. G. Assessing mitochondrial dysfunction in cells. *Biochemical Journal* **435**, 297–312 (2011).
33. Fitzgerald, J. A., Kirla, K. T., Zinner, C. P. & Vom Berg, C. M. Emergence of consistent intra-individual locomotor patterns during zebrafish development. *Sci Rep* **9**, 13647 (2019).
34. Gaballah, S. *et al.* Evaluation of Developmental Toxicity, Developmental Neurotoxicity, and Tissue Dose in Zebrafish Exposed to GenX and Other PFAS. *Environ Health Perspect* **128**, 047005 (2020).

35. Menger, F., Pohl, J., Ahrens, L., Carlsson, G. & Örn, S. Behavioural effects and bioconcentration of per- and polyfluoroalkyl substances (PFASs) in zebrafish (*Danio rerio*) embryos. *Chemosphere* **245**, 125573 (2020).
36. Rericha, Y. *et al.* Behavior Effects of Structurally Diverse Per- and Polyfluoroalkyl Substances in Zebrafish. *Chem. Res. Toxicol.* **34**, 1409–1416 (2021).
37. Ulhaq, M., Örn, S., Carlsson, G., Morrison, D. A. & Norrgren, L. Locomotor behavior in zebrafish (*Danio rerio*) larvae exposed to perfluoroalkyl acids. *Aquatic Toxicology* **144–145**, 332–340 (2013).
38. Wasel, O., Thompson, K. M. & Freeman, J. L. Assessment of unique behavioral, morphological, and molecular alterations in the comparative developmental toxicity profiles of PFOA, PFHxA, and PFBA using the zebrafish model system. *Environment International* **170**, 107642 (2022).
39. Rericha, Y. *et al.* Sulfonamide functional head on short-chain perfluorinated substance drives developmental toxicity. *iScience* **25**, 103789 (2022).
40. Rappazzo, K., Coffman, E. & Hines, E. Exposure to Perfluorinated Alkyl Substances and Health Outcomes in Children: A Systematic Review of the Epidemiologic Literature. *IJERPH* **14**, 691 (2017).
41. Maisonet, M., Näyhä, S., Lawlor, D. A. & Marcus, M. Prenatal exposures to perfluoroalkyl acids and serum lipids at ages 7 and 15 in females. *Environment International* **82**, 49–60 (2015).

REVIEWERS' COMMENTS

Reviewer #1 (Remarks to the Author):

The manuscript is much improved after the revisions. There are still a few statements that irk me, but I will have to let some of these go. The work is valuable and it's important that it is published soon. The change in title makes me feel a lot more comfortable.

A few minor things for consideration:

"PFAS are recognized internationally as recalcitrant, mobile, and toxic environmental contaminants." In lines 26-27 of the abstract. This is an overgeneralized statement. PFAS are extremely diverse in their structures and properties. Jennifer Guelfo has previously pointed out this diversity in another paper that she authored. A common feature of all PFAS is their high persistence. Not all PFAS are toxic and mobile (e.g. PTFE). Suggest rewording this sentence.

"Despite this, virtually nothing is known about environmental impacts of bis-FASIs released during LiB manufacture, use, and disposal." In lines 27-28 of the abstract. This is rather strong considering that previous work has identified them in batteries and some of their properties are known. I'd prefer, "they are poorly understood" rather than "virtually nothing is known".

In the introduction the authors now point out some of the other environmental issues with LiBs. I'd like this strengthened if possible by adding one more sentence. For example, a very well-read paper that could be referred to is the study by Notter et al. <https://pubs.acs.org/doi/10.1021/es903729a>. It still remains one of the best read papers in ES&T. They don't mention PFAS of course and the largest impacts are considered to be "the supply of copper and aluminum for the production of the anode and the cathode, plus the required cables or the battery management system."

Looking forward to seeing the paper in print and citing it in the future.

Reviewer #2 (Remarks to the Author):

I've completed the review of the revised version of this manuscript by Guelfo et al. as well as read through their thorough responses to the second round of comments. The authors have responded and

addressed the questions and concerns that I had. Their responses to reviewer 1's comments seem appropriate to me, but I will defer to reviewer 1's expertise. I look forward to seeing this paper published. The research and data contained in this manuscript will be helpful and of interest to many folks in academia, government and industry.

RESPONSE TO COMMENTS

Lithium-ion battery components are at the nexus of sustainable energy and environmental PFAS release

REVIEWERS' COMMENTS

Reviewer #1 (Remarks to the Author):

1. **The manuscript is much improved after the revisions. There are still a few statements that irk me, but I will have to let some of these go. The work is valuable and it's important that it is published soon. The change in title makes me feel a lot more comfortable.**

We would again like to thank Reviewer 1 for the time they invested in review of our manuscript. We appreciate the value that they have placed in our work, and we are glad that we were able to arrive at modifications to the manuscript that (mostly) satisfied all sides. We feel the manuscript is much stronger after careful consideration of all comments provided.

2. **A few minor things for consideration:**

“PFAS are recognized internationally as recalcitrant, mobile, and toxic environmental contaminants.” In lines 26-27 of the abstract. This an overgeneralized statement. PFAS are extremely diverse in their structures and properties. Jennifer Guelfo has previously pointed out this diversity in another paper that she authored. A common feature of all PFAS is their high persistence. Not all PFAS are toxic and mobile (e.g. PTFE). Suggest rewording this sentence.

Thank you for the feedback. We agree that this sentence could be less generalized and have edited the statement in the Abstract to read as follows:

“PFAS are recognized internationally as recalcitrant contaminants, a subset of which are known to be mobile, and toxic.”

3. **“Despite this, virtually nothing is known about environmental impacts of bis-FASIs released during LiB manufacture, use, and disposal.” In lines 27-28 of the abstract. This is rather strong considering that previous work has identified them in batteries and some of their properties are known. I'd prefer, “they are poorly understood” rather than “virtually nothing is known”.**

Thank you for the feedback. Reviewer 1 is correct that prior studies have noted their use in batteries, and some of their properties are known. We note that we had to reduce the abstract length ~50% at the request of the journal, and have essentially no words left in our word limit. Therefore, we have made a minor modification to soften the statement in the abstract. It now reads:

“Despite this, little is known about environmental impacts of bis-perfluoroalkyl sulfonimides released during lithium-ion battery manufacture, use, and disposal.”

4. In the introduction the authors now point out some of the other environmental issues with LiBs. I'd like this strengthened if possible by adding one more sentence. For example, a very well-read paper that could be referred to is the study by Notter et al. <https://pubs.acs.org/doi/10.1021/es903729a>. It still remains one of the best read papers in ES&T. They don't mention PFAS of course and the largest impacts are considered to be "the supply of copper and aluminum for the production of the anode and the cathode, plus the required cables or the battery management system."

Looking forward to seeing the paper in print and citing it in the future.

Thank you, we are looking forward to sharing this work with a wider audience! We also appreciate the suggestion for an additional citation. Reviewer 1 references language we added to the manuscript about other environmental issues with LiBs in the Introduction. We assume they meant the language added in the conclusions (now labeled as "Discussion") in the last revision because similar edits were not made to the Introduction. We agree that the paper cited by the reviewer is relevant and important. We have added a citation to this work and modified the relevant section in the conclusions to read as follows:

"Results of this study provide a clear indication that the impacts of manufacturing, use, and end-of-life management associated with infrastructure components such as LiBs require additional consideration along with other issues such as resource recovery. This includes environmental impacts associated with other fluorinated, but non-PFAS LiB electrolytes PF_6^- and BF_4^- detected in drinking water and with metals (e.g., Cu, Al) used in LiB electrodes, cables, and battery packs."

Reviewer #2 (Remarks to the Author):

5. I've completed the review of the revised version of this manuscript by Guelfo et al. as well as read through their thorough responses to the second round of comments. The authors have responded and addressed the questions and concerns that I had. Their responses to reviewer 1's comments seem appropriate to me, but I will defer to reviewer 1's expertise. I look forward to seeing this paper published. The research and data contained in this manuscript will be helpful and of interest to many folks in academia, government and industry.

We would like to extend our appreciation to Reviewer 2 for their support of our manuscript and the effort invested in reviewing our work.